# Online Reward-Weighted Fine-Tuning of Flow Matching with Wasserstein Regularization

**Jiajun Fan**[1*]**, Shuaike Shen**[2†]**, Chaoran Cheng**[1]**, Yuxin Chen**[3†]**, Chumeng Liang**[4†]**, Ge Liu**[1*]

[1] University of Illinois Urbana-Champaign, [2] Zhejiang University.

[3] Tsinghua University, [4] University of Southern California.

[1] {jiajunf3, chaoran7, geliu}@illinois.edu

[2,3,4] {shenshuaike256, itsme.yxchen, caradryan2022}@gmail.com

## Abstract

Recent advancements in reinforcement learning (RL) have achieved great success in fine-tuning diffusion-based generative models. However, fine-tuning continuous flow-based generative models to align with arbitrary user-defined reward functions remains challenging, particularly due to issues such as policy collapse from overoptimization and the prohibitively high computational cost of likelihoods in continuous-time flows. In this paper, we propose an easy-to-use and theoretically sound RL fine-tuning method, which we term Online Reward-Weighted Conditional Flow Matching with Wasserstein-2 Regularization (ORW-CFM-W2). Our method integrates RL into the flow matching framework to fine-tune generative models with arbitrary reward functions, without relying on gradients of rewards or filtered datasets. By introducing an online reward-weighting mechanism, our approach guides the model to prioritize high-reward regions in the data manifold. To prevent policy collapse and maintain diversity, we incorporate Wasserstein-2 (W2) distance regularization into our method and derive a tractable upper bound for it in flow matching, effectively balancing exploration and exploitation of policy optimization. We provide theoretical analyses to demonstrate the convergence properties and induced data distributions of our method, establishing connections with traditional RL algorithms featuring Kullback-Leibler (KL) regularization and offering a more comprehensive understanding of the underlying mechanisms and learning behavior of our approach. Extensive experiments on tasks including target image generation, image compression, and text-image alignment demonstrate the effectiveness of our method, where our method achieves optimal policy convergence while allowing controllable trade-offs between reward maximization and diversity preservation.

## 1 Introduction

Generative models have achieved remarkable success in producing high-fidelity data across various domains, including text and images (Ouyang et al., 2022a; Esser et al., 2024). Among these, continuous flow-based models have gained attention for their more concise and flexible design of the ODE-based denoising process to model complex data distributions (Tong et al., 2024a; Lipman et al., 2023). However, fine-tuning such flow-based models to align with arbitrary user-defined reward objectives remains challenging. While recent advancements in reinforcement learning (RL) have demonstrated considerable success in fine-tuning diffusion-based generative models (Rafailov et al., 2023), their application to continuous flow-based models remains underexplored. The primary challenges lie in lacking computationally efficient divergence measurement and tractable methods to avoid overoptimization (Black et al., 2024) and policy collapse in fine-tuning continuous-time flow matching models. Particularly, traditional policy gradient methods (Black et al., 2024) struggle in this continuous domain due to the high computation cost of computing the exact ODE likelihood for continuous normalizing flows (CNFs), which requires solving intricate transport dynamics and tracking probability flows with model divergence calculation across time (See App. B.2.4).

---

*Corresponding authors.

†Work done as research interns at University of Illinois Urbana-Champaign.

Furthermore, existing diffusion-based RLHF methods, such as DPO (Rafailov et al., 2023) and DDPO (Black et al., 2024), face significant limitations when applied to CNFs. DPO relies on pairwise reward comparisons and filtered datasets, making it unsuitable for optimizing arbitrary reward functions. DDPO adopts a PPO-style policy optimization that depends on frequent likelihood calculations, which are computationally intractable for CNFs. Moreover, methods for diffusion often rely on tractable ELBO likelihood approximation or MDP formulation, which does not apply to CNFs. To tackle these challenges, we propose a novel RL fine-tuning framework that bypasses the need for explicit likelihood estimation by focusing on *reward-weighted objectives* tailored specifically for continuous flow-based models, which we call Reward-Weighted Conditional Flow Matching (RW-CFM). Our approach significantly reduces computational overhead while maintaining optimal performance.

In the meanwhile, many existing reward-weighted RL fine-tuning methods require train on offline datasets manually collected by humans (Huguet et al., 2024) or derived from pre-trained models (Lee et al., 2023). While this approach ensures stable learning, it restricts the model's ability to explore optimally in an online setting, inducing what is known as the *online-offline gap* (Tang et al., 2024) (See Fig. 1 and Figs. 2). This limitation reduces the model's flexibility and can lead to sub-optimal convergence. In

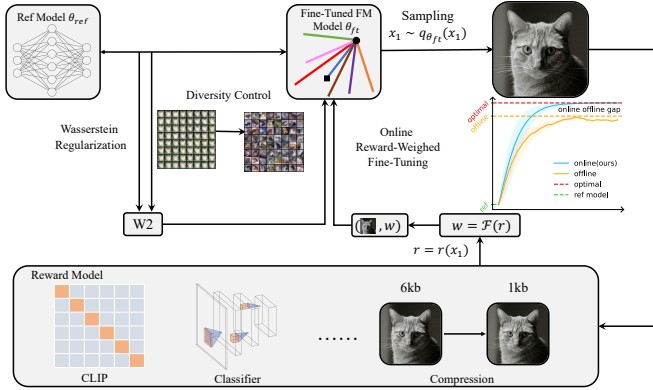

Figure 1: A General Architecture of Our Method.

contrast, online RL enables the generative model to continuously update its training distribution, allowing for greater adaptability and diverse exploration within the reward space. Despite the success in traditional RL domains, the convergence behavior of online RL in fine-tuning generative models, particularly continuous flow-based models, has not yet been widely studied. To leverage the advantages of online RL for generative models, we propose an online reward-weighted fine-tuning method called Online Reward-Weighted Conditional Flow Matching (ORW-CFM). In this paper, we theoretically demonstrate that in online reward-weighted settings, the continuous updates of the training data distribution of the fine-tuned generative model based on reward feedback can cause the learned policy to collapse into a delta distribution (i.e., becoming extremely exploitative; see Lemma 1). Similar phenomena have been experimentally observed in mode collapse of GAN (Goodfellow et al., 2014) and over-optimization in diffusion models (Black et al., 2024). Nevertheless, this issue has not yet been fully addressed or theoretically demonstrated in previous works (Black et al., 2024).

To address these challenges and avoid policy collapse, we introduce a divergence regularization approach to bound the distance between the fine-tuned model and the pre-trained reference model into our online RL method. This regularization ensures that the policy updates remain within a controllable and stable range, preventing collapse while still allowing the model to explore and improve. One of the major challenges in fine-tuning CNFs lies in finding a computationally tractable distance measure between different flow matching models, especially considering the KL divergence used in classic RL methods is inefficient in CNFs. To overcome this, we instead propose to use the Wasserstein-2 (W2) distance (Arjovsky et al., 2017) to measure the distance between the fine-tuned model and the reference model. In this paper, we derive a tractable upper bound of W2 distance in flow matching (See Theorem 3) and incorporate it into our online RL method, inducing a reward-distance trade-off in fine-tuning flow matching models. We demonstrate both theoretically and empirically that W2 distance regularization allows for a controllable and stable fine-tuning process, enabling our agent to balance the reward objective and generative diversity.

In summary, for the first time, we propose an online RL method to fine-tune continuous flow-based generative models towards arbitrary reward objectives with sufficient theoretical guarantees and detailed analysis of the convergence behavior in both online and offline settings. Our approach can leverage classifier-guided reward models, compression rate, and text-image similarity score with CLIP models (Radford et al., 2021b), eliminating the need for time-consuming likelihood calculations that involve the integration of the model divergence across the entire continuous trajectory. Our novel

framework significantly reduces computational costs and enhances the feasibility of using RL to fine-tune continuous flow-based generative models. The main contributions of our paper are:

- **A Novel RL-Based Fine-Tuning Framework for CNFs**: We propose the Online Reward-Weighted Conditional Flow Matching (ORW-CFM) method, a novel RL approach for fine-tuning continuous flow-based generative models. Our method enables optimization with user-defined reward functions without the need for likelihood calculation (Black et al., 2024), filtered dataset (Rafailov et al., 2023) or differentiable reward (Clark et al., 2024), expanding the applicability of flow-based models in various downstream generative tasks.

- **Tractable Wasserstein Regularization**: To prevent policy collapse and maintain the diversity of the fine-tuned models, we introduce Wasserstein-2 (W2) distance regularization into our online method, and derive a tractable upper bound of it in flow matching. This regularization effectively balances exploration and exploitation by bounding the distance between the fine-tuned model and the pre-trained reference model, inducing a theoretically and empirically controllable convergent behavior of fine-tuning flow matching models.

- **Theoretical Analysis and RL Connections:** We provide comprehensive theoretical analyses of our method, including convergence properties and the induced data distributions. We establish connections between our approach and traditional RL algorithms with KL regularization, offering new insights into fine-tuning flow matching generative models.

- **Empirical Validation:** Through extensive experiments on tasks such as target image generation, image compression, and text-image alignment, we empirically demonstrate the effectiveness of our method. Our approach achieves optimal policy convergence, allowing controllable trade-offs between reward maximization and generative diversity.

## 2 RELATED WORKS

**Conditional Flow Matching** Conditional Flow Matching (CFM) (Lipman et al., 2023; Tong et al., 2024a) and related methods have been pivotal in transforming simple data distributions into complex, task-specific ones. Lipman et al. (2023) and Tong et al. (2024a) focus on training flow-based models by conditioning the generative flow on known distributions, allowing for more precise control over the generative process. However, while these models achieve great success in generation tasks, how to fine-tune FM models to fit arbitrary user-defined objectives has not yet been widely studied.

**Fine-tuning From Human Feedback** Recent works typically leverage DPO (Rafailov et al., 2023) or policy gradient RL methods (Black et al., 2024; Fan et al., 2023) to achieve RLHF (Ouyang et al., 2022a). However, since the likelihood of the generative policy cannot be easily calculated in continuous-time flow models, none of these approaches can be easily adapted to fine-tune flow matching models. Additionally, DPO (Rafailov et al., 2023) relies heavily on a filtered dataset, which does not work with a general reward model, while DPOK (Fan et al., 2023) relies on KL divergence, which is computationally inefficient and lacks effective evidence lower bound (ELBO) in FM.

**Fine-Tuning Generative Models with RL** RL has been increasingly adopted to fine-tune generative models, allowing them to adapt to specific downstream tasks by optimizing for user-defined reward objectives. Black et al. (2024); Fan et al. (2023); Lee et al. (2023) used RL to fine-tune diffusion models to align generated data with task-specific rewards. Lee et al. (2023) proposed an offline reward-weighted method, while Black et al. (2024) proposed an online version without divergence regularization, which either failed to converge into optimal policy or collapsed into an extremely greedy policy without diversity (Black et al., 2024). Other reward-weighted methods like ReFT, ReST (Huguet et al., 2024; Gülçehre et al., 2023) are almost limited to the offline settings with given datasets and lack theoretical analysis of the convergent behavior. Importantly, how to adopt online RL methods to fine-tune FM models has not yet been widely studied in both theory and practice.

## 3 PRELIMINARIES

### 3.1 FLOW MATCHING MODELS

Flow Matching (FM) is a promising technique for training generative models by transforming a simple initial distribution $p(x_0)$ like Gaussian noises, into a complex target distribution $p(x_1)$, which represents real-world data (Esser et al., 2024). This transformation works continuously over time, governed by a time-dependent vector field $u_t(x)$ and the flow ordinary differential equation (ODE): $\frac{dx}{dt} = u_t(x)$. The goal of FM is to learn a vector field that transports samples from the base distribution to the target distribution by minimizing the difference between the learned vector field and the true dynamics of the probability flow. This process can be formalized by the Flow Matching loss: $\mathcal{L}_{\text{FM}}(\theta) = \mathbb{E}_{t\sim\mathcal{U}(0,1),x\sim p_t(x)}[\|v_\theta(t,x) - u_t(x)\|^2]$. However, the original FM objective is generally intractable because the time-varying distribution $p_t(x)$ and the true vector field $u_t(x)$ are often intractable. To address this, Conditional Flow Matching (CFM) has been proposed (Lipman et al., 2023), which simplifies the task by conditioning the flow on target samples to derive a tractable objective. The CFM loss is defined as follows:

$$\mathcal{L}_{\text{CFM}}(\theta) = \mathbb{E}_{t\sim\mathcal{U}(0,1),x_1\sim q(x_1),x\sim p_t(x|x_1)}\left[\|v_\theta(t,x) - u_t(x\mid x_1)\|\right]^2 \tag{1}$$

wherein $u_t(x\mid x_1)$ becomes tractable by defining explicit conditional probability paths from $x_0$ to $x_1$, such as OT-paths (Tong et al., 2024a) or linear interpolation paths (Lipman et al., 2023). In this paper, we adopt OT-paths for all experiments based on TorchCFM (Tong et al., 2024b) and Diffusers (von Platen et al., 2022).

### 3.2 REWARD WEIGHTED REGRESSION

Reward Weighted Regression (RWR) (Peters & Schaal, 2007; Black et al., 2024) is an approach used in RL for policy optimization. In this framework, actions that yield higher rewards are assigned greater importance, and the policy is updated by re-weighting actions based on the rewards they produce. The key idea behind RWR is to adjust the probability of selecting actions proportionally to their associated rewards, encouraging the agent to focus on high-reward actions.

Specifically, the policy $\pi(a\mid s)$ is updated by re-weighting the actions via $\pi_{\text{new}}(a\mid s) \propto \pi_{\text{old}}(a\mid s) \cdot \exp(\tau * r(s,a))$, where $r(s,a)$ is the reward for taking action $a$ in state $s$, and $\tau$ is a temperature parameter that controls the trade-off between exploration and exploitation. Higher rewards lead to higher probabilities for the corresponding actions, allowing the agent to improve its policy over time by favoring actions that maximize cumulative rewards: $\pi^* = \arg\max_\pi \mathbb{E}_\pi[\sum_{t=0}^T \gamma^t r(s_t, a_t)]$.

## 4 METHODS

### 4.1 PROBLEM STATEMENT

In this paper, we aim to fine-tune continuous flow-based models using a reward-weighted RL method (Peters & Schaal, 2007; Peng et al., 2019) to optimize a user-defined reward function $r(x_1)$. Specifically, our objective is to find the optimal generative policy model, denoted as $\pi^*$, which maximizes the expected reward throughout data generation:

$$\pi^* = \arg\max_\pi J(\pi) = \arg\max_\pi \mathbb{E}_{x_1\sim\pi(x_1)}[r(x_1)] \tag{2}$$

where the policy $\pi$ refers to the parameterized flow matching data generative model $p^\theta(x_1)$, determining how data points (e.g., images) are generated, which is learned solely from scalar rewards, without ground-truth labels or filtered dataset. In fine-tuning tasks, we may also consider constrained optimization, where the distance between the pre-trained model and fine-tuned model will be constrained (Ouyang et al., 2022a). Considering the computational inefficiency of KL in continuous flow models, we instead propose a Wasserstein-2 distance to achieve that.

### 4.2 FINE-TUNING FLOW MATCHING MODELS WITH OFFLINE REWARD-WEIGHTING

To find the optimal data generation policy in equation 2, in this section, we extend the standard CFM framework by introducing a reward-weighting mechanism to focus the model's learning on more

important or highly rewarded regions of the data space, which thus improve its performance towards user-defined reward objectives $r(x_1)$.

Let $\mathcal{X} \subseteq \mathbb{R}^d$ be the data space, and let $q(x_1)$ be the probability density function (pdf) of the random variable $x_1 \in \mathcal{X}$. We consider a continuous-time flow over $t \in [0, 1]$, where $t$ is sampled from the uniform distribution $\mathcal{U}[0, 1]$. The conditional pdf of $x$ at time $t$, given $x_1$, is denoted by $p_t(x \mid x_1)$. The true conditional vector field is $u_t(x \mid x_1)$, and the model's learned vector field is $v_t(x; \theta)$, where $\theta$ represents the model parameters. In CFM (Tong et al., 2024a), the goal is to learn $v_t(x; \theta)$ such that it closely approximates $u_t(x \mid x_1)$ with equal importance via equation 1. Based on that, we introduce a weighting function $w(x_1) \propto r(x_1) : \mathcal{X} \rightarrow [0, \infty)$ into CFM loss, inducing a reward-weighted CFM method (RW-CFM) as equation 3. Then, we can have the Theorem 1 (Proof in App. C.2):

**Theorem 1.** *Let $w : \mathcal{X} \rightarrow [0, \infty)$ be a measurable weighting function such that $0 < Z = \int_{\mathcal{X}} w(x_1) q(x_1) dx_1 < \infty$ and $w(x_1) \propto r(x_1)$. Define the reward weighted CFM loss function:*

$$\mathcal{L}_{\mathrm{RW-CFM}}(\theta) = \mathbb{E}_{t \sim \mathcal{U}[0,1], x_1 \sim q(x_1), x \sim p_t(x|x_1)} \left[ w(x_1) \|v_t(x; \theta) - u_t(x \mid x_1)\|^2 \right] \tag{3}$$

*where $w(x_1)$ is a weighting function, $q(x_1)$ is the original data distribution (e.g., data distribution induced by well-learned pre-trained models), $p_t(x \mid x_1)$ is the conditional distribution at time $t$, $v_t(x; \theta)$ is the model's vector field, and $u_t(x \mid x_1)$ is the true vector field conditioned on $x_1$. Then, training the model using this RW-CFM loss leads the model to learn a new data distribution:*

$$p^{new}(x_1) = \frac{w(x_1) q(x_1)}{Z} \tag{4}$$

*where $Z = \int w(x_1) q(x_1) dx_1$ is the normalization constant.*

Based on Theorem 1, it is obvious that the model trained with the RW-CFM loss will focus on high-reward regions compared to the original data distribution $q(x_1)$. However, since the training data is sampled from a static dataset $q(x_1)$, which is equivalent to an offline RL setting (Black et al., 2024), the model may not find the optimal data distribution $p^*$ (i.e., optimal policy $\pi^*$) that maximizes the expected reward (i.e., online and offline gap, see Fig. 2). To address this, we propose using the fine-tuned flow matching models as the data sampling policy (i.e., $q(x_1) = p^\theta(x_1)$), enabling an online generation of training data and thereby inducing an online RW-CFM method.

### 4.3 Fine-Tuning Flow Matching Models with Online Reward-Weighting

As discussed, to mitigate the online-offline gap and achieve better performance, we can instead adopt an online RL algorithm, where the fine-tuned FM models are used as a **data generation/sampling policy** to continuously generate new training data and explore high-value regions, inducing what we call the Online Reward-Weighted Conditional Flow Matching (ORW-CFM) method:

$$\mathcal{L}_{\mathrm{ORW\text{-}CFM}}(\theta) = \mathbb{E}_{t \sim \mathcal{U}[0,1], x_1 \sim p^\theta(x_1), x \sim p_t(x|x_1)} \left[ w(x_1) \|v_t(x; \theta) - u_t(x \mid x_1)\|^2 \right], \tag{5}$$

In the online setting, the model can adaptively sample from the current learned data distribution, and fine-tune itself based on the newly generated samples, inducing an online policy iteration that promotes more efficient exploration of the high-reward areas in the data space. In general, the data distribution learned by the ORW-CFM method follows Theorem 2 (Proof in App. C.3):

**Theorem 2** (Online Reward Weighted CFM). *Let $q(x_1)$ be the initial data distribution, and $w(x_1)$ be a non-negative weighting function integrable with respect to $q(x_1)$, and proportional to user-defined reward $r(x_1)$. Assume that at each epoch $n$, the model $v_\theta$ perfectly learns the distribution implied by the ORW-CFM loss in equation 5. Then, the learned data distribution after $N$ epochs is given by:*

$$q_\theta^N(x_1) = \frac{w(x_1)^N q(x_1)}{Z_N} \tag{6}$$

*where $Z_N = \int_{\mathcal{X}} w(x_1)^N q(x_1) dx_1$ is the normalization constant ensuring $q_\theta^N(x_1)$ is a valid probability distribution.*

Based on Theorem 2, we can easily derive the limiting behavior of ORW-CFM method (Proof in App. C.3) as the training converges (i.e., the learned generative policy collapses into an extremely exploitative/greedy policy without diversity as the case where $\alpha = 0$ in Fig. 3):

**Lemma 1** (Limiting Case). *We now consider the limiting cases where $N \to \infty$:*

*If $w(x_1) \propto r(x_1) > 0$ for all $x_1 \in \mathcal{X}$ and attains its maximum rewards at $x_1^*$, then as $N \to \infty$, the learned data distribution $q_\theta^N(x_1)$ converges to a Dirac delta function centered at $x_1^*$ :*

$$\lim_{N \to \infty} q_\theta^N(x_1) = \delta(x_1 - x_1^*) \tag{7}$$

From Lemma 1, in the online setting, as training progresses, the ORW-CFM method may converge to the optimal policy that yields maximum reward without any diversity, potentially leading to a policy collapse problem (See Figs. 3, 6 and 11 where $\alpha = 0$):

**Corollary 1** (Policy Collapse/Overoptimization). *Overemphasizing high-reward regions can lead to a lack of diversity in generated samples, similar to mode collapse in GANs (Goodfellow et al., 2014), overoptimization in fine-tuning diffusion models (Black et al., 2024), or policy collapse in RL (Haarnoja et al., 2018; Sutton & Barto, 1998).*

Based on Lemma 1 and Theorem 2, assuming an exponential weighting function $w(x_1) = \exp(\tau * r(x_1))$ case, we can have $w(x_1)^N = \exp(N * \tau * r(x_1))$, where $\tau$ can control the convergence/collapse speed of learned policy, and $\tau \to \infty$ also induces a policy collapse problem (See Fig. 2). To mitigate the policy collapse problem, we introduce Wasserstein-2 regularization into our method, which can bound the distance between the current learned model and the pre-trained model. This helps maintain diversity and prevents the model from collapsing into an extremely greedy policy, inducing an exploration-exploitation trade-off in fine-tuning flow matching models.

## 4.4 WASSERSTEIN-2 REGULARIZATION IN FLOW MATCHING

To address the policy collapse problem in the ORW-CFM method, we introduce Wasserstein-2 (W2) regularization, which helps maintain diversity by bounding the distance between the current learned model and a pre-trained reference model. This regularization induces an exploration-exploitation trade-off, allowing the model to avoid focusing solely on high-reward regions at the expense of diversity. In general, we can write the Wasserstein-2 Distance as follows (Arjovsky et al., 2017):

**Definition 4.1** (Wasserstein-2 Distance). *Given two probability measures $\mu$ and $\nu$ on $\mathbb{R}^n$, the squared Wasserstein-2 distance between $\mu$ and $\nu$ is defined as:*

$$W_2^2(\mu, \nu) = \inf_{\gamma \in \Pi(\mu,\nu)} \int_{\mathbb{R}^n \times \mathbb{R}^n} \|x - y\|^2 d\gamma(x, y) \tag{8}$$

*where $\Pi(\mu, \nu)$ denotes the set of all couplings of $\mu$ and $\nu$; that is, all joint distributions $\gamma$ on $\mathbb{R}^n \times \mathbb{R}^n$ with marginals $\mu$ and $\nu$.*

However, it is intractable to directly introduce W2 distance into our loss function. Therefore, we instead derive and introduce its upper bound as our distance regularizer (Proof in App. C.4):

**Theorem 3** (W2 Bound for Flow Matching). *We consider two flow matching models parameterized by $\theta_1$ and $\theta_2$, inducing time-evolving data distributions $p_t^{\theta_1}(x)$ and $p_t^{\theta_2}(x)$, respectively. The models define vector fields $v^{\theta_1}(t, x)$ and $v^{\theta_2}(t, x)$ that transport an initial distribution $p_0(x)$ to final distributions $p_1^{\theta_1}(x)$ and $p_1^{\theta_2}(x)$ at time $t = 1$. Assume that $v^{\theta_2}(t, x)$ is Lipschitz continuous in $x$ with Lipschitz constant L. Then, the squared Wasserstein-2 distance between the distributions $p_1^{\theta_1}$ and $p_1^{\theta_2}$ induced by the flow matching models at time $t = 1$ satisfies:*

$$W_2^2\left(p_1^{\theta_1}, p_1^{\theta_2}\right) \leq e^{2L} \int_0^1 \mathbb{E}_{x \sim p_s^{\theta_1}}\left[\left\|v^{\theta_1}(s, x) - v^{\theta_2}(s, x)\right\|^2\right] ds \tag{9}$$

Since the integral can be approximated by Monte Carlo sampling, this bound allows us to constrain the discrepancy between the learned and reference models, effectively preventing the model from collapsing into a greedy policy. By controlling the Wasserstein-2 distance, the model maintains a balance between exploring new areas of the data space and exploiting known high-reward regions in a constrained neighborhood from the pre-trained model, thus preserving diversity.

### 4.5 FINE-TUNING FLOW MATCHING MODELS WITH W2 REGULARIZATION

The offline version of the reward-weighted method can be considered as the first step of the online version (Black et al., 2024). For simplicity, we only derive the ORW-CFM method with W2 distance regularization (i.e., ORW-CFM-W2 method) in this section, and left the offline version (i.e., RW-CFM with W2 distance regularization, namely RW-CFM-W2 method) in App. C.5 for ease of reading.

To further improve the stability of the ORW-CFM method and prevent policy collapse, we incorporate W2 regularization into our method, which has not been widely studied (Black et al., 2024). This regularization introduces a balance between exploration and exploitation by penalizing divergence from the reference model, thus maintaining diversity in the learned model. The W2 distance bound can be directly incorporated into the ORW-CFM loss, inducing an ORW-CFM-W2 method:

$$\mathcal{L}_{\text{ORW-CFM-W2}} = \mathbb{E}_{t \sim U(0,1), x_1 \sim q(x_1; \theta_{\text{ft}}), x \sim p_t(x|x_1)} [w(x_1) \left\| v^{\theta_{\text{ft}}}(x, t) - u_t(x \mid x_1) \right\|^2 \tag{10}$$
$$+ \alpha * \left\| v^{\theta_{\text{ft}}}(x, t) - v^{\theta_{\text{ref}}}(x, t) \right\|^2],$$

wherein $\alpha$ is a trade-off coefficient, $v^{\theta_{\text{ft}}}(x, t) = v_t(x; \theta_{\text{ft}})$ and $v^{\theta_{\text{ref}}}(x, t)$ is the reference model. Then, the learned data distribution after n epochs follows the Theorem 4 (Proof in App. C.6):

**Theorem 4.** *Under the online RL setup with ORW-CFM-W2 loss, the data distribution after n epochs evolves according to the following rules:*

$$q_\theta^n(x_1) \propto \left[ w(x_1) q_\theta^{n-1}(x_1) \exp\left(-\beta D^{n-1}(x_1)\right) \right], \tag{11}$$

*where, $D^{n-1}(x_1) = \mathbb{E}_{t, x \sim p_t(x|x_1)} \left[ \left\| v^{\theta^{n-1}}(t, x) - v^{\theta_{ref}}(t, x) \right\|^2 \right]$, $\beta = \gamma\alpha, \gamma > 0$ is a scaling constant, $\theta^{n-1}$ denotes the fine-tuned model parameters after epoch $n-1$.*

The W2 regularization term prevents the model from solely collapsing into high-reward regions and ensures diversity in the learned policy (See cases where $\alpha > 0$ in Figs.3, 4 and 11), also inducing the exploration-exploitation trade-off.

In practice, we often use an exponential function or Boltzmann distribution to formulate the weighting function as $w(x_1) = \exp(\tau * r(x_1))$, where $\tau$ is an entropy coefficient that controls the entropy of the induced policy (Haarnoja et al., 2018). Thus, we can derive the induced data distribution for the exponential case under the ORW-CFM-W2 loss (Proof in App. C.7):

**Theorem 5.** *Given $w(x_1) = \exp(\tau * r(x_1))$, the induced data distribution after $N$ epochs of training under ORW-CFM-W2 Loss as equation 10 is:*

$$q_\theta^N(x_1) \propto \exp\left(\tau N r(x_1) - \beta \sum_{n=1}^{N} D^{n-1}(x_1)\right) q(x_1) \tag{12}$$

*where, $D^{n-1}(x_1) = \mathbb{E}_{t, x \sim p_t(x|x_1)} \left[ \left\| v^{\theta^{n-1}}(t, x) - v^{\theta_{ref}}(t, x) \right\|^2 \right]$, $\beta = \gamma\alpha, \gamma > 0$ is a scaling constant, $\theta^{n-1}$ denotes the model parameters after epoch $n-1$.*

Based on this, we can derive the learning behavior of ORW-CFM-W2 under two limiting cases:

**Case 1** (Dominant Regularization Term: $\alpha \to \infty$ or $\tau = 0$). *If $D^{n-1}(x_1)$ grows rapidly with N or $\alpha$ is large, since $\beta \propto \alpha$, the regularization term dominates, namely $\beta \sum_{n=1}^{N} D^{n-1}(x_1) \gg \tau * r(x_1)$. The induced distribution is heavily penalized for deviations from the reference model, potentially leading to a distribution similar to the initial distribution $q(x_1)$ (See $\tau = 0$ in Fig. 2).*

**Case 2** (Dominant Reward Term: $\alpha = 0$ or $\tau \to \infty$). *If $D^{n-1}(x_1)$ grows slowly with N or $\alpha$ is small, since $\beta \propto \alpha$, the reward term dominates the regularization term, namely $\tau * r(x_1) \gg \beta \sum_{n=1}^{N} D^{n-1}(x_1)$. The induced distribution $q_\theta^N(x_1)$ concentrates on the $x_1$ maximizing $r(x_1)$, similar to the case without considering $D^{n-1}(x_1)$ as Lemma 1 (See $\alpha = 0$ in Fig. 3).*

Though the Boltzmann distribution is commonly used for the weighting function to handle the normalization constant $Z$, the derivation is similar to the exponential form. For completeness, we provide the derivation of the Boltzmann form weighting function in App. C.8 for ease of reading.

Interestingly, we can interpret our ORW-CFM-W2 method from the perspective of policy iteration in RL (Sutton & Barto, 1998), which can connect our method to KL-constrained policy optimization (Ouyang et al., 2022a; Peng et al., 2019). See App. C.9 for more detailed discussion.

## 5 EXPERIMENTS

We empirically evaluate our ORW-CFM-W2 method in equation 10 by fine-tuning small-scale FM models with a U-net architecture from TorchCFM (Tong et al., 2024a) and large-scale FM models like Stable Diffusion 3 (SD3) (Esser et al., 2024) from diffusers (von Platen et al., 2022) for reproducibility. In all experiments, we use the exponential weighting function described in Theorem 5. We also explore the impact of each component of our method and conducted detailed analyses of convergent behavior in various hyperparameter settings. More experimental details can be found in App. D and E with our Pseudocode in App. H. In general, we focus on the following questions:

- **Optimal Policy Convergence:** Can our online RL methods converge to an optimal generative data distribution that maximizes various user-defined reward objectives as we derived in Theorem 2 and Lemma 1 while the offline baseline method converges to a sub-optimal?

- **Reward-Diversity Trade-off:** Does W2 regularization effectively prevent policy collapse by maintaining a balance between reward maximization and diversity as Theorem 5?

- **Stable and Controllable Policy Optimization:** Can our method achieve a stable and controllable fine-tuning process of FM models in different settings, and control the convergent behavior of learned policy by adjusting entropy coefficient $\tau$ and W2 coefficient $\alpha$?

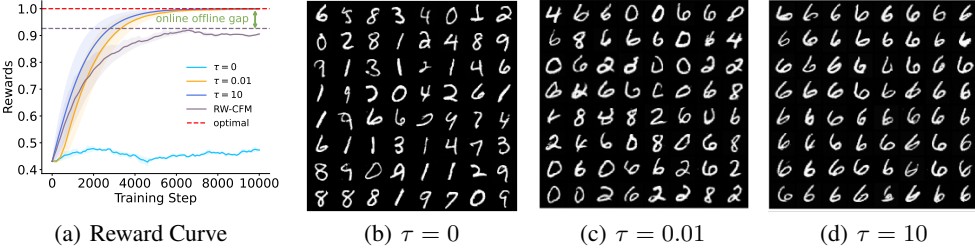

(a) Reward Curve       (b) $\tau = 0$       (c) $\tau = 0.01$       (d) $\tau = 10$

Figure 2: Learning curve and generated images in target image generation task with different $\tau$ while $\alpha = 0$. As $\tau$ increases, the convergent policy becomes increasingly greedy, and the diversity decreases. $\tau = 0$ remains the similar distribution as pre-trained model (See Theorem 5).

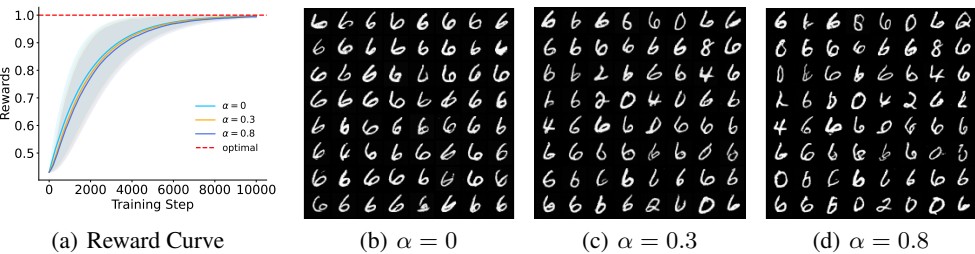

(a) Reward Curve       (b) $\alpha = 0$       (c) $\alpha = 0.3$       (d) $\alpha = 0.8$

Figure 3: Learning curve and generated images in target image generation task with different $\alpha$ while $\tau = 10$. From $\alpha = 0$ to $\alpha = 0.8$, the diversity of the convergent generative distribution increases without sacrificing too much performance. $\alpha = 0$ collapse to a Delta Distribution as Lemma 1.

### 5.1 TARGET IMAGE GENERATION

At first, we evaluate our ORW-CFM method, with and without W2 regularization, by fine-tuning a pre-trained model of MNIST (LeCun et al., 1998) to generate only even numbers using reward signals. The reward function is defined as $r(x_1) = p_{\text{even}}(x_1) - p_{\text{odd}}(x_1)$, calculated by a pre-trained binary classifier. An optimal expected return of 1 indicates the model exclusively generates even numbers.

Figs. 2 and 3 demonstrate that our method quickly reaches near-optimal performance across different settings of the entropy coefficient $\tau$ and regularization coefficient $\alpha$. From Fig. 2, increasing $\tau$ leads to a more greedy, reward-maximizing policy at the cost of diversity, consistent with the theoretical limits in Theorem 5 and Lemma 1, while $\tau = 0$ remains a similar performance as the pre-trained model. Besides, from Fig. 3, W2 regularization controlled by $\alpha$ remains diverse by controlling how far the

fine-tuned model diverges from the pre-trained model, enabling a balance between exploration and exploitation. In general, ORW-CFM, especially with W2 regularization, achieves a stable fine-tuning process via maximizing the rewards in a constrained neighborhood of the pre-trained model, inducing a flexible balance between reward and diversity, avoiding extremely exploitative policies.

## 5.2 IMAGE COMPRESSION

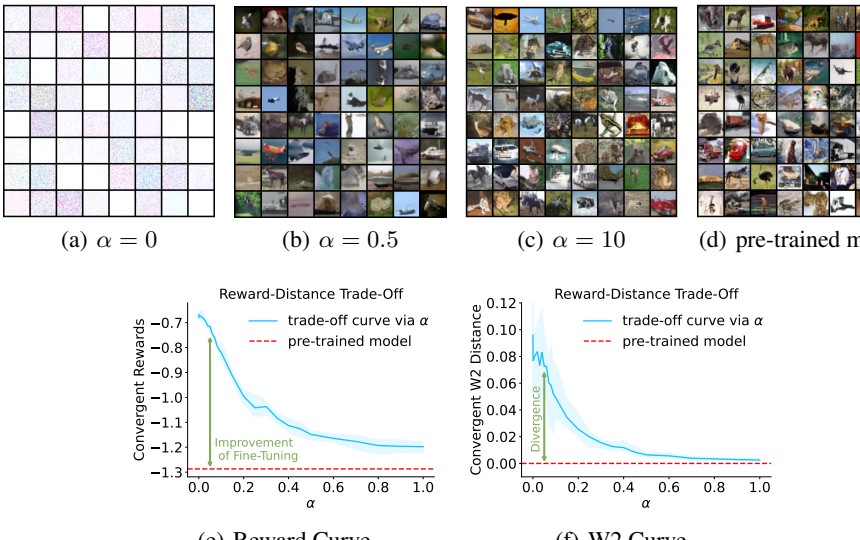

(a) $\alpha = 0$      (b) $\alpha = 0.5$      (c) $\alpha = 10$      (d) pre-trained model

(e) Reward Curve          (f) W2 Curve

Figure 4: Reward-Diversity Trade-off and W2 Distance Control via $\alpha$ in Image Compression Task (Black et al., 2024) of CIFAR-10 (Krizhevsky & Hinton, 2009) with $\tau = 1$. W2 distance is estimated by its upper bound (see App. C.4). Each point in (e) and (f) corresponds to one group of experiments with $\alpha$ varying from 0 to 1. As $\alpha$ increases, the final reward decays while distance between the fine-tuned model and the reference model become closer, inducing a controllable fine-tuning process.

The effect of W2 regularization and exploration-exploitation trade-off is particularly evident in the image compression task using the model pre-trained in CIFAR-10, as shown in Figs. 4. In Fig. 4, we see that the introduction of W2 regularization prevents the policy from collapsing into an extremely greedy solution. As $\alpha$ increases, the model explores optimal solutions within a constrained neighborhood of the pre-trained model, preserving diversity while still optimizing the reward.

The reward-distance trade-off curve in Fig. 4 further highlights this balance. Varying $\alpha$ provides an explicit trade-off between maximizing the reward and minimizing the divergence from the pre-trained model. As $\alpha$ decreases, the policy becomes more exploitative, potentially collapsing into a Delta distribution, as described in Lemma 1. Conversely, increasing $\alpha$ encourages broader exploration of the data space, ensuring a more diverse generative policy. Therefore, proper W2 regularization effectively balances exploration and exploitation, avoiding policy collapse.

## 5.3 TEXT-IMAGE ALIGNMENT

**Aligning Large-Scale Flow Matching Models** To further validate our method's effectiveness, we evaluate our ORW-CFM-W2 method on fine-tuning Stable Diffusion 3 (SD3) (Esser et al., 2024). Besides, we adopt LoRA (Hu et al., 2022) for efficient fine-tuning. As shown in Figure 5, we compare our method against RAFT (Dong et al., 2023) and ReFT (Huguet et al., 2024) on spatial relationship prompts used in DPOK (Fan et al., 2023). Our approach demonstrates better positional relationship control while maintaining image quality and semantic coherence. On challenging prompts like "a banana on the left of an apple" and "a cat on the left of a dog", our method consistently generates images with correct object placement and natural appearances, outperforming previous approaches in terms of both alignment accuracy and generation quality. Results on these complex compositional prompts further demonstrate our approach's capability in handling multiple semantic constraints while maintaining generation diversity. See App. A for more details and additional results.

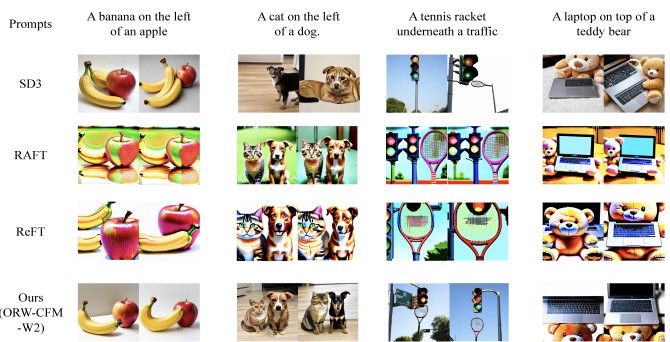

Figure 5: General Comparison of Different Fine-tuning Methods on SD3 via CLIP Rewards.

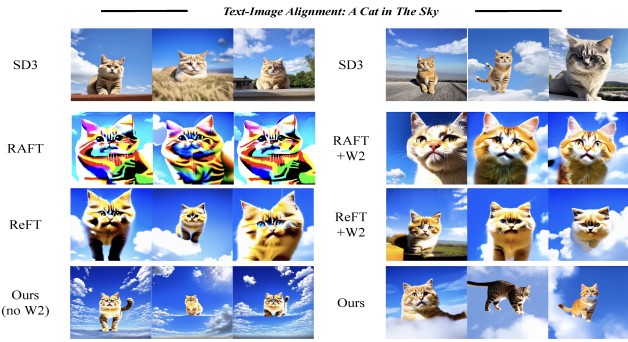

Figure 6: Ablation Studies on Text-Image Alignment Tasks of SD3. All use CLIP rewards.

**Ablation Results** The ablation studies in Figure 6 demonstrate several key findings that align with our theoretical predictions. Our ORW-CFM method, even without W2 regularization ($\alpha = 0$), achieves better semantic alignment than other online methods like RAFT and ReFT, though all methods without W2 regularization exhibit clear signs of policy collapse, generating nearly identical images despite achieving high rewards - empirically validating our theoretical prediction in Lemma 1. The introduction of W2 regularization (RAFT+W2, ReFT+W2, and our ORW-CFM-W2 method, $\alpha = 1$) successfully prevents this collapse while maintaining performance, validating our tractable W2 distance bound in Theorem 3. Our method achieves the best balance between reward optimization and diversity preservation, with generated images showing consistent semantic understanding of "cat in sky" while maintaining natural variations in compositions. See App. A.2 for more details.

## 6 CONCLUSION

In this paper, we present the first theoretically-grounded online RL framework for fine-tuning flow matching models - Online Reward-Weighted Conditional Flow Matching with Wasserstein-2 Regularization (ORW-CFM-W2) without requiring calculations of ELBO, exact likelihood or KL divergence, which are intractable in ODE-based flow matching models. Our work provides the first theoretical analysis of convergent behavior in flow matching online fine-tuning, proving that unregularized methods inevitably converge to a Delta distribution (Lemma 1), inducing policy collapse that maximizes rewards at the cost of diversity. This theoretical discovery motivates our second major contribution - the derivation of a tractable Wasserstein-2 regularization that can be directly computed from vector fields, providing an efficient way to prevent collapse while maintaining diversity. Extensive experiments across target image generation, image compression, and text-image alignment demonstrate that our method achieves optimal convergence while maintaining stable learning, with ablation studies validating both the online reward-weighting mechanism for optimal convergence and W2 regularization for diversity preservation. See App. I for more discussions.

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

## A  TEXT-IMAGE ALIGNMENT EXPERIMENTS OF STABLE DIFFUSION 3

In this section, to further empirically demonstrate our performance on fine-tuning large-scale generative models like the stable diffusion series (Esser et al., 2024), we conduct extensive experiments on fine-tuning Stable Diffusion 3 (SD3) (Esser et al., 2024), which adopts a flow matching architecture (Lipman et al., 2023; Esser et al., 2024). For reproducibility, we built our code on the open-sourced diffuser codebase (von Platen et al., 2022), and adopted LoRA (Hu et al., 2022) and fp16 precision to reduce GPU memory requirements. Unless otherwise specified, we use $\alpha = 1, \tau = 0.1$ for all experiments in this section.

To verify the effectiveness of our method, we primarily use the raw CLIP score (i.e., logit) (Radford et al., 2021a) as our rewards for text-image alignment for SD3 fine-tuning. We also demonstrate our method's adaptability by extending it to various other text-image alignment reward models, including HPS-V2 (Wu et al., 2023), Pick Score (Kirstain et al., 2023), and Alpha Clip (Sun et al., 2024). Notably, our method learns entirely from self-generated data (Dong et al., 2023; Shumailov et al., 2024) without requiring manually collected (Ouyang et al., 2022a) or filtered datasets (Rafailov et al., 2023), while effectively preventing policy collapse through our theoretically-derived W2 regularization (See App. C.4).

### A.1  TEXT-IMAGE ALIGNMENT WITH SPATIAL UNDERSTANDING

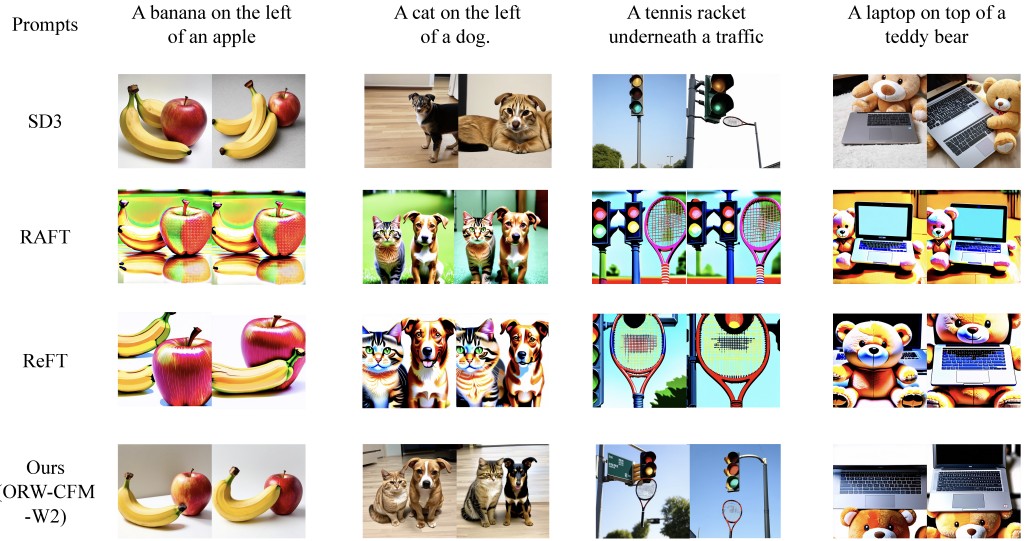

Figure 7: General comparison of positional relationship understanding between different fine-tuning methods on SD3 (Esser et al., 2024). We compare our proposed ORW-CFM-W2 method against RAFT (Dong et al., 2023), ReFT (online variants, training on self-generated data) (Huguet et al., 2024), and the original SD3 model (Esser et al., 2024) across various positional relationship prompts from DrawBench (Saharia et al., 2022) used in DPOK (Fan et al., 2023). Each column shows generations for a specific prompt testing spatial understanding (e.g., "left of", "on top of", "underneath"). The results demonstrate that our method achieves better positional relationship control while maintaining image quality and semantic coherence. This suggests that our Wasserstein-2 regularized online reward-weighted approach effectively fine-tunes flow-based models to better understand and generate complex spatial relationships between objects. Model generations shown are sampled with the same random seeds across methods for fair comparison.

In the first experiments, we aim to improve the model's understanding of spatial relationships between objects, a challenging task that requires both semantic understanding and precise positional control.

To demonstrate the effectiveness of our method, we use the raw CLIP score (i.e., logits) (Radford et al., 2021a; Xu et al., 2023; Dong et al., 2023) directly as our text-image alignment reward model,

eliminating the need for probability conversion or complex reward shaping (Kapturowski et al., 2019). This straightforward approach further demonstrates our method's ability to optimize directly from raw similarity scores, making it more practical for real-world applications (Dong et al., 2023; Radford et al., 2021a).

Figure 7 (i.e., Same as Figure 5, but with more detailed discussion for ease of reading.) compares our method against several baselines: RAFT (Dong et al., 2023), ReFT (online variants, training on self-generated data) (Huguet et al., 2024), and the original SD3 model. The results demonstrate several key advantages of our approach:

1. Spatial Semantic Coherence/Alignment: Our method shows superior understanding of spatial relationships ("left of", "on top of", "underneath") while maintaining high image quality. This validates our theoretical analysis that ORW-CFM-W2 can effectively optimize arbitrary reward functions without compromising generation quality.

2. Controlled Optimization: As shown in Figure 3 and Figure 7, online RL methods trained on self-generated data (Dong et al., 2023; Huguet et al., 2024; Black et al., 2024; Shumailov et al., 2024) normally face great challenges of policy collapse, where models converge to limited, homogeneous outputs that maximize rewards but lack diversity - a phenomenon theoretically characterized in Lemma 1. To address this challenge, we introduce W2 regularization, which effectively prevents over-optimization and policy collapse (Lemma 1). Our theoretical analysis in Theorem 5 demonstrates that this regularization enables a controlled trade-off between reward maximization and output diversity (See Figure 4). The experimental results validate this prediction, showing consistent spatial relationships while maintaining natural variations in object appearances, styles, and compositions.

3. Easy-to-Use and Stable Fine-tuning Method: Our method achieves these improvements without requiring filtered datasets (Rafailov et al., 2023), likelihood calculations (Black et al., 2024) or differentiable rewards (Domingo-Enrich et al., 2024), demonstrating the practical advantages of our reward-weighted flow matching framework. This aligns with our theoretical framework that bypasses the computational challenges of likelihood estimation in continuous flow models.

In general, these results further validate our theoretical contributions in a practical, large-scale setting. The successful fine-tuning of SD3 demonstrates that our method can effectively balance between reward maximization and diversity preservation, as predicted by our reward-distance trade-off analysis (Theorem 4 and 5). Furthermore, the stable learning behavior supports our theoretical analysis of the convergence properties under W2 regularization. These experiments also highlight the practical significance of our two key innovations: the online reward-weighting mechanism and the W2 regularization. The online approach allows the model to continuously explore and improve spatial understanding, while the W2 regularization prevents the collapse into simplistic or degenerate solutions - a common challenge in fine-tuning large generative models (Dong et al., 2023; Black et al., 2024; Shumailov et al., 2024).

### A.2 ABLATION STUDIES AND POLICY COLLAPSE ANALYSIS

In the second part of SD3 fine-tuning experiments, to empirically validate our theoretical analysis of policy collapse in online RL fine-tuning as Lemma 1 and demonstrate the effectiveness of our W2 regularization via ablation, we further fine-tune SD3 models to align out-of-distribution (OOD) prompt "a cat in the sky". This setup provides an ideal test case for examining both the policy collapse phenomenon and our method's ability to maintain diversity while optimizing rewards in different online methods. The ablation results have been concluded in Figure 8.

**Effectiveness of Online Reward Weighting (ORW-CFM)** The ablation studies in Figure 8 demonstrate that our ORW-CFM method, even without W2 regularization ($\alpha = 0$), achieves better semantic alignment than other online methods like RAFT (Dong et al., 2023) and ReFT (Huguet et al., 2024). This superior performance validates our theoretical framework for online reward-weighted fine-tuning of flow matching models (See Lemma Optimal Convergent Behavior in Lemma 1). The method successfully optimizes for the desired semantic content, achieving higher-quality generations that better match the prompt "cat in sky" compared to baseline approaches.

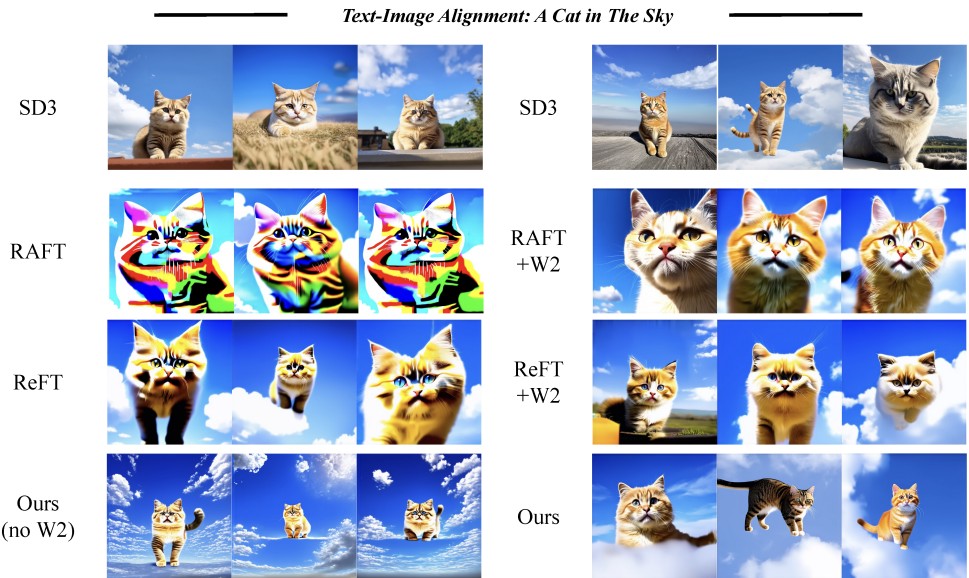

Figure 8: Comparison of different fine-tuning methods on fine-tuning SD3 for the prompt "a cat in the sky" (reward is also the CLIP score Radford et al. (2021a); Xu et al. (2023); Dong et al. (2023)). The results demonstrate that policy collapse frequently occurs during online RL fine-tuning (Black et al., 2024; Dong et al., 2023; Shumailov et al., 2024). The baseline SD3 Esser et al. (2024) shows limited spatial/semantic understanding for unusual prompts (i.e., hard to see in our normal life). While RAFT (Dong et al., 2023) and ReFT (Huguet et al., 2024) without W2 regularization achieve high rewards (e.g., successful cat placement), they exhibit clear signs of policy collapse, generating nearly identical images with limited diversity. Our method without W2 regularization (Ours no W2) similarly suffers from policy collapse (also see Figure 2), aligning with our theoretical prediction of the convergent behavior in Lemma 1. In contrast, methods incorporating W2 regularization (RAFT+W2, ReFT+W2, and Ours) maintain generation diversity while successfully positioning cats in the sky. Our approach is particularly good at balancing reward optimization with diversity preservation, demonstrating the effectiveness of our theoretically-derived W2 regularization bound (Theorem 3) in preventing policy collapse while maintaining high-quality generations. The results empirically validate our theoretical analysis of the reward-diversity trade-off in Theorem 5.

**Policy Collapse Phenomenon**    However, in Figure 8, all methods without W2 regularization (RAFT, ReFT, and Ours without W2) exhibit clear signs of policy collapse (Shumailov et al., 2024; Black et al., 2024), generating nearly identical images despite achieving high rewards. This empirically validates our theoretical prediction in Lemma 1 that unregularized online reward-weighted methods naturally converge to a delta distribution at the maximum reward point, sacrificing diversity for reward maximization.

**Effectiveness of W2 Regularization** Based on Figure 8 and Table 1, the introduction of W2 regularization (RAFT+W2, ReFT+W2, and our ORW-CFM-W2 method, $\alpha = 1$) demonstrates the effectiveness of our theoretically-derived tractable W2 distance bound from Theorem 3. This regularization successfully prevents policy collapse while maintaining performance across all methods, enabling controlled divergence from the reference model. The consistent improvement in output diversity across different baseline methods validates our theoretical framework for controlling model behavior through W2 regularization in flow matching architectures.

**Optimal Balance Through ORW-CFM-W2**    In Figure 8, our ORW-CFM-W2 method achieves the best balance between reward optimization and diversity preservation - generated images show consistent high-level semantic understanding of "cat in sky" while maintaining natural variations in cat appearances, sky conditions, and overall compositions. While both RAFT+W2 and ReFT+W2 show diversity improvements from W2 regularization, their outputs still exhibit less variation in style

and composition compared to ORW-CFM-W2. This improved performance aligns with our theoretical analysis in Theorem 5 regarding the reward-diversity trade-off in the exponential weighting case. The results also validate our optimization framework's ability to achieve stable convergence without collapsing into extreme exploitation.

**Theoretical and Practical Validation**  These experimental results in Figure 8 provide strong empirical validation of both key components of our approach. First, they demonstrate the ORW-CFM method's ability to achieve optimal policy convergence, though with potential collapse if unregularized (as predicted by our theoretical analysis of online reward-weighted learning). Second, they show how W2 regularization effectively prevents this collapse while preserving output quality and diversity - a capability derived from our tractable W2 distance bound for flow matching. The comprehensive improvement across different baseline methods (Dong et al., 2023; Huguet et al., 2024) illustrates how our theoretical framework enables stable fine-tuning that maximizes rewards while maintaining the rich generative capabilities of flow-based models (Lipman et al., 2023; Esser et al., 2024).

Table 1: Performance and Diversity comparison of different fine-tuning methods on text-image alignment using SD3. 'CLIP Score' measures alignment with the text prompt using the CLIP model's similarity score (higher is better). 'Diversity Score' quantifies the diversity of generated images by measuring the mean pairwise distance between CLIP embeddings of generated samples (higher indicates more diverse outputs). We use Euclidean distance to calculate the distance between any two embeddings when calculating mean pairwise distance. Our method achieves the highest CLIP score while maintaining strong diversity comparable to the base SD3 model. The addition of W2 regularization ('+W2' variants) helps preserve generation diversity across all methods, with our full approach (Ours) achieving the best balance between alignment (35.46) and diversity (4.21) compared to baseline methods RAFT and ReFT. All scores are calculated by averaging across 64 samples.

| Method | CLIP Score ↑ | Diversity Score ↑ |
|---|---|---|
| *Effectiveness of ORW-CFM without W2* | | |
| SD3 (Baseline) | 28.69 | **4.78** |
| ORW-CFM (Ours w/o W2) | **33.63** | 3.47 |
| RAFT | 29.30 | 2.05 |
| ReFT | 29.32 | 3.26 |
| *Effectiveness of W2 regularization* | | |
| ORW-CFM-W2 (Ours) | **35.46** | **4.21** |
| RAFT + W2 | 30.88 | 2.81 |
| ReFT + W2 | 32.03 | 3.63 |

## A.3 ADAPTABILITY ACROSS DIFFERENT REWARD MODELS

In addition to the three reward functions in Section 5, to demonstrate the broad applicability and reward-agnostic nature/property of our methods, we further evaluate our performance using three different text-image alignment reward models: HPS-V2 (Wu et al., 2023), Pick Score (Kirstain et al., 2023), and Alpha Clip (Sun et al., 2024). Using the challenging Spatial prompt "a train on top of a surfboard," we examine how our method adapts to different reward signals/models while maintaining generation quality and semantics coherence.

In general, the results shown in Figure 9 demonstrate several key strengths of our approach:

**Reward Adaptability:** Our method shows consistent performance improvement across all three reward models, successfully positioning trains on surfboards while maintaining realistic scene composition. This empirically validates our theoretical framework's ability to optimize arbitrary reward functions without requiring reward-specific modifications (Dong et al., 2023) or filtered datasets (Rafailov et al., 2023). The high-quality generations across different reward models demonstrate that our W2 regularization (Theorem 3) provides effective stabilization regardless of the underlying reward mechanism (It is worth mentioning that appropriately increasing the $\alpha$ value can be used to adjust the distance from the reference model, which is also demonstrated in Figure 4).

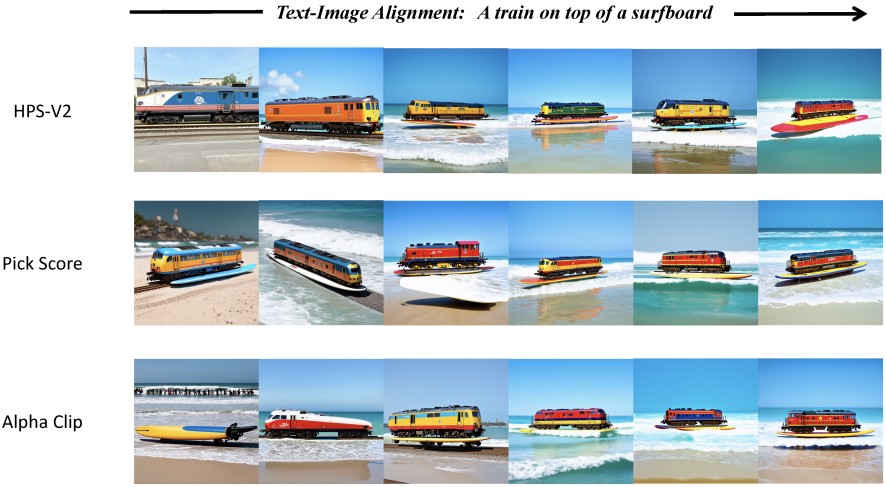

Figure 9: Comparison of fine-tuning methods for the prompt "a train on top of a surfboard", demonstrating our method's adaptability across different reward models (HPS-V2 Wu et al. (2023), Pick Score Kirstain et al. (2023), and Alpha Clip scores Sun et al. (2024)). The results validate our theoretical analysis that ORW-CFM-W2 can effectively optimize diverse reward functions while maintaining generation quality through W2 regularization. 1) Top row: Using HPS-V2 reward shows our method successfully aligns locomotive placement while preserving realistic wave interactions. 2) Middle row: Pick Score reward further demonstrates robust spatial understanding across varying train models and surfboard configurations. 3) Bottom row: Alpha Clip reward showcases consistent performance even with different text-image alignment rewards, validating the reward-agnostic nature/property of our approach. This comprehensive evaluation across multiple reward models empirically confirms our theoretical prediction that the W2 regularization (Theorem 3) enables stable fine-tuning regardless of the underlying reward mechanism. The consistent high-quality generations across all three reward models demonstrate the practical advantages of our reward-weighted flow matching framework, which achieves strong performance without requiring filtered datasets (Rafailov et al., 2023), likelihood calculations (Black et al., 2024), reward-specific optimizations (Dong et al., 2023; Huguet et al., 2024), or differentiable rewards (Domingo-Enrich et al., 2024). These results particularly highlight how our method's reward-diversity trade-off (Theorem 5) generalizes effectively across different reward models while preventing policy collapse (Lemma 1).

**Spatial Understanding:** Across all reward models, our method demonstrates robust spatial understanding:

1. **HPS-V2:** Achieves precise locomotive placement while capturing realistic wave interactions.
2. **Pick Score:** Maintains consistent spatial relationships across varying train models and surfboard configurations.
3. **Alpha Clip:** Shows stable performance even with a different similarity metric framework.

**Diversity Preservation:** Consistent with our theoretical analysis in Theorem 5, the generated images maintain natural variations in train appearances, surfboard designs, and ocean conditions while adhering to the spatial constraints. This demonstrates how our W2 regularization effectively prevents policy collapse (Lemma 1) regardless of the reward models used.

**Stable Fine-tuning:** The consistent quality/alignment improvement across reward models validates our theoretical contribution regarding the convergence behavior of ORW-CFM-W2 (See App. C.6). As predicted by our analysis of the exponential weighting case (Theorem 5), the method achieves

stable learning without collapsing into extreme exploitation, even when faced with different reward scales and architectures. It is also worth mentioning that the reward model is very important for the success of the fine-tuning task (Dong et al., 2023; Black et al., 2024; Rafailov et al., 2023; Xu et al., 2023), so how to select the best reward model still needs further exploration.

All in all, the successful adaptation across multiple reward models showcases how our online reward-weighting mechanism and W2 regularization work together to enable robust, stable fine-tuning without requiring reward-specific optimizations. This generality makes our method particularly valuable for real-world applications where different reward models might be needed for different tasks or domains (Black et al., 2024; Shumailov et al., 2024).

Furthermore, these results address a key challenge in flow matching fine-tuning: **the ability to optimize diverse reward objectives without compromising generation quality or diversity while training on self-generated data**. Our method's consistent performance improvement across reward models demonstrates that the reward-distance trade-off formulation (See Theorem 5) provides a easy-to-use and theoretically-grounded approach to balancing task-specific optimization with output diversity, regardless of how the reward is computed.

### A.4 COMPOSITIONAL AND COMPLEX SEMANTIC UNDERSTANDING

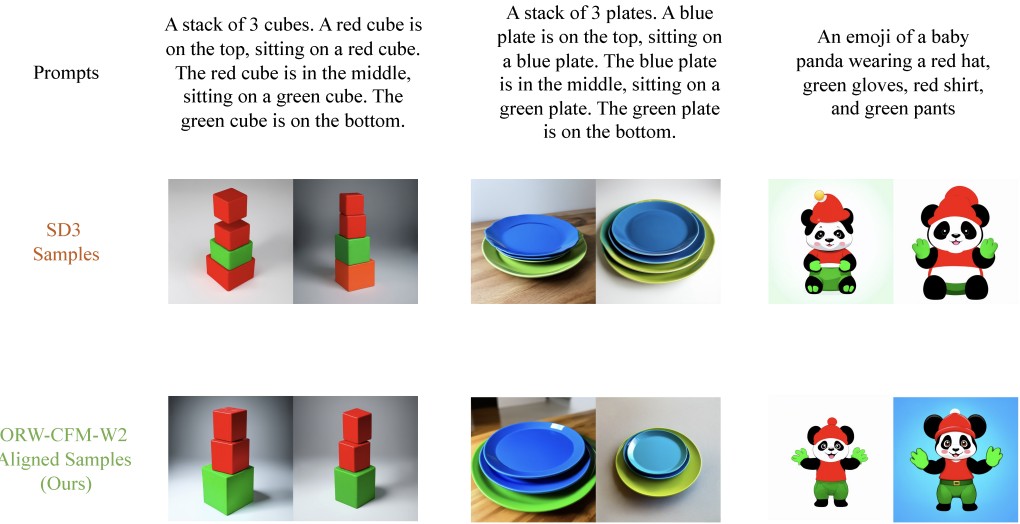

Figure 10: Additional evaluation of ORW-CFM-W2's semantic understanding and compositional generation capabilities on complex structured prompts from DrawBench (Saharia et al., 2022) used in DPOK (Fan et al., 2023). Each column demonstrates our method's ability to handle multi-faceted requirements including spatial relationships, color specifications (red, green), and detailed object attributes. The figure compares baseline SD3 (Esser et al., 2024) samples with our ORW-CFM-W2 results across three challenging compositional prompts: stacked colored cubes, stacked colored plates, and a customized panda emoji. Our method shows consistently better semantic alignment while maintaining generation diversity, validating our theoretical framework's ability to prevent mode collapse (Shumailov et al., 2024) through W2 regularization.

In the fourth experiments of fine-tuning SD3 models (Esser et al., 2024), to further validate our method's capability in handling complex semantic relationships and demonstrate the practical benefits of our theoretical framework, we evaluate ORW-CFM-W2 on a set of challenging compositional prompts that test multiple aspects of semantic understanding. These experiments not only demonstrate our method's effectiveness on large-scale flow matching models like SD3, but also empirically validate our theoretical predictions about controlled optimization and diversity preservation. We use CLIP

score as our text-image alignment reward model (Radford et al., 2021a; Xu et al., 2023; Dong et al., 2023).

**Compositional Attribute Control:** As shown in Figure 10, our method successfully handles prompts with multiple interleaved requirements spanning **spatial relationships, color specifications, and object attributes**. In the cube-stacking example, the model demonstrates precise control over both spatial ("on top", "in the middle", "on the bottom") and color attributes (two red cubes on a green cube), while maintaining physically plausible stacking arrangements. This validates that our ORW-CFM-W2 can effectively achieve fine-grained fine-tuning without falling into degenerate solutions.

**Spatial Color Relationships:** The stacked plates example particularly highlights our method's ability to jointly optimize multiple constraints - positional relationships (top, middle, bottom) and color attributes (blue plates on top/middle, green plate on bottom). The consistent high-quality results validate our theoretical prediction that our methods enables stable fine-tuning even when dealing with complex, spatially-dependent color constraints.

**Multi-attribute Character Specification:** The panda emoji example demonstrates fine-grained control over multiple clothing attributes (red hat, green gloves, red shirt, green pants) while maintaining the core semantic concept and cartoon-style consistency. This success in handling detailed attribute specifications without semantic drift empirically supports our analysis of the reward-diversity trade-off (Theorem 4), showing how our method balances attribute-level precision with overall coherence.

In general, across all test cases, our method achieves notably better semantic alignment compared to the baseline SD3 samples while maintaining image quality and stylistic diversity. This improvement is particularly evident in cases requiring precise attribute combinations or spatial relationships, where baseline samples often exhibit attribute mixing or spatial inconsistencies. The results demonstrate that our theoretically-grounded approach effectively addresses the policy collapse problem (Lemma 1) through W2 regularization, enabling precise semantic control without sacrificing generation diversity. These results offer strong empirical validation of our theoretical framework. The successful handling of complex compositional prompts demonstrates that our online reward-weighting mechanism and W2 regularization work together to enable robust, stable fine-tuning of large-scale flow matching models. This generality makes our method particularly valuable for real-world applications requiring precise semantic control and attribute specification while maintaining natural variations in the generated outputs. All in all, the consistent performance across diverse prompt types demonstrates our ability to optimize multiple semantic objectives without compromising generation quality or diversity. Our method's success in handling complex compositional prompts (Saharia et al., 2022; Fan et al., 2023) while maintaining coherent generations provides strong empirical support for our theoretical contributions while demonstrating the practical utility of our approach in real-world applications.

# B  BACKGROUND

## B.1  REINFORCEMENT LEARNING

### B.1.1  RL FORMULATION

In RL (Sutton & Barto, 1998), an agent interacts with an environment by taking actions $a \in \mathcal{A}$ in states $s \in \mathcal{S}$, aiming to maximize cumulative rewards over time. The agent's behavior is governed by a policy $\pi(a \mid s)$, which represents the probability of taking action $a$ given state $s$. The main goal of RL is to optimize the policy to maximize the expected return:

$$J(\pi) = \mathbb{E}_{\pi} \left[ \sum_{t=0}^{T} \gamma^t r\left(s_t, a_t\right) \right] \tag{13}$$

where $\gamma \in [0, 1]$ is the discount factor that prioritizes early rewards, $T$ is the time horizon (which could be infinite), $r\left(s_t, a_t\right)$ is the reward at time step $t$.

The core problem in RL is how to learn the optimal policy $\pi^*(a \mid s)$ that maximizes the expected return $J(\pi)$. A special case of RL, where there is no states but only actions for each agent, is known as the multi-armed bandit problem (Sutton & Barto, 1998).

### B.1.2  REWARD-WEIGHTED REGRESSION

In Reward-Weighted Regression (RWR), the policy update is done by re-weighting the probability of actions based on the rewards they produce. The updated policy is given by:

$$\pi_{\text{new}}\left(a \mid s\right) \propto \pi_{\text{old}}\left(a \mid s\right) \cdot \exp(\tau * r(s, a)) \tag{14}$$

where $\tau$ is a temperature parameter controlling the trade-off between exploration and exploitation.

### B.1.3  ADVANTAGE-WEIGHTED REGRESSION

Advantage-Weighted Regression (AWR) refines RWR by weighting actions based on their advantage, which measures how much better an action is compared to the expected value of other actions:

$$\pi_{\text{new}}\left(a \mid s\right) \propto \pi_{\text{old}}\left(a \mid s\right) \cdot \exp(\tau A(s, a)) \tag{15}$$

where $A(s, a) = r(s, a) - V(s)$ is the advantage function.

In practice, we can also call the data-collection policy, the behavior policy $\mu$ (Sutton & Barto, 1998; Schulman et al., 2017; Peng et al., 2019), and the optimal target policy $\pi^{\star}$.

Then, the update rules of AWR becomes:

$$\pi_{\text{new}}\left(a \mid s\right) \propto \mu(a \mid s) \cdot \exp(\tau A(s, a)) \tag{16}$$

And the optimal policy is therefore given by (Peng et al., 2019),

$$\pi^*(\mathbf{a} \mid \mathbf{s}) = \frac{1}{Z(\mathbf{s})} \mu(\mathbf{a} \mid \mathbf{s}) \exp\left(\tau * A^{\mu}\left(\mathbf{s}_t, \mathbf{a}_t\right)\right) \tag{17}$$

where $Z(s)$ is the partition function that normalizes the conditional action distribution.

## B.2  FLOW MATCHING

### B.2.1  FLOW MATCHING FORMULATION

Flow Matching (Lipman et al., 2023) is a methodology for training continuous normalizing flows, which are generative models that transform a simple base distribution into a complex target distribution through a continuous, invertible mapping. The core idea of FM is to define a continuous flow that

gradually transports samples from an initial distribution $p_0(x)$ (usually a simple distribution like a standard Gaussian) to a target distribution $p_1(x)$ (the data distribution we aim to model).

This transformation is governed by an ordinary differential equation:

$$\frac{dx}{dt} = u_t(x) \tag{18}$$

where $x \in \mathbb{R}^d$ is a data point in $d$-dimensional space, $t \in [0, 1]$ is the time parameter, and $u_t(x)$ is a time-dependent vector field that dictates the dynamics of the flow at each point in time and space.

The solution to this ODE defines a flow $\phi_t(x)$ that maps a point $x$ at time $t$ to another point at time $t + \delta t$. The transformation over the entire time interval $[0, 1]$ effectively transports the entire distribution $p_0(x)$ to $p_1(x)$.

The evolution of the probability density $p_t(x)$ over time is described by the continuity equation:

$$\frac{\partial p_t(x)}{\partial t} + \nabla \cdot (p_t(x)u_t(x)) = 0 \tag{19}$$

where $\nabla.$ denotes the divergence operator. The goal in FM is to learn the vector field $u_t(x)$ such that the flow induced by $u_t(x)$ transforms $p_0(x)$ into $p_1(x)$ as $t$ progresses from 0 to 1.

### B.2.2 CONDITIONAL FLOW MATCHING

Conditional Flow Matching (Lipman et al., 2023) is an extension of FM that introduces conditioning to simplify the learning process. Instead of modeling the entire distribution transformation directly, CFM focuses on modeling the conditional dynamics of the flow given a target sample $x_1$ from the data distribution.

In CFM, we consider pairs $(x_0, x_1)$, where $x_0 \sim p_0(x)$ and $x_1 \sim p_1(x)$. The conditional flow transports $x_0$ to $x_1$ over time $t \in [0, 1]$ using a vector field $u_t(x \mid x_1)$ that depends on $x_1$:

$$\frac{dx}{dt} = u_t(x \mid x_1) \tag{20}$$

The objective in CFM is to learn a parametric model $v_\theta(t, x)$ that approximates $u_t(x \mid x_1)$. The loss function used to train the model is:

$$\mathcal{L}_{\text{CFM}}(\theta) = \mathbb{E}_{t, x_1, x} \left[ \|v_\theta(t, x) - u_t(x \mid x_1)\|^2 \right] \tag{21}$$

where the expectation is taken over $t \sim \mathcal{U}(0, 1)$, $x_1 \sim p_1(x)$, and $x \sim p_t(x \mid x_1)$, the conditional distribution at time $t$ given $x_1$.

By conditioning on $x_1$, CFM reduces the complexity of the learning problem, as the model only needs to learn how to transform $x_0$ to $x_1$ for given pairs, rather than modeling the entire joint distribution.

### B.2.3 OT-CFM

Our baseline code is built upon torchcfm (Tong et al., 2024a), where we use OT-CFM loss to learn flow matching models (i.e., OT-paths). Optimal Transport theory provides a framework for finding the most efficient way to transport one probability distribution to another (Tong et al., 2024a). OT-CFM integrates optimal transport into the conditional flow matching framework to define vector fields based on optimal transport maps.

In OT-CFM (Tong et al., 2024a), the vector field $u_t(x \mid x_1)$ is derived from the optimal transport plan that minimizes the cost of transporting $p_0(x)$ to $p_1(x)$. The cost is typically defined in terms of the expected squared Euclidean distance between transported points.

By using optimal transport, OT-CFM ensures that the flow from $x_0$ to $x_1$ follows the most efficient path, which can improve the convergence and quality of the generative model. The OT-CFM loss function remains similar to the CFM loss but incorporates the optimal transport-based vector field:

$$\mathcal{L}_{\text{OT}-\text{CFM}}(\theta) = \mathbb{E}_{t,x_1,x}\left[\left\|v_\theta(t,x) - u_t^{\text{OT}}(x \mid x_1)\right\|^2\right] \tag{22}$$

where $u_t^{\text{OT}}(x \mid x_1)$ represents the optimal transport vector field.

### B.2.4 LIKELIHOOD CALCULATION

As a special case of CNF, flow matching models share the benefit of exact likelihood calculation. Specifically, the learned vector field induces a probability path induced by the pushforward:

$$p_t := (\psi_t)_* p_0 \tag{23}$$

where $p_0$ is the initial noise distribution and $\psi_t(x) := x_t$ is the flow governed by the learned vector field and the flow ODE as $\frac{dx_t}{dt} = v_t(x_t)$. The continuity equation in equation 19 allows us to calculate such change-of-probability term with the estimation of the model divergence as (Lipman et al., 2023):

$$\log p_1(x_1) = \log p_0(x_0) + \int_1^0 \text{div}(v_t)(x_t)dt \tag{24}$$

The trajectory is generated reverse through time from $t = 1$ (arbitrary data) to $t = 0$ via the learned vector field. Direct calculation of the model divergence $\text{div}(v_t)$ is prohibitively expensive. Instead, Hutchinson's trace estimator is used to efficiently approximate the model divergence as (Hutchinson, 1989):

$$\text{div}(v_t)(x) = \mathbb{E}_{\varepsilon \sim \mathcal{N}(0,I)}[\varepsilon^\top \nabla v_t(x)\varepsilon] \tag{25}$$

In contrast to diffusion models that rely on variational bounds that can be calculated in closed form as the summation of KL divergence (Ho et al., 2020), **the likelihood for flow-based models requires expensive simulation and the calculation of vector-Jacobian product to estimate the model divergence**. Therefore, the likelihood-based RL formulation for diffusion models in Black et al. (2024); Schulman et al. (2017); Rafailov et al. (2023); Fan et al. (2023) is non-trivial and computationally expensive for flow models.

## C  PROOFS

### C.1  INDUCED DATA DISTRIBUTION

In practice, in generative model, we are normally very interested in the induced data distribution $p_{new}(x_1)$ after learning under a design loss function $\mathcal{L}$. In order to facilitate the subsequent derivation, in this section, we first derive a general Theorem of Induced Data Distribution.

**Theorem 6** (Induced Data Distribution). *Given the loss function:*

$$\mathcal{L}_{total}(\theta) = \mathbb{E}_{x_1 \sim q(x_1)} \left[ \tilde{L}(x_1) \right] \tag{26}$$

*where $\tilde{L}(x_1)$ is any per-sample loss function, the induced data distribution over $x_1$ is:*

$$p_{new}(x_1) \propto q(x_1) \exp\left(-\gamma \tilde{L}(x_1)\right) \tag{27}$$

*with $\gamma > 0$ being a positive constant.*

*Proof of Theorem 6, from RL Perspective.*  In fact, most previous RLHF (Ziegler et al., 2019; Stiennon et al., 2020; Ouyang et al., 2022b) and RL works (Peng et al., 2019) have found that assume a reward model $r(x)$ that captures human preferences, with the goal of modifying the base generative model $q(x)$ such that it generates the following tilted distribution:

$$p_{new}(x_1) = p^*(x_1) \propto q(x_1) \exp(\tau * r(x_1)) \tag{28}$$

just set the reward model as the negative of per-sample loss $\tilde{L}(x_1)$, namely:

$$r(x_1) = -\tilde{L}(x_1) \tag{29}$$

and set $\tau = \gamma$, which is a constant, and we can obtain:

$$p_{new}(x_1) = p^*(x_1) \propto q(x_1) \exp\left(-\gamma \tilde{L}(x_1)\right) \tag{30}$$

Then, the proof concludes. $\qquad\square$

*Proof of Theorem 6, from Energy Function Perspective.*  In standard EBMs, the probability density over $x_1$ is defined using an energy function $E(x_1)$ :

$$p_{\text{EBM}}(x_1) = \frac{1}{Z} \exp\left(-\beta E(x_1)\right) \tag{31}$$

where $\beta > 0$ is the inverse temperature, and $Z$ is the partition function ensuring normalization:

$$Z = \int \exp\left(-\beta E(x_1)\right) dx_1 \tag{32}$$

This formulation assumes a uniform base measure over $x_1$.

We consider the per-sample loss $\tilde{L}(x_1)$ with the energy function $E(x_1)$, and set $\beta = \gamma$. Therefore:

$$p_{\text{new}}(x_1) = \frac{1}{Z} \exp\left(-\gamma \tilde{L}(x_1)\right) \tag{33}$$

When the data distribution $q(x_1)$ is not uniform, or when we have prior knowledge about $x_1$ like $q(x_1)$, it is appropriate to include $q(x_1)$ in the model (LeCun et al., 2006). This leads to a modified probability density:

$$p_{\text{new}}\left(x_1\right) = \frac{1}{Z} q\left(x_1\right) \exp\left(-\gamma \tilde{L}\left(x_1\right)\right) \tag{34}$$

Here, $q\left(x_1\right)$ acts as a base measure or prior distribution, and $\tilde{L}\left(x_1\right)$ corresponds to the energy function $E\left(x_1\right)$. The inclusion of $q\left(x_1\right)$ is necessary to accurately represent the underlying distribution of the data when it is not uniform.

**Remark.** *The total loss function is:*

$$\mathcal{L}_{total}\left(\theta\right) = \int q\left(x_1\right) \tilde{L}\left(x_1\right) dx_1 \tag{35}$$

*Minimizing $\mathcal{L}_{total}\left(\theta\right)$ with respect to $\theta$ corresponds to adjusting the model parameters to reduce the expected loss over the data distribution $q\left(x_1\right)$.*

*We can interpret this minimization as seeking a distribution $p_{new}\left(x_1\right)$ that places higher probability on samples with lower $\tilde{L}\left(x_1\right)$. This aligns with the principle of maximum likelihood estimation in EBMs, where we aim to maximize the likelihood of observed data under the model.*

Then, the proof concludes. $\square$

*Proof of Theorem 6, from Bayesian Inference Perspective.* In Bayesian inference, the posterior distribution $p_{\text{new}}\left(x_1 \mid \mathcal{D}\right)$ over a latent variable $x_1$ given data $\mathcal{D}$ is proportional to the product of the prior $q\left(x_1\right)$ and the likelihood $p\left(\mathcal{D} \mid x_1\right)$ :

$$p_{\text{new}}\left(x_1 \mid \mathcal{D}\right) = \frac{1}{Z} q\left(x_1\right) p\left(\mathcal{D} \mid x_1\right) \tag{36}$$

where $Z$ is the normalization constant:

$$Z = \int q\left(x_1\right) p\left(\mathcal{D} \mid x_1\right) dx_1 \tag{37}$$

In statistical modeling and machine learning, the loss function $\tilde{L}\left(x_1\right)$ quantifies the discrepancy between the model's predictions and the observed data. It is common to relate the loss function to the likelihood function through the negative log-likelihood:

$$\tilde{L}\left(x_1\right) = -\frac{1}{\gamma} \log p\left(\mathcal{D} \mid x_1\right) \tag{38}$$

Rearranging this equation gives:

$$p\left(\mathcal{D} \mid x_1\right) = \exp\left(-\gamma \tilde{L}\left(x_1\right)\right) \tag{39}$$

The posterior distribution becomes:

$$p_{\text{new}}\left(x_1\right) = \frac{1}{Z} q\left(x_1\right) \exp\left(-\gamma \tilde{L}\left(x_1\right)\right) \tag{40}$$

where:

$$Z = \int q\left(x_1\right) \exp\left(-\gamma \tilde{L}\left(x_1\right)\right) dx_1 \tag{41}$$

which ensures that $p_{\text{new}}\left(x_1\right)$ integrates to one.

Then, the proof concludes. $\square$

**Corollary 2** (Limiting Behavior of Well Learned Model). *As $\tilde{L}(x_1) \to 0$, the exponential term approaches 1 :*

$$\lim_{\tilde{L}(x_1) \to 0} \exp\left(-\gamma \tilde{L}(x_1)\right) = 1 \tag{42}$$

*Thus, $p_{new}(x_1) \propto q(x_1)$, meaning the induced distribution is the same as the data sampling distribution $q(x_1)$.*

**Corollary 3** (Weighted Prior Distribution). *Considering a weighed prior sampling distribution $p_{eff}(x_1) = w(x_1)q(x_1)$, given the loss function:*

$$\mathcal{L}_{total}(\theta) = \mathbb{E}_{x_1 \sim p_{eff}(x_1)}\left[\tilde{L}(x_1)\right] \tag{43}$$

*where $\tilde{L}(x_1)$ is any per-sample loss function, the induced data distribution over $x_1$ is:*

$$
\begin{aligned}
p_{new}(x_1) &\propto p_{eff}(x_1) \exp\left(-\gamma \tilde{L}(x_1)\right) \\
&\propto w(x_1)q(x_1) \exp\left(-\gamma \tilde{L}(x_1)\right)
\end{aligned}
\tag{44}
$$

*with $\gamma > 0$ being a positive constant.*

*Proof of Corollary 3.* Substituting $q(x_1) = p_{eff}(x_1)$ into the Theorem 6, we can have the induced data distribution:

$$
\begin{aligned}
p_{new}(x_1) &\propto p_{eff}(x_1) \exp\left(-\gamma \tilde{L}(x_1)\right) \\
&\propto w(x_1)q(x_1) \exp\left(-\gamma \tilde{L}(x_1)\right)
\end{aligned}
\tag{45}
$$

then, the proof concludes. $\qquad\square$

**Corollary 4** (Constant Loss). *If $\tilde{L}(x_1) = c$ for all $x_1$, then:*

$$p_{new}(x_1) \propto q(x_1) \exp(-\gamma c) = q(x_1) \times \text{ constant.} \tag{46}$$

*Thus, $p_{new}(x_1)$ is proportional to $q(x_1)$, and the induced distribution remains unchanged.*

## C.2 REWARD WEIGHTED CONDITIONAL FLOW MATCHING

We start from a conditional flow matching setting, where a model parameterized by $\theta$ defines a vector field $v_\theta(t, x)$ that transports samples from a base distribution to a target distribution over time t $\in [0, 1]$. The standard CFM loss function is as follows:

$$\mathcal{L}_{\text{CFM}}(\theta) = \mathbb{E}_{t \sim \mathcal{U}(0,1), q(x_1), p_t(x|x_1)} \|v_\theta(t, x) - u_t(x \mid x_1)\|^2, \tag{47}$$

which aims to align $v_\theta(t, x)$ with the true conditional vector field $u_t(x|x_1)$ by minimizing the expected squared error over the data distribution $q(x_1)$.

In practical applications, it is often desirable to focus the model's learning capacity on specific regions of the data space. This can be achieved by incorporating a weighting function $w(x_1)$ into the loss function, effectively reweighting the data distribution. However, the theoretical implications of this modification on the learned distribution have not been fully studied.

In this part, we rigorously analyze the impact of using a weighted regression loss in CFM training. We prove that incorporating a weighting function into the CFM loss leads the model to learn a new distribution over $x_1$, given by $p^{\text{new}}(x_1) = \frac{w(x_1)q(x_1)}{Z}$, where $q(x_1)$ is the original data distribution and $Z$ is a normalization constant. Our analysis provides theoretical support for the use of weighting functions to control the focus of generative models, with implications for target data generation, RLHF (Ouyang et al., 2022a) and RL fine-tuning (Black et al., 2024; Rafailov et al., 2023).

Let $\mathcal{X} \subseteq \mathbb{R}^d$ denote the data space, and let $q(x_1)$ be the probability density function (pdf) of the random variable $x_1 \in \mathcal{X}$. We consider a continuous-time flow over $t \in [0, 1]$, where $t$ is sampled from the uniform distribution $\mathcal{U}[0, 1]$. The conditional pdf of $x$ at time $t$ given $x_1$ is denoted by $p_t(x \mid x_1)$.

The true conditional vector field is $u_t(x \mid x_1)$, and the model's vector field is $v_t(x; \theta)$, where $\theta$ represents the model parameters. The goal is to learn $v_t(x; \theta)$ such that it approximates $u_t(x \mid x_1)$

We introduce a weighting function $w : \mathcal{X} \to [0, \infty)$, which is measurable and integrable with respect to $q(x_1)$.

Then, we can have the following Theorem (For ease of reading, we rewrite the Theorem 1 to be proved in this section as follows):

**Theorem.** *Let $w : \mathcal{X} \to [0, \infty)$ be a measurable weighting function such that $0 < Z = \int_{\mathcal{X}} w(x_1) q(x_1) dx_1 < \infty$. Define the reward weighted CFM loss function:*

$$\mathcal{L}_{\text{RW-CFM}}(\theta) = \mathbb{E}_{t \sim \mathcal{U}[0,1], x_1 \sim q(x_1), x \sim p_t(x|x_1)} \left[ w(x_1) \|v_t(x; \theta) - u_t(x \mid x_1)\|^2 \right] \tag{48}$$

*where $w(x_1)$ is a weighting function, $q(x_1)$ is the original data distribution, $p_t(x \mid x_1)$ is the conditional distribution at time $t$, $v_t(x; \theta)$ is the model's vector field, and $u_t(x \mid x_1)$ is the true vector field conditioned on $x_1$. Then, training the model using this weighted loss function leads the model to learn a new data distribution:*

$$p^{new}(x_1) = \frac{w(x_1) q(x_1)}{Z} \tag{49}$$

*where $Z = \int w(x_1) q(x_1) dx_1$ is the normalization constant.*

*Proof of Theorem 1.* We first aim to show that incorporating the weighting function $w(x_1)$ into the loss function effectively changes the distribution over $x_1$ from $q(x_1)$ to $p^{\text{new}}(x_1)$.

**Lemma 2.** *The weighting function modifies the data distribution over $x_1$ from $q(x_1)$ to:*

$$p^{new}(x_1) = \frac{w(x_1) q(x_1)}{Z} \tag{50}$$

*Proof of Lemma 2.* The weighted loss function can be explicitly written as:

$$\mathcal{L}_{\text{RW-CFM}}(\theta) = \int_0^1 \int_{\mathcal{X}} w(x_1) q(x_1) \int_{\mathcal{X}} \|v_t(x; \theta) - u_t(x \mid x_1)\|^2 p_t(x \mid x_1) \, dx dx_1 dt \tag{51}$$

Then, we can define the reweighted distribution $p^{\text{new}}(x_1)$ as:

$$p^{\text{new}}(x_1) = \frac{w(x_1) q(x_1)}{Z}, \quad \text{where} \quad Z = \int_{\mathcal{X}} w(x_1) q(x_1) \, dx_1 \tag{52}$$

Since $w(x_1) \geq 0$ and $Z < \infty, p^{\text{new}}(x_1)$ is a valid pdf over $\mathcal{X}$.

Substituting $p^{\text{new}}(x_1)$ into the loss function, we have:

$$\mathcal{L}_{\text{RW-CFM}}(\theta) = Z \int_0^1 \int_{\mathcal{X}} p^{\text{new}}(x_1) \int_{\mathcal{X}} \|v_t(x;\theta) - u_t(x \mid x_1)\|^2 p_t(x \mid x_1) \, dx dx_1 dt \tag{53}$$

This can be expressed as an expectation:

$$\mathcal{L}_{\text{RW-CFM}}(\theta) = Z \mathbb{E}_{t \sim \mathcal{U}[0,1], x_1 \sim p^{\text{new}}(x_1), x \sim p_t(x|x_1)} \left[ \|v_t(x;\theta) - u_t(x \mid x_1)\|^2 \right] \tag{54}$$

The gradient of $\mathcal{L}_{\text{RW-CFM}}(\theta)$ is:

$$\nabla_\theta \mathcal{L}_{\text{RW-CFM}}(\theta) = Z \mathbb{E}_{t \sim \mathcal{U}[0,1], x_1 \sim p^{\text{new}}(x_1), x \sim p_t(x|x_1)} \left[ \nabla_\theta \|v_t(x;\theta) - u_t(x \mid x_1)\|^2 \right] \tag{55}$$

Since $Z$ is constant with respect to $\theta$, it does not affect the optimization process. Therefore, minimizing $\mathcal{L}_{\text{RW-CFM}}(\theta)$ is equivalent to minimizing the expected loss under the distribution $p^{\text{new}}(x_1)$

**Remark.** *According to Corollary 3, set $p_{eff}(x_1) = \frac{w(x_1) q(x_1)}{Z}$, we have:*

$$\begin{aligned} p_{new}(x_1) &\propto p_{eff}(x_1) \exp\left(-\gamma \tilde{L}(x_1)\right) \\ &\propto w(x_1) q(x_1) \exp\left(-\gamma \tilde{L}(x_1)\right) \end{aligned} \tag{56}$$

*When the model fits the data well (i.e., attain convergence), and $\tilde{L}(x_1) \to 0$, then according to Corollary 2, we have:*

$$\begin{aligned} p_{new}(x_1) &\propto w(x_1) q(x_1) \exp\left(-\gamma \tilde{L}(x_1)\right) \\ &\propto w(x_1) q(x_1) \end{aligned} \tag{57}$$

**Remark.** *During optimization, the model minimizes the expected squared difference between the model's vector field and the true vector field, but crucially, this expectation is now taken with respect to the reweighted distribution $p^{new}(x_1)$. As a result, the model prioritizes accurately modeling the flow dynamics in regions of higher weight, effectively learning to generate data according to $p^{new}(x_1)$.*

The proof concludes. □

Based on Lemma 2, it's obvious that Theorem 1 holds. □

Based on equation 55, the reward weighted conditional flow matching loss in equation 48 actually learn a weighted distribution $p^{new}(x_1)$:

$$p^{\text{new}}(x_1) = \frac{w(x_1) q(x_1)}{Z} \tag{58}$$

**Remark** (Normalization Constant $Z$). *$Z$ is a finite normalization constant ensuring $p^{new}(x_1)$ is a valid probability distribution (i.e., integrates to 1). In practice, during training, the normalization constant can be ignored because it does not affect the optimization process since it is constant with respect to $\theta$.*

**Lemma 3** (Finite Normalization Constant $Z$). *The normalization constant $Z$ is finite if $w(x_1)$ is integrable with respect to $q(x_1)$.*

*Proof of Lemma 3.* Since $w(x_1)$ is a weighting function and $q(x_1)$ is a probability density function, the integral:

$$Z = \int_{\mathcal{X}} w(x_1) q(x_1) \, dx_1 \tag{59}$$

is finite if $w(x_1)$ does not introduce singularities and is integrable over $\mathcal{X}$. $\qquad\square$

**Remark** (Data Traversal). *In fact, to optimize the reward weighted CFM loss in equation 48 requires the agent to nearly traverse almost possible samples $x_1$, which may also similar to the exploration problem in RL, namely how to explore the data space $\mathcal{X}$. Therefore, maintaining the ability to continuously explore the data space is important for RL method (Ecoffet et al., 2021; Haarnoja et al., 2018).*

C.3 ONLINE REWARD WEIGHTED CONDITIONAL FLOW MATCHING

In App. C.2 we have discussed offline version of Reward Weighted Conditional Flow Matching, which is learned by optimizing the following loss:

$$\mathcal{L}_{\text{RW}-\text{CFM}}(\theta) = \mathbb{E}_{t\sim\mathcal{U}[0,1],x_1\sim q(x_1),x\sim p_t(x|x_1)}\left[w\left(x_1\right)\left\|v_t(x;\theta) - u_t\left(x\mid x_1\right)\right\|^2\right], \quad (60)$$

wherein $q\left(x_1\right)$ is the data distribution approximated by the training dataset or sampled from pre-trained model, $p_t\left(x\mid x_1\right)$ is the conditional pdf of $x$ at time $t$ given $x_1$ (Tong et al., 2024a).

We consider a N epoch online fine-tuning scenario, where our vector field model $v_{\theta_\text{ft}}$ is initialized by a pre-trained model $v_{\theta_\text{ref}}$ that can capture the origin data distribution $q(x_1)$, namely the data distribution $p^{ref}(x_1)$ induced by pre-trained model $v_{\theta_\text{ref}}$ can meet $p^{\theta_\text{ref}}(x_1) \approx q(x_1)$. At each epoch of training, the agent will sample data from reweighted data distribution $p^{\theta_\text{ft}}(x_1)$ induced by $v_{\theta_\text{ft}}$, and use the newly sampled data to train the model under the Reward Weighted CFM Loss, then the online reward weighted CFM loss can be writen as:

$$\mathcal{L}_{\text{ORW}-\text{CFM}}(\theta) = \mathbb{E}_{t\sim\mathcal{U}[0,1],x_1\sim p^{\theta_\text{ft}}(x_1),x\sim p_t(x|x_1)}\left[w\left(x_1\right)\left\|v_t(x;\theta) - u_t\left(x\mid x_1\right)\right\|^2\right], \quad (61)$$

wherein $p^{\theta_\text{ft}}\left(x_1\right)$ is the re-weighted data distribution given by fine-tuned vector fields $v_{\theta_\text{ft}}$. Then, we can have the following (For ease of reading, we rewrite the Theorem 2 as follows):

**Theorem** (Online Reward Weighted CFM). *Let $q\left(x_1\right)$ be the initial data distribution, and $w\left(x_1\right)$ be a non-negative weighting function integrable with respect to $q\left(x_1\right)$. Assume that at each epoch $n$, the model $v_\theta$ perfectly learns the distribution implied by the Online Reward-Weighted CFM loss. Then, the data distribution after $N$ epochs is given by:*

$$q_\theta^N\left(x_1\right) = \frac{w\left(x_1\right)^N q\left(x_1\right)}{Z_N} \quad (62)$$

*where $Z_N = \int_\mathcal{X} w\left(x_1\right)^N q\left(x_1\right) dx_1$ is the normalization constant ensuring $q_\theta^N\left(x_1\right)$ is a valid probability distribution.*

*Proof of Theorem 2.* We can use induction to present the proof.

**Example 1** (Base Case $N = 1$). *We first consider the basic case, where $N = 1$, namely single epoch of learning, which equals to offline RW-CFM. Then, The initial data distribution is $q_\theta^0\left(x_1\right) = q\left(x_1\right)$.*

*According to Theorem 1, after applying the online reward weighted CFM loss, we can obtain the new distribution is:*

$$q_\theta^1\left(x_1\right) = \frac{w\left(x_1\right) q\left(x_1\right)}{Z_1} \quad (63)$$

*where $Z_1 = \int_\mathcal{X} w\left(x_1\right) q\left(x_1\right) dx_1$*

Assume that after $n - 1$ epochs, the distribution is:

$$q_\theta^{n-1}\left(x_1\right) = \frac{w\left(x_1\right)^{n-1} q\left(x_1\right)}{Z_{n-1}} \quad (64)$$

with $Z_{n-1} = \int_\mathcal{X} w\left(x_1\right)^{n-1} q\left(x_1\right) dx_1$.

According to Theorem 1, at epoch $n$, the weighted loss leads to an effective distribution:

$$q_\theta^n\left(x_1\right) = \frac{w\left(x_1\right) q_\theta^{n-1}\left(x_1\right)}{Z_n} \quad (65)$$

where $Z_n = \int_\mathcal{X} w\left(x_1\right) q_\theta^{n-1}\left(x_1\right) dx_1$. Substituting $q_\theta^{n-1}\left(x_1\right)$:

$$q_\theta^n(x_1) = \frac{w(x_1)\left(\frac{w(x_1)^{n-1}q(x_1)}{Z_{n-1}}\right)}{Z_n} = \frac{w(x_1)^n q(x_1)}{Z_{n-1}Z_n} \tag{66}$$

Define $\tilde{Z}_n = Z_{n-1}Z_n$, then:

$$q_\theta^n(x_1) = \frac{w(x_1)^n q(x_1)}{\tilde{Z}_n} \tag{67}$$

By induction, $\tilde{Z}_n = \prod_{k=1}^n Z_k$, and we can write:

$$q_\theta^n(x_1) = \frac{w(x_1)^n q(x_1)}{\prod_{k=1}^n Z_k} \tag{68}$$

Then, the proof concludes.

$\square$

**Corollary 5** (Normalization Constant of ORW-CFM). *The normalization constant after $N$ epochs is:*

$$Z_N = \int_{\mathcal{X}} w(x_1)^N q(x_1)\, dx_1 \tag{69}$$

*Proof of Corollary 5.* By induction on the normalization constants $Z_n$ and the fact that $q_\theta^n(x_1)$ must integrate to 1 , we have:

$$Z_N = \prod_{k=1}^N Z_k = \int_{\mathcal{X}} w(x_1)^N q(x_1)\, dx_1 \tag{70}$$

$\square$

**Remark** (RL Perspective of ORW-CFM). *From a RL perspective, this iterative reweighting process can be interpreted as an analog to policy iteration:*

- *Policy Representation: The data distribution $q_\theta^n(x_1)$ represents the policy at epoch $n$.*

- *Policy Improvement: The application of the weighting function $w(x_1) \propto r(x_1)$ corresponds to evaluating and improving the policy based on the rewards.*

- *Value Estimation: The weighting function can be seen as the expected return or value function guiding the policy update.*

- *Exploration Exploitation Trade-off: Similar to the exploration-exploitation trade-off, there is a balance between focusing on high-reward regions and maintaining diversity.*

*This process also aligns with the concept of importance sampling in RL, where the probability of selecting certain actions (data points) is adjusted based on their estimated value, and enhance the high-rewarded probability area and reduce the low-rewarded probability area.*

**Corollary 6** (Convergence Behavior). *According to 2, as $N \to \infty$, the distribution $q_\theta^N(x_1)$ may become increasingly concentrated on the subset of $\mathcal{X}$ where $w(x_1)$ is maximized.*

**Corollary 7** (Mode Collapse Risk/Overoptimization). *Overemphasis on high-reward regions can lead to a lack of diversity in generated samples, similar to mode collapse in GANs (Goodfellow et al., 2014) or overoptimization in fine-tuning diffusion models (Black et al., 2024).*

In this paper, we introduce two methods to handle the Overoptimization and ease the mode collapse risk in online RW-CFM algorithms. The first is to introduce a W2 distance bound between fine-tuned model $\theta_{\text{ft}}$ and reference model $\theta_{\text{ref}}$, which works similar to the KL bound in previous RL (Schulman et al., 2017) and RLHF methods (Black et al., 2024; Rafailov et al., 2023). The second is to incorporate a entropy regularization (i.e., $w(x_1) = \exp(\tau * x_1)$) or smoothing the weighting function (i.e., $w(x_1) = \text{Softmax}(\tau * x_1)$) can mitigate excessive concentration or extremely greedy policy.

**Lemma 4** (Limiting Case). *We now consider the limiting cases where $N \to \infty$:*

*If $w(x_1) > 0$ for all $x_1 \in \mathcal{X}$ and attains its maximum at $x_1^*$, then as $N \to \infty$, the distribution $q_\theta^N(x_1)$ converges to a Dirac delta function centered at $x_1^*$ :*

$$\lim_{N \to \infty} q_\theta^N(x_1) = \delta(x_1 - x_1^*) \tag{71}$$

*Proof of Lemma 1.* Given that $w(x_1) > 0$ for all $x_1 \in \mathcal{X}$ and attains its maximum at $x_1^*$, we define:

$$\epsilon(x_1) = \frac{w(x_1)}{w(x_1^*)} \tag{72}$$

which means, $\epsilon(x_1^*) = 1$ since $w(x_1^*)/w(x_1^*) = 1$, and $0 \leq \epsilon(x_1) < 1$ for $x_1 \neq x_1^*$, since $w(x_1) < w(x_1^*)$.

Then, we can rewrite $q_\theta^N(x_1)$ using $\epsilon(x_1)$ as:

$$q_\theta^N(x_1) = \frac{[w(x_1^*)]^N \epsilon(x_1)^N q(x_1)}{Z_N} \tag{73}$$

Then for $x_1 \neq x_1^*$, we can have: $\epsilon(x_1)^N \to 0$ as $N \to \infty$ since $\epsilon(x_1) < 1$. Thus $q_\theta^N(x_1) \to 0, \forall x_1 \neq x_1^*$.

And for $x_1 = x_1^*$, we can have $\epsilon(x_1^*)^N = 1$, and $q_\theta^N(x_1^*) = \frac{[w(x_1^*)]^N q(x_1)}{Z_N}$.

And we can have the normalization constant as follows:

$$Z_N = \int_{\mathcal{X}} w(x_1)^N q(x_1) \, dx_1 = [w(x_1^*)]^N \int_{\mathcal{X}} \epsilon(x_1)^N q(x_1) \, dx_1. \tag{74}$$

Similarly, we can obtain $Z_N \approx [w(x_1^*)]^N q(x_1^*)$ as $N \to \infty$.

Then, we can have the limit behavior:

For $x_1 \neq x_1^*$ :

$$q_\theta^N(x_1) = \frac{[w(x_1^*)]^N \epsilon(x_1)^N q(x_1)}{[w(x_1^*)]^N q(x_1^*)} = \frac{\epsilon(x_1)^N q(x_1)}{q(x_1^*)} \to 0. \tag{75}$$

For $x_1 = x_1^*$ :

$$q_\theta^N(x_1^*) = \frac{[w(x_1^*)]^N q(x_1^*)}{[w(x_1^*)]^N q(x_1^*)} = 1 \tag{76}$$

Then, we can have:

$$\lim_{N \to \infty} q_\theta^N(x_1) = \delta(x_1 - x_1^*). \tag{77}$$

$\square$

According to Lemma 1, iteratively utilize the ORW-CFM loss to fine-tune the flow matching model without bounding the distance between reference model and fine-tuned model may lead us to a greedy policy over $x_1$, which may induce the overoptimization problem (Black et al., 2024). In this paper we introduce W2 Distance into the ORW-CFM training to bound the distance between reference model and fine-tuned model.

## C.4 WASSERSTEIN-2 DISTANCE FOR FLOW MATCHING

We consider two flow matching models parameterized by $\theta_1$ and $\theta_2$, inducing time-evolving data distributions $p_t^{\theta_1}(x)$ and $p_t^{\theta_2}(x)$, respectively. The models define vector fields $v^{\theta_1}(t, x)$ and $v^{\theta_2}(t, x)$ that transport an initial distribution $p_0(x)$ to final distributions $p_1^{\theta_1}(x)$ and $p_1^{\theta_2}(x)$ at time $t = 1$.

We aim to bound the Wasserstein-2 distance between $p_1^{\theta_1}(x)$ and $p_1^{\theta_2}(x)$ in terms of the difference between $v^{\theta_1}$ and $v^{\theta_2}$.

First, we can write down the Wasserstein-2 Distance as follows (For ease of reading, we rewrite the Theorem 3 to be proved in this section as follows):

**Definition C.1** (Wasserstein-2 Distance). *Given two probability measures $\mu$ and $\nu$ on $\mathbb{R}^n$, the squared Wasserstein- 2 distance between $\mu$ and $\nu$ is defined as:*

$$W_2^2(\mu, \nu) = \inf_{\gamma \in \Pi(\mu,\nu)} \int_{\mathbb{R}^n \times \mathbb{R}^n} \|x - y\|^2 d\gamma(x, y) \tag{78}$$

*where $\Pi(\mu, \nu)$ denotes the set of all couplings of $\mu$ and $\nu$; that is, all joint distributions $\gamma$ on $\mathbb{R}^n \times \mathbb{R}^n$ with marginals $\mu$ and $\nu$.*

**Theorem** (W2 Bound for Flow Matching). *Assume that $v^{\theta_2}(t, x)$ is Lipschitz continuous in $x$ with Lipschitz constant L. Then, the squared Wasserstein-2 distance between the distributions $p_1^{\theta_1}$ and $p_1^{\theta_2}$ induced by the flow matching models at time $t = 1$ satisfies:*

$$W_2^2\left(p_1^{\theta_1}, p_1^{\theta_2}\right) \le e^{2L} \int_0^1 \mathbb{E}_{x \sim p_s^{\theta_1}}\left[\left\|v^{\theta_1}(s, x) - v^{\theta_2}(s, x)\right\|^2\right] ds \tag{79}$$

Before providing the full proof of Theorem 3, we first want to prove the Differential Flow Discrepancy Lemma:

**Lemma 5** (Differential Flow Discrepancy Lemma). *Let $\phi_t^{\theta_i}(x_0)$ be the flow map induced by $v^{\theta_i}$ starting from $x_0$. The difference $\Delta_t(x_0) = \phi_t^{\theta_1}(x_0) - \phi_t^{\theta_2}(x_0)$ satisfies the differential inequality:*

$$\frac{d}{dt}\|\Delta_t(x_0)\| \le \left\|v^{\theta_1}\left(t, \phi_t^{\theta_1}(x_0)\right) - v^{\theta_2}\left(t, \phi_t^{\theta_1}(x_0)\right)\right\| + L\|\Delta_t(x_0)\| \tag{80}$$

*Proof of Lemma 5.* Let $\Delta_t(x_0) = \phi_t^{\theta_1}(x_0) - \phi_t^{\theta_2}(x_0)$. Then:

$$\frac{d}{dt}\Delta_t(x_0) = v^{\theta_1}\left(t, \phi_t^{\theta_1}(x_0)\right) - v^{\theta_2}\left(t, \phi_t^{\theta_2}(x_0)\right) \tag{81}$$

Adding and subtracting $v^{\theta_2}\left(t, \phi_t^{\theta_1}(x_0)\right)$ :

$$\frac{d}{dt}\Delta_t(x_0) = \underbrace{v^{\theta_1}\left(t, \phi_t^{\theta_1}(x_0)\right) - v^{\theta_2}\left(t, \phi_t^{\theta_1}(x_0)\right)}_{\delta_v(t)} + \underbrace{v^{\theta_2}\left(t, \phi_t^{\theta_1}(x_0)\right) - v^{\theta_2}\left(t, \phi_t^{\theta_2}(x_0)\right)}_{\delta_\phi(t)}. \tag{82}$$

**Assumption 1** (Lipschitz Continuity). *Assume that $v^{\theta_2}(t, x)$ is Lipschitz continuous in $x$ with Lipschitz constant $L_v$ :*

$$\left\|v^{\theta_2}(t, x) - v^{\theta_2}(t, y)\right\| \le L_v \|x - y\| \tag{83}$$

By the Lipschitz continuity of $v^{\theta_2}$ :

$$\|\delta_\phi(t)\| \le L\|\Delta_t(x_0)\| \tag{84}$$

Thus:

$$\left\|\frac{d}{dt}\Delta_t(x_0)\right\| \le \|\delta_v(t)\| + L\|\Delta_t(x_0)\| \tag{85}$$

Then, the proof of Lemma 5 concludes.

$\square$

*Proof of Theorem 3.* Using Lemma 5, define $f(t) = \|\Delta_t(x_0)\|$. Then:

$$\frac{d}{dt}f(t) \leq \|\delta_v(t)\| + Lf(t) \tag{86}$$

Multiplying both sides by $e^{-Lt}$:

$$\frac{d}{dt}\left(e^{-Lt}f(t)\right) \leq e^{-Lt}\|\delta_v(t)\| \tag{87}$$

Integrating from 0 to $t$:

$$e^{-Lt}f(t) - f(0) \leq \int_0^t e^{-Ls}\|\delta_v(s)\|\,ds \tag{88}$$

**Assumption 2.** *Assume that the initial noise distribution $x_0 \sim p(x_0)$ are shared in two flow matching models, as thus $f(0) = \|\Delta_0(x_0)\| = 0$*

Since $f(0) = 0$:

$$f(t) \leq e^{Lt}\int_0^t e^{-Ls}\|\delta_v(s)\|\,ds \tag{89}$$

At $t = 1$:

$$f(1) \leq e^L \int_0^1 e^{-Ls}\left\|v^{\theta_1}\left(s, \phi_s^{\theta_1}(x_0)\right) - v^{\theta_2}\left(s, \phi_s^{\theta_1}(x_0)\right)\right\|\,ds \tag{90}$$

Taking the expectation over $x_0 \sim p_0$ and applying Jensen's inequality:

$$\mathbb{E}_{x_0}\left[f(1)^2\right] \leq e^{2L}\left(\int_0^1 \mathbb{E}_{x \sim p_s^{\theta_1}}\left[\left\|v^{\theta_1}(s,x) - v^{\theta_2}(s,x)\right\|^2\right]\,ds\right) \tag{91}$$

**Lemma 6.** *Let $\phi_t^{\theta_i}(x_0)$ be the flow map induced by $v^{\theta_i}$ starting from $x_0$. The flow matching models define vector fields $v^{\theta_1}(t,x)$ and $v^{\theta_2}(t,x)$ that transport an initial distribution $p_0(x)$ to final distributions $p_1^{\theta_1}(x)$ and $p_1^{\theta_2}(x)$ at time $t = 1$. Let $\Delta_t(x_0) = \phi_t^{\theta_1}(x_0) - \phi_t^{\theta_2}(x_0)$ and $f(t) = \|\Delta_t(x_0)\|$. Then, the squared Wasserstein-2 distance between the distributions $p_1^{\theta_1}$ and $p_1^{\theta_2}$ induced by the flow matching models at time $t = 1$ satisfies:*

$$W_2^2\left(p_1^{\theta_1}, p_1^{\theta_2}\right) \leq \mathbb{E}_{x_0}\left[f(1)^2\right] \tag{92}$$

According to Lemma 6, we have:

$$W_2^2\left(p_1^{\theta_1}, p_1^{\theta_2}\right) \leq \mathbb{E}_{x_0}\left[f(1)^2\right] \leq e^{2L}\int_0^1 \mathbb{E}_{x \sim p_s^{\theta_1}}\left[\left\|v^{\theta_1}(s,x) - v^{\theta_2}(s,x)\right\|^2\right]\,ds \tag{93}$$

Then, the proof of Theorem 3 concludes. $\square$

*Proof of Lemma 6.* Recall the definition of the Wasserstein-2 distance:

$$W_2^2\left(p_1^{\theta_1}, p_1^{\theta_2}\right) = \inf_{\gamma \in \Pi\left(p_1^{\theta_1}, p_1^{\theta_2}\right)} \int_{\mathbb{R}^n \times \mathbb{R}^n} \|x - y\|^2 d\gamma(x,y) \tag{94}$$

We construct a specific coupling $\gamma$ between $p_1^{\theta_1}$ and $p_1^{\theta_2}$ by mapping the same initial sample $x_0 \sim p_0$ through both flow maps:

- $x_1^{\theta_1} = \phi_1^{\theta_1}(x_0)$.

- $x_1^{\theta_2} = \phi_1^{\theta_2}(x_0)$.

This coupling $\gamma$ is defined via the joint distribution of $\left(x_1^{\theta_1}, x_1^{\theta_2}\right)$.

By the definition of the Wasserstein-2 distance, for any coupling $\gamma$ :

$$W_2^2\left(p_1^{\theta_1}, p_1^{\theta_2}\right) \leq \int_{\mathbb{R}^n \times \mathbb{R}^n} \|x - y\|^2 d\gamma(x, y) \tag{95}$$

Using our constructed coupling:

$$\int_{\mathbb{R}^n \times \mathbb{R}^n} \|x - y\|^2 d\gamma(x, y) = \mathbb{E}_{x_0}\left[\left\|\phi_1^{\theta_1}(x_0) - \phi_1^{\theta_2}(x_0)\right\|^2\right] = \mathbb{E}_{x_0}\left[f(1)^2\right] \tag{96}$$

Thus, we have:

$$W_2^2\left(p_1^{\theta_1}, p_1^{\theta_2}\right) \leq \mathbb{E}_{x_0}\left[f(1)^2\right] \tag{97}$$

Then, the Proof concludes. $\qquad\square$

**Corollary 8** (Uniform Vector Field Deviation Bound). *If there exists a constant $\varepsilon > 0$ such that for all $s \in [0, 1]$ :*

$$\mathbb{E}_{x \sim p_s^{\theta_1}}\left[\left\|v^{\theta_1}(s, x) - v^{\theta_2}(s, x)\right\|^2\right] \leq \varepsilon^2 \tag{98}$$

*then the Wasserstein-2 distance between the induced distributions is bounded by:*

$$W_2\left(p_1^{\theta_1}, p_1^{\theta_2}\right) \leq \varepsilon e^L \tag{99}$$

*Proof of Corollary 8.* Given the assumption in Corollary 8, from Theorem 3:

$$W_2^2\left(p_1^{\theta_1}, p_1^{\theta_2}\right) \leq e^{2L} \int_0^1 \varepsilon^2 ds = \varepsilon^2 e^{2L} \tag{100}$$

Thus:

$$W_2\left(p_1^{\theta_1}, p_1^{\theta_2}\right) \leq \varepsilon e^L \tag{101}$$

$\qquad\square$

**Corollary 9** (W2 Bound for Flow Matching Fine-tuning). *Considering we have two flow matching models, the fine-tuned model $v^{\theta_{\mathrm{ft}}}$ and the reference/pre-trained model $v^{\theta_{ref}}$. According to Theorem 3, we can bound the Wasserstein-2 $(W_2)$ distance between the distributions $p_{\theta_{\mathrm{ft}}}(x_1)$ and $p_{\theta_{ref}}(x_1)$ induced by these two flow matching models:*

$$\mathbb{E}_{x_0}\left[\|\Delta_1(x_0)\|^2\right] \leq e^{2L_v} \int_0^1 \mathbb{E}_{x \sim p_s(x; \theta_{ft})}\left[\left\|v^{\theta_{\mathrm{ft}}}(s, x) - v^{\theta_{\mathrm{ref}}}(s, x)\right\|^2\right] ds \tag{102}$$

*Proof of Corollary 9.* Set $\theta_1 = \theta_{\mathrm{ft}}$ and $\theta_2 = \theta_{\mathrm{ref}}$, according to the Theorem 3, the proof concludes.
$\qquad\square$

**Corollary 10** (Monte Carlo Sampling for W2 Bound in Flow Matching). *However, the integral of time $t$ in Theorem 3 is normally intractable and hard to obtain. In practice, we can use Monte Carlo sampling to approximate the integral:*

$$\int_0^1 \mathbb{E}_{x \sim p_s(x;\theta_1)} \left[ \left\| v^{\theta_1}(s,x) - v^{\theta_2}(s,x) \right\|^2 \right] ds$$
$$= \mathbb{E}_{t \sim U(0,1), x_1 \sim q(x_1;\theta_1), x \sim p_t(x|x_1)} \left[ \left\| v^{\theta_1}(t,x) - v^{\theta_2}(t,x) \right\|^2 \right]$$

According to Corollary 10, we can obtain the W2 bound RW-CFM loss as follows:

$$\mathcal{L}_{RW-CFM-W2} = \mathbb{E}_{t \sim U(0,1), x_1 \sim q(x_1), x \sim p_t(x|x_1)} [w(x_1) \| v_t(x;\theta) - u_t(x \mid x_1) \|^2$$
$$+ \alpha * \| v_{\theta_{ft}}(t,x) - v_{\theta_{ref}}(t,x) \|^2],$$

wherein $\theta_{ft} = \theta$ is our fine-tuned model, and $\theta_{ref}$ is reference/pre-trained model, $\alpha$ is the trade-off coefficient, indicates how close we want the final convergence point to be to the reference model.

Also, according to Corollary 10, we can obtain the W2 bound ORW-CFM loss as follows:

$$\mathcal{L}_{ORW-CFM-W2} = \mathbb{E}_{t \sim U(0,1), x_1 \sim q(x_1;\theta_{ft}), x \sim p_t(x|x_1)} [w(x_1) \| v_t(x;\theta) - u_t(x \mid x_1) \|^2$$
$$+ \alpha * \| v_{\theta_{ft}}(t,x) - v_{\theta_{ref}}(t,x) \|^2],$$

wherein $\alpha$ is the trade-off coefficient, indicates how close we want the final convergence point to be to the reference model (See the Divergence Controlled by $\alpha$ in 4).

## C.5 RW-CFM with W2 Distance Bound

In this section, we provide a detailed analysis of how the inclusion of a Wasserstein regularization term in the Reward-Weighted Conditional Flow Matching (RW-CFM) loss affects the induced data distribution $p_{\text{new}}(x_1)$. Specifically, we prove that the new distribution becomes proportional to $w(x_1) q(x_1) \exp(-\alpha D(x_1))$, where $D(x_1)$ is a divergence measure between the fine-tuned model and the reference model.

**Theorem 7** (Learned Distribution Induced by RW-CFM-W2 Loss). *Let $w : \mathcal{X} \rightarrow [0, \infty)$ be a measurable weighting function, $q(x_1)$ be the original data distribution, and $\theta_{ref}$ be a fixed reference model. Consider the loss function:*

$$
\begin{aligned}
\mathcal{L}_{\text{RW-CFM-W2}}(\theta) = \mathbb{E}_{t \sim \mathcal{U}[0,1], x_1 \sim q(x_1), x \sim p_t(x|x_1)} [w(x_1) \|v_t(x; \theta) - u_t(x \mid x_1)\|^2 \\
+ \alpha \|v_t(x; \theta) - v_t(x; \theta_{ref})\|^2]
\end{aligned}
\tag{103}
$$

*where $u_t(x \mid x_1)$ is the true vector field conditioned on $x_1$, $v_t(x; \theta)$ is the model's vector field, and $\alpha > 0$ is the regularization coefficient. Then, minimizing $\mathcal{L}_{\text{RW-CFM-W2}}(\theta)$ over $\theta$ induces a new data distribution:*

$$
p_{\text{new}}(x_1) \propto w(x_1) q(x_1) \exp(-\beta D(x_1))
\tag{104}
$$

*where $D(x_1) = \mathbb{E}_{t, x \sim p_t(x|x_1)} \left[ \|v^\theta(t, x) - v^{\theta_{ref}}(t, x)\|^2 \right]$, $\beta = \gamma\alpha$, $\gamma > 0$ is a scaling constant.*

*Proof of Theorem 7.* We can re-write the RW-CFM-W2 Loss as follows:

$$
\mathcal{L}_{\text{RW-CFM-W2}}(\theta) = \mathbb{E}_{t, x_1, x} \left[ w(x_1) \|v_t(x; \theta) - u_t(x \mid x_1)\|^2 + \alpha \|v^\theta(t, x) - v^{\theta_{ref}}(t, x)\|^2 \right],
\tag{105}
$$

where $\alpha$ controls the regularization strength.

Furthermore, we can write the loss function as an expectation over $x_1$ :

$$
\mathcal{L}_{\text{total}}(\theta) = \int q(x_1) [w(x_1) L_{\text{cfm}}(x_1) + \alpha L_{\text{reg}}(x_1)] \, dx_1
\tag{106}
$$

where the first term gives $L_{\text{cfm}}(x_1) = \mathbb{E}_{t, x \sim p_t(x|x_1)} \left[ \|v_t(x; \theta) - u_t(x \mid x_1)\|^2 \right]$, and the second term gives $L_{\text{reg}}(x_1) = \mathbb{E}_{t, x \sim p_t(x|x_1)} \left[ \|v^\theta(t, x) - v^{\theta_{ref}}(t, x)\|^2 \right]$.

**Assumption 3.** *Assuming that the expectation $L_{\text{reg}}(x_1)$ has an upper bound $\varepsilon$ :*

$$
L_{\text{reg}}(x_1) \leq \varepsilon, \quad \forall x_1
\tag{107}
$$

*Since we're interested in how $L_{\text{reg}}(x_1)$ varies with $x_1$, we can let $D(x_1) = L_{\text{reg}}(x_1)$.*

Then, our total loss is:

$$
\mathcal{L}_{\text{total}}(\theta) = \int q(x_1) [w(x_1) L_{\text{cfm}}(x_1) + \alpha D(x_1)] \, dx_1
\tag{108}
$$

We consider the optimization over $\theta$, which indirectly affects the distribution $p_{\text{new}}(x_1)$ that the model learns.

Then, consider the model as implicitly assigning higher importance to values of $x_1$ that minimize the total loss per $x_1$ :

$$
\tilde{L}(x_1) = w(x_1) L_{\text{cfm}}(x_1) + \alpha D(x_1)
\tag{109}
$$

**Remark** (Probabilistic Perspective). *In optimization, samples $x_1$ with lower $\tilde{L}(x_1)$ have a higher impact on training. To make this explicit, consider a probabilistic model where the likelihood of $x_1$ is proportional to $\exp\left(-\gamma \tilde{L}(x_1)\right)$, where $\gamma$ is a positive constant (Hinton, 2002).*

Therefore, the induced distribution over $x_1$ becomes (detailed proof can see App. C.1):

$$p_{\text{new}}(x_1) \propto q(x_1) \exp\left(-\gamma \left[w(x_1) L_{\text{cfm}}(x_1) + \alpha D(x_1)\right]\right) \tag{110}$$

We can also induce the similar results from the perspective of energy function (LeCun et al., 2006).

**Lemma 7** (Energy Function Perspective). *The per-sample loss contribution $\tilde{L}(x_1)$ can be interpreted as an energy function $E(x_1)$, and the induced distribution follows a Boltzmann distribution (LeCun et al., 2006):*

$$\begin{aligned} p_{new}(x_1) &\propto q(x_1) \exp\left(-E(x_1)\right) \\ &\propto q(x_1) \exp\left(-\gamma\left[w(x_1) L_{cfm}(x_1) + \alpha D(x_1)\right]\right) \end{aligned} \tag{111}$$

*Proof of Lemma 7.* By interpreting the per-sample loss as an energy $E(x_1)$, we have:

$$\begin{aligned} E(x_1) &= \tilde{L}(x_1) \\ &= w(x_1) L_{\text{cfm}}(x_1) + \alpha D(x_1) \end{aligned} \tag{112}$$

The induced distribution is then:

$$\begin{aligned} p_{\text{new}}(x_1) &\propto q(x_1) \exp\left(-E(x_1)\right) \\ &\propto q(x_1) \exp\left(-\gamma\left[w(x_1) L_{\text{cfm}}(x_1) + \alpha D(x_1)\right]\right) \end{aligned} \tag{113}$$

This aligns with the Boltzmann distribution in statistical mechanics, where higher energy states (higher loss) are less probable. $\qquad\square$

Besides, based on Theorem 6, we can have:

$$\begin{aligned} p_{\text{new}}(x_1) &\propto q(x_1) \exp\left(-\gamma * \tilde{L}(x_1)\right) \\ &\propto q(x_1) \exp\left(-\gamma\left[w(x_1) L_{\text{cfm}}(x_1) + \alpha D(x_1)\right]\right) \end{aligned} \tag{114}$$

**Assumption 4.** *Assuming $L_{cfm}(x_1)$ is approximately constant or negligible compared to $D(x_1)$, namely:*

- *When the model fits the data well, $L_{cfm}(x_1)$ is small. (i.e., $w(x_1)$ dominates)*

- *The term $w(x_1) L_{cfm}(x_1)$ becomes proportional to $w(x_1)$.*

**Remark.** *Generally speaking, if we fine-tune from a well-learned pre-trained model, the model can usually fit the data well, so $L_{cfm}(x_1)$ is rather small. In practice, $w(x_1)$ can generally dominates.*

Thus, we can write:

$$p_{\text{new}}(x_1) \propto q(x_1) w(x_1) \exp\left(-\beta D(x_1)\right) \tag{115}$$

where $\beta = \gamma\alpha$ absorbs the constants. Then the proof concludes.

$$\square$$

**Remark** (Importance Sampling Perspective). *We can also derive similar results from the perspective of importance sampling. We start from the total loss function:*

$$
\begin{aligned}
\mathcal{L}_{total}\left(\theta\right) &= \int w\left(x_1\right) q\left(x_1\right) \left[L_{cfm}\left(x_1\right) + \frac{\alpha}{w\left(x_1\right)} D\left(x_1\right)\right] dx_1 \\
&= E_{x \sim w\left(x_1\right) q\left(x_1\right)} \left[\left[L_{cfm}\left(x_1\right) + \frac{\alpha}{w\left(x_1\right)} D\left(x_1\right)\right]\right]
\end{aligned}
\tag{116}
$$

*We denote the reward weighted distribution as $p_w(x)$:*

$$
p_w(x) = w\left(x_1\right) q\left(x_1\right)
\tag{117}
$$

*Then the loss function can be re-write as follows:*

$$
\mathcal{L}_{total}\left(\theta\right) = E_{x \sim p_w(x)} \left[\left[L_{cfm}\left(x_1\right) + \frac{\alpha}{w\left(x_1\right)} D\left(x_1\right)\right]\right]
\tag{118}
$$

*Then the induced data distribution $p_{new}(x)$ becomes:*

$$
p_{new}\left(x_1\right) \propto p_w\left(x_1\right) \exp\left(-\gamma \left[L_{cfm}\left(x_1\right) + \frac{\alpha}{w\left(x_1\right)} D\left(x_1\right)\right]\right)
\tag{119}
$$

*Under Assumption 4, we can obtain:*

$$
\begin{aligned}
p_{new}\left(x_1\right) &\propto p_w\left(x_1\right) \exp\left(-\frac{\gamma * \alpha}{w\left(x_1\right)} D\left(x_1\right)\right) \\
&\propto p_w\left(x_1\right) \exp\left(-\frac{\beta}{w\left(x_1\right)} D\left(x_1\right)\right) \\
&\propto w(x_1) q\left(x_1\right) \exp\left(-\frac{\beta}{w\left(x_1\right)} D\left(x_1\right)\right)
\end{aligned}
\tag{120}
$$

*In practice, we can assume that $\forall x_1, \beta \gg w(x_1)$, then $\beta$ dominates, and we can obtain:*

$$
p_{new}\left(x_1\right) \propto w(x_1) q\left(x_1\right) \exp\left(-\beta D\left(x_1\right)\right)
\tag{121}
$$

*Also, we can set $\alpha = \alpha_0 * w(x_1)$, then:*

$$
\exp\left(-\frac{\beta}{w\left(x_1\right)} D\left(x_1\right)\right) = \exp\left(-\beta_0 D\left(x_1\right)\right)
\tag{122}
$$

*wherein $\beta_0 = \alpha_0 * \gamma$.*

**Corollary 11.** *As $\alpha \to 0, \beta \to 0$, the induced distribution $p_{new}\left(x_1\right)$ approaches $w\left(x_1\right) q\left(x_1\right) / Z$, recovering the standard RW-CFM result.*

*Proof of Corollary 11.* When $\alpha \to 0, \beta \to 0$, the exponential term $\exp\left(-\beta D\left(x_1\right)\right) \to 1$, so:

$$
p_{new}\left(x_1\right) \to \frac{w\left(x_1\right) q\left(x_1\right)}{Z}
\tag{123}
$$

where $Z$ is the normalization constant ensuring that $p_{new}\left(x_1\right)$ integrates to 1. $\square$

**Corollary 12.** *As $\alpha \to \infty, \beta \to \infty$, the induced distribution $p_{new}\left(x_1\right)$ concentrates on minimizing $D\left(x_1\right)$, effectively aligning with the reference model's distribution (See cases with increased $\alpha$, such as $\alpha = 10$, in Fig. 4).*

*Proof of Corollary 12.* When $\alpha \to \infty, \beta \to \infty$, the exponential term $\exp\left(-\beta D\left(x_1\right)\right)$ approaches zero unless $D\left(x_1\right) = 0$. Therefore, $p_{new}\left(x_1\right)$ concentrates on the set of $x_1$ where $v_t\left(x; \theta_{ref}\right)$ matches $u_t\left(x \mid x_1\right)$. Namely, the fine-tuned model remains unchanged from the reference model. $\square$

**Remark.** *The theorem 7 demonstrates that incorporating the Wasserstein regularization term into the RW-CFM loss results in an induced data distribution that not only depends on the reward weighting $w(x_1)$ but also exponentially penalizes data points based on their divergence $D(x_1)$ from the reference model (See Trade-off curve via traversing $\alpha$ Fig. 4). This provides a mechanism to control the trade-off between fitting the reward weighted data and adhering to the behavior of a pre-trained or reference model (i.e., reward-diversity trade-off).*

*The divergence $D(x_1)$ acts as a measure of how much the true dynamics $u_t(x \mid x_1)$ differ from the reference model's dynamics $v_t(x; \theta_{ref})$ along trajectories starting from $x_1$. By adjusting $\alpha$, we can control the extent to which the model prioritizes matching the reference model versus fitting the reward-weighted data.*

## C.6 ORW-CFM WITH W2 DISTANCE BOUND

Similar to Theorem 7, we can also derive the induced data distribution learned under the online reward weighted CFM loss and W2 Distance Bound.

According to Theorem 3, We can directly introduce the W2 Distance Bound into ORW-CFM loss equation 5:

$$\mathcal{L}_{ORW-CFM-W2} = \mathbb{E}_{t\sim U(0,1),x_1\sim q(x_1;\theta_{\mathrm{ft}}),x\sim p_t(x|x_1)}[w(x_1)\|v_t(x;\theta_{\mathrm{ft}}) - u_t(x\mid x_1)\|^2 \\ + \alpha * \|v_{\theta_{\mathrm{ft}}}(t,x) - v_{\theta_{\mathrm{ref}}}(t,x)\|^2], \tag{124}$$

wherein $\alpha$ is a trade-off coefficient.

For ease of reading, we rewrite the Theorem 4 to be proved in this section as follows:

**Theorem.** *Under the online reinforcement learning setup with ORW-CFM-W2 loss, the data distribution after n epochs evolves according to:*

$$q_\theta^n(x_1) \propto \left[w(x_1)\, q_\theta^{n-1}(x_1)\exp\left(-\beta D^{n-1}(x_1)\right)\right], \tag{125}$$

*where,* $D^{n-1}(x_1) = \mathbb{E}_{t,x\sim p_t(x|x_1)}\left[\left\|v^{\theta^{n-1}}(t,x) - v^{\theta_{ref}}(t,x)\right\|^2\right]$, $\beta = \gamma\alpha$, $\theta^{n-1}$ *denotes the model parameters after epoch* $n-1$.

*Proof of Theorem 4.* Similar to the proof of 2, we also proceed by induction.

**Case 3** (Base Case $(n=1)$). *At* $n=1$, $q_\theta^1(x_1)$ *is obtained by training on data sampled from* $q(x_1)$. *From previous analysis, we have:*

$$q_\theta^1(x_1) \propto w(x_1)\, q(x_1)\exp\left(-\beta D^0(x_1)\right) \tag{126}$$

*where* $D^0(x_1)$ *measures the discrepancy between the initial model (before training) and the reference model.*

Assume that at epoch $n-1$, the data distribution is $q_\theta^{n-1}(x_1)$. During epoch $n$, the model is trained using data sampled from $q_\theta^{n-1}(x_1)$. By applying ORW-CFM-W2 Loss, compared to RW-CFM-W2 Loss, the loss function remains the same, but the data distribution changes to $q_\theta^{n-1}(x_1)$.

The total loss associated with $x_1$ at epoch $n$ is:

$$\tilde{L}^n(x_1) = w(x_1)\, L_{\mathrm{cfm}}^n(x_1) + \alpha D^{n-1}(x_1) \tag{127}$$

where $L_{\mathrm{cfm}}^n(x_1)$ is the data fitting loss of CFM at epoch $n$, and $D^{n-1}(x_1)$ is the discrepancy from the reference model using parameters $\theta^{n-1}$.

Similarly, we can assume that the probability of $x_1$ being selected for training at epoch $n$ is proportional to $\exp\left(-\gamma\tilde{L}^n(x_1)\right)$. Therefore:

$$q_\theta^n(x_1) \propto q_\theta^{n-1}(x_1)\exp\left(-\gamma\tilde{L}^n(x_1)\right) \tag{128}$$

Substituting $\tilde{L}^n(x_1)$ :

$$q_\theta^n(x_1) \propto q_\theta^{n-1}(x_1)\exp\left(-\gamma\left[w(x_1)\, L_{\mathrm{cfm}}^n(x_1) + \alpha D^{n-1}(x_1)\right]\right). \tag{129}$$

Assuming $L_{\mathrm{cfm}}^n(x_1)$ is approximately constant or negligible compared to $\alpha D^{n-1}(x_1)$ (i.e., well learned), we simplify:

$$q_\theta^n(x_1) \propto w(x_1)\, q_\theta^{n-1}(x_1)\exp\left(-\beta D^{n-1}(x_1)\right) \tag{130}$$

Then, the induction step completes. This concludes the proof.

Besides, we can also derive the proof via the Importance Sampling Perspective as equation 116. The process is very similar to the offline cases. We just need to substitute $\tilde{L}^n(x_1)$ into equation 116, take out the weights $w(x)$ to form $w(x)q_\theta^{n-1}(x_1)$ and reuse Theorem 6. $\qquad\square$

**Corollary 13** (Recursive Formulation). *The data distribution after $N$ epochs training under $\mathcal{L}_{ORW-CFM-W2}$ is given by:*

$$q_\theta^N(x_1) \propto [w(x_1)]^N q(x_1) \exp\left(-\beta \sum_{n=1}^{N} D^{n-1}(x_1)\right) \tag{131}$$

*Proof of Corollary 13.* By recursively applying Theorem 4, we have:

$$
\begin{aligned}
q_\theta^N(x_1) &\propto w(x_1) q_\theta^{N-1}(x_1) \exp\left(-\beta D^{N-1}(x_1)\right) \\
&\propto w(x_1) \left(w(x_1) q_\theta^{N-2}(x_1) \exp\left(-\beta D^{N-2}(x_1)\right)\right) \exp\left(-\beta D^{N-1}(x_1)\right) \\
&\propto [w(x_1)]^N q(x_1) \exp\left(-\beta \sum_{n=1}^{N} D^{n-1}(x_1)\right)
\end{aligned}
\tag{132}
$$

$\square$

**Corollary 14** (Convergence Behavior). *If $D^n(x_1)$ decreases with each epoch and $w(x_1)$ is constant (i.e., $w(x_1) = w$ ), then $q_\theta^N(x_1)$ converges towards a distribution that places higher probability on data points where the model aligns closely with the reference model (See $\tau = 0$ cases in Fig. 2).*

*Proof of Corollary 14.* As $D^n(x_1)$ decreases, the exponential term $\exp\left(-\beta \sum_{n=1}^{N} D^{n-1}(x_1)\right)$ increases for those $x_1$ where the model aligns better with the reference model. With $w(x_1)$ constant, the weighting does not favor any specific $x_1$, so the distribution $q_\theta^N(x_1)$ is increasingly influenced by the cumulative discrepancy term, leading to higher probabilities for data points where the model matches the reference model. $\qquad\square$

## C.7 EXPONENTIAL WEIGHT FUNCTION

As we discussed, normally we can have $w(x_1) \propto r(x_1)$ and $w(x_1) \propto \mathcal{F}(r(x_1))$. In practice, we normally have several form of the weighting function $\mathcal{F}$ to enable $w(x_1) > 0$ (Peng et al., 2019; Peters & Schaal, 2007; Black et al., 2024), which maps from reward function to weight. In this section, based on the previous derivation, we focus on a special cases of fine-tuning flow matching model with ORW-CFM-W2 loss, where $w(x_1) \propto \exp(\tau * r(x_1))$, wherein $\tau$ is the temperature coefficient.

Then we can simply have the following Theorem (For ease of reading, we rewrite the Theorem 5 to be proved in this section as follows):

**Theorem.** *Given* $w(x_1) = \exp(\tau r(x_1))$, *the induced data distribution after* $N$ *epochs of training under ORW-CFM-W2 Loss is:*

$$q_\theta^N(x_1) \propto \exp\left(\tau N r(x_1) - \beta \sum_{n=1}^{N} D^{n-1}(x_1)\right) q(x_1) \tag{133}$$

*Proof of Theorem 5.* According to Corollary 13, we can re-write the final induced data distribution learned under ORW-CFM-W2 loss function as:

$$q_\theta^N(x_1) \propto [w(x_1)]^N q(x_1) \exp\left(-\beta \sum_{n=1}^{N} D^{n-1}(x_1)\right) \tag{134}$$

Let $r(x_1)$ be a reward function, and $\tau > 0$ a temperature coefficient. The weighting function becomes:

$$w(x_1) = \exp(\tau * r(x_1)) \tag{135}$$

Substituting into equation 134:

$$q_\theta^N(x_1) \propto q(x_1) \exp\left(\tau N r(x_1) - \beta \sum_{n=1}^{N} D^{n-1}(x_1)\right) \tag{136}$$

Then, the proof concludes. $\square$

**Corollary 15** (Limiting Behavior). *If* $\beta \sum_{n=1}^{N} D^{n-1}(x_1)$ *grows proportionally with* $N$, *i.e.,* $\beta \sum_{n=1}^{N} D^{n-1}(x_1) = N\delta(x_1)$, *the induced data distribution becomes:*

$$q_\theta^N(x_1) \propto \exp(N(\tau r(x_1) - \delta(x_1))) q(x_1) \tag{137}$$

*This suggests that the effective reward is adjusted by* $-\delta(x_1)$ *due to the W2 distance bound.*

*Proof of Corollary 15.* Given the proportional growth, we can write:

$$\beta \sum_{n=1}^{N} D^{n-1}(x_1) = N\delta(x_1) \tag{138}$$

Substituting into the expression for $q_\theta^N(x_1)$:

$$q_\theta^N(x_1) \propto \exp(N\tau * r(x_1) - N\delta(x_1)) q(x_1) = \exp(N(\tau * r(x_1) - \delta(x_1))) q(x_1) \tag{139}$$

$\square$

**Corollary 16** (The Effect of W2 Discrepancy). *The W2 distance bound $D^{n-1}(x_1)$ serves as a regularization term that penalizes the discrepancy between the current model $v^{\theta^{n-1}}$ and a reference model $v^{\theta_{ref}}$. This term affects the induced data distribution by effectively adjusting the reward: $\tilde{r}(x_1) = r(x_1) - \frac{\delta(x_1)}{\tau}$.*

*The induced distribution then becomes:*

$$q_\theta^N(x_1) \propto \exp(\tau N \tilde{r}(x_1)) q(x_1) \tag{140}$$

*This adjustment means that the W2 distance bound reduces the attractiveness of certain samples based on how much the model deviates from the reference model when conditioned on those samples (See reward-distance reward-off in Figs. 4).*

According to different hyper-parameter settings, we discuss the limiting behavior of the method of the following cases:

**Case 4** (Dominant Reward Term). *If $D^{n-1}(x_1)$ grows slowly with $N$ or $\beta$ is small, the reward term dominates the regularization term, namely $\tau * r(x_1) \gg \beta \sum_{n=1}^{N} D^{n-1}(x_1)$. The induced distribution $q_\theta^N(x_1)$ concentrates on the $x_1$ maximizing $r(x_1)$, similar to the case without considering $D^{n-1}(x_1)$ (See $\tau = 10$ in Fig. 2).*

**Case 5** (Dominant Regularization Term). *If $D^{n-1}(x_1)$ grows rapidly with $N$ or $\beta$ is large, the regularization term dominates, namely $\beta \sum_{n=1}^{N} D^{n-1}(x_1) \gg \tau * r(x_1)$. The induced distribution is heavily penalized for deviations from the reference model, potentially leading to a distribution similar to the initial distribution $q(x_1)$ (See $\alpha = 10$ in Fig. 3).*

**Case 6** (Balanced Terms). *When both terms are balanced, namely $\tau * r(x_1) \approx \beta \sum_{n=1}^{N} D^{n-1}(x_1)$, the induced distribution reflects a trade-off between maximizing rewards and staying close to the reference model. The model achieves a compromise, focusing on high-reward samples that do not deviate significantly from the reference.*

**Lemma 8** (Growth Rate of $D^{n-1}(x_1)$). *Assuming that $D^{n-1}(x_1) \leq D_{\max}$ for all $n$ and $x_1$, the total W2 distance bound over $N$ epochs satisfies:*

$$\beta \sum_{n=1}^{N} D^{n-1}(x_1) \leq N\beta D_{\max} \tag{141}$$

*Proof of Lemma 8.* Since $D^{n-1}(x_1) \leq D_{\max}$, summing over $n$:

$$\beta \sum_{n=1}^{N} D^{n-1}(x_1) \leq \beta N D_{\max} \tag{142}$$

$\square$

Based on the above derivation, we can discuss the regularization effect of $D^{n-1}(x_1)$ and how to tune the trade-off coefficient $\beta$. The W2 distance bound term acts as a regularizer that:

- Penalizes Model Drift: Prevents the model from deviating excessively from the reference model $v^{\theta_{ref}}$.

- Encourages Smooth Updates: Ensures that updates to the model parameters are gradual, promoting stability during training.

- Balances Exploration and Exploitation: Mitigates the risk of overoptimization to high-reward samples by considering the model's consistency.

Besides, the parameter $\beta$ controls the strength of the regularization:

- Large $\beta$ : Strong regularization, the model remains close to the reference, potentially underexploiting high-reward regions and inducing more diversity/exploration (See $\tau = 1$ in Fig. 11.).

- Small $\beta$ : Weak regularization, the model prioritizes reward maximization, possibly at the cost of stability and induce the over-optimization problem (Black et al., 2024) (See $\tau = 0$ in Fig. 11. )

**Corollary 17** (Optimal $\beta$ for Exploration-Exploitation Trade-off). *An optimal $\beta^*$ exists that balances the influence of the reward term and the W2 distance bound, satisfying:*

$$\beta^* = \frac{\tau \bar{r}}{\bar{D}} \tag{143}$$

*where $\bar{r}$ is the average reward and $\bar{D}$ is the average W2 distance bound.*

*Proof.* Setting the magnitudes of the reward and regularization terms equal, we have:

$$N\tau\bar{r} = N\beta^*\bar{D} \implies \beta^* = \frac{\tau\bar{r}}{\bar{D}} \tag{144}$$

where $\bar{r}$ is the average reward and $\bar{D}$ is the average W2 distance bound. $\qquad\square$

**Remark.** *The regularization terms are crucial for ensuring stable learning dynamics, preventing catastrophic forgetting (Kirkpatrick et al., 2016). Besides, Regularization terms also offer a trust region constraint like TRPO (Schulman et al., 2015) or policy update in proximal area like PPO (Schulman et al., 2017).*

## C.8 BOLTZMANN WEIGHT FUNCTION

In this section, based on the previous derivation, we focus on a special cases of fine-tuning flow matching model with ORW-CFM-W2 loss, where $w(x_1) \propto \text{Softmax}(\tau * r(x_1))$, wherein $\tau$ is the temperature coefficient.

According to Corollary 13, we can also re-write the final induced data distribution learned under ORW-CFM-W2 loss function as:

$$q_\theta^N(x_1) \propto [w(x_1)]^N q(x_1) \exp\left(-\beta \sum_{n=1}^N D^{n-1}(x_1)\right) \tag{145}$$

**Definition C.2** (Boltzmann Weighting Function). *The Boltzmann weighting function over the set of all possible $x_1$ is defined as:*

$$w(x_1) = \text{softmax}(\tau r(x_1)) = \frac{\exp(\tau r(x_1))}{Z} \tag{146}$$

*where $\tau \in \mathbb{R}$ is the temperature parameter, $Z = \sum_{x_1'} \exp(\tau r(x_1'))$ is the partition function ensuring that $\sum_{x_1} w(x_1) = 1$.*

Then, we can have the following:

**Theorem 8.** *With $w(x_1) = \text{softmax}(\tau r(x_1))$, the induced data distribution after $N$ epochs is:*

$$q_\theta^N(x_1) \propto \left(\frac{\exp(\tau r(x_1))}{Z}\right)^N q(x_1) \exp\left(-\beta \sum_{n=1}^N D^{n-1}(x_1)\right) \tag{147}$$

*Proof of Theorem 8.* Starting from the recursive formulation:

$$q_\theta^N(x_1) \propto w(x_1) q_\theta^{N-1}(x_1) \exp\left(-\beta D^{N-1}(x_1)\right) \tag{148}$$

By induction, assume the expression holds for $N-1$ :

$$q_\theta^{N-1}(x_1) \propto \left(\frac{\exp(\tau r(x_1))}{Z}\right)^{N-1} q(x_1) \exp\left(-\beta \sum_{n=1}^{N-1} D^{n-1}(x_1)\right). \tag{149}$$

At epoch $N$ :

$$
\begin{aligned}
q_\theta^N(x_1) &\propto w(x_1) q_\theta^{N-1}(x_1) \exp\left(-\beta D^{N-1}(x_1)\right) \\
&\propto \left(\frac{\exp(\tau r(x_1))}{Z}\right) \left(\frac{\exp(\tau r(x_1))}{Z}\right)^{N-1} q(x_1) \exp\left(-\beta \sum_{n=1}^N D^{n-1}(x_1)\right) \\
&= \left(\frac{\exp(\tau r(x_1))}{Z}\right)^N q(x_1) \exp\left(-\beta \sum_{n=1}^N D^{n-1}(x_1)\right)
\end{aligned} \tag{150}
$$

Thus, the expression holds for $N$. Then, the proof concludes. $\square$

**Corollary 18** (Simplifying Induced Distribution). *The induced data distribution can be expressed as:*

$$q_\theta^N(x_1) \propto \exp\left(N\tau r(x_1) - \beta \sum_{n=1}^N D^{n-1}(x_1)\right) q(x_1) \tag{151}$$

*up to a normalization constant independent of $x_1$.*

*Proof of Corollary 18.* Since $Z$ is constant with respect to $x_1$, we can absorb $Z^N$ into the proportionality constant:

$$
\begin{aligned}
q_\theta^N(x_1) &\propto \frac{\exp\left(N\tau r(x_1)\right)}{Z^N} q(x_1) \exp\left(-\beta \sum_{n=1}^{N} D^{n-1}(x_1)\right) \\
&\propto \exp\left(N\tau r(x_1) - \beta \sum_{n=1}^{N} D^{n-1}(x_1)\right) q(x_1).
\end{aligned}
\tag{152}
$$

$\square$

Despite the presence of the partition function $Z$, the induced data distribution $q_\theta^N(x_1)$ under softmax weighting ultimately depends on the unnormalized exponentiated rewards, similar to the exponential weighting case. The key difference lies in the normalization:

- Exponential Weighting: Does not normalize $w(x_1)$, leading to potential numerical instability.
- Softmax Weighting: Normalizes $w(x_1)$ at each epoch, ensuring $\sum_{x_1} w(x_1) = 1$.

However, when considering the evolution over multiple epochs, the normalization constants $Z^N$ accumulate but do not affect the proportionality of $q_\theta^N(x_1)$ with respect to $x_1$. And, in practice, softmax weighting normally requires a large batch size for efficient learning.

**Corollary 19** (Exploration Exploitation Trade-Off). *Proper tuning of $\tau$ and $\beta$ is essential to balance exploration (sampling diverse $x_1$) and exploitation (focusing on high-reward $x_1$):*

- *High $\tau$ : Increases emphasis on high-reward samples.*

- *High $\beta$ : Increases the penalty for deviations from the reference model.*

**Remark.** *In practice, the Boltzmann Weight Function can help stabilize the learning process and keep numerical stability. However, in practice, when the batch size is smaller, the Boltzmann Weight Function methods normally fail to learn a optimal results in similar training steps as Exponential weighting function methods.*

## C.9  RL PERSPECTIVES OF ORW-CFM-W2

In this section, we start to offer a different perspective of the learning behavior of models under ORW-CFM-W2 loss from the perspective of RL.

We first do the following parallels: 1) Data Points $x_1$ as Actions: In our setup, each data point $x_1$ corresponds to an action $a$ in RL. 2) Discrepancy Term as Regularizer: The discrepancy term $D^{n-1}(x_1)$ acts as a regularizer, similar to entropy regularization or KL divergence penalties in RL.

Then, we define the objective that our RL agents want to optimize:

**Definition C.3** (RL Objective of ORW-CFM).

$$\mathcal{J}(\theta) = \mathbb{E}_{x_1 \sim q_\theta^n(x_1)}[r(x_1)] - \frac{1}{\lambda} D_{\mathrm{KL}}\left(q_\theta^n(x_1) \| q_\theta^{n-1}(x_1)\right) \tag{153}$$

where $q_\theta^n(x_1)$ is the data distribution at epoch $n$. $q_\theta^{n-1}(x_1)$ is the data distribution at epoch $n-1$. $D_{\mathrm{KL}}$ acts as a regularizer to keep the new distribution close to the previous one. $\lambda$ controls the strength of the regularization.

We can obtain the following Theorem (For ease of reading, we rewrite the Theorem C.9 to be proved in this section as follows):

**Theorem.** *In RL settings, We aim to find $q_\theta^n(x_1)$ that maximizes $\mathcal{J}(\theta)$. This involves solving:*

$$q_\theta^n(x_1) = \arg\max_q \mathbb{E}_{x_1 \sim q}[r(x_1)] - \frac{1}{\lambda} D_{\mathrm{KL}}\left(q(x_1) \| q_\theta^{n-1}(x_1)\right) \tag{154}$$

*And the optimal data distribution (policy) $q_\theta^n(x_1)$ can be obtained as:*

$$q_\theta^n(x_1) \propto q_\theta^{n-1}(x_1) \exp\left(\lambda r(x_1)\right) \tag{155}$$

*Proof of Theorem C.9.* The optimization problem can be solved using variational methods. The KL divergence between $q(x_1)$ and $q_\theta^{n-1}(x_1)$ is:

$$D_{\mathrm{KL}}\left(q(x_1) \| q_\theta^{n-1}(x_1)\right) = \int q(x_1) \log \frac{q(x_1)}{q_\theta^{n-1}(x_1)} dx_1 \tag{156}$$

Introducing a Lagrangian to enforce the normalization constraint $\int q(x_1)\, dx_1 = 1$ :

$$\mathcal{L} = \int q(x_1)\left[r(x_1) - \frac{1}{\lambda}\log\frac{q(x_1)}{q_\theta^{n-1}(x_1)}\right] dx_1 - \eta\left(\int q(x_1)\, dx_1 - 1\right). \tag{157}$$

Then, we take the functional derivative of $\mathcal{L}$ with respect to $q(x_1)$ and set it to zero:

$$\frac{\delta \mathcal{L}}{\delta q(x_1)} = r(x_1) - \frac{1}{\lambda}\left(\log\frac{q(x_1)}{q_\theta^{n-1}(x_1)} + 1\right) - \eta = 0 \tag{158}$$

Simplifying:

$$r(x_1) - \frac{1}{\lambda}\log\frac{q(x_1)}{q_\theta^{n-1}(x_1)} - \left(\frac{1}{\lambda} + \eta\right) = 0 \tag{159}$$

Rewriting the equation:

$$\log\frac{q(x_1)}{q_\theta^{n-1}(x_1)} = \lambda\left(r(x_1) - C\right) \tag{160}$$

where $C = \frac{1}{\lambda} + \eta$ is a constant independent of $x_1$.

Exponentiating both sides:

$$\frac{q(x_1)}{q_\theta^{n-1}(x_1)} = \exp(\lambda r(x_1) - \lambda C) \tag{161}$$

Thus:

$$q_\theta^n(x_1) = q_\theta^{n-1}(x_1) \exp(\lambda r(x_1)) e^{-\lambda C} \tag{162}$$

Since $e^{-\lambda C}$ is a normalization constant, we have:

$$q_\theta^n(x_1) \propto q_\theta^{n-1}(x_1) \exp(\lambda r(x_1)) \tag{163}$$

Then, the proof concludes. □

Similarly, if we introduce the Regularization Term $D^{n-1}(x_1)$, we can have the following Theorem (For ease of reading, we rewrite the Theorem C.9 to be proved in this section as follows):

**Theorem.** *Considering an additional discrepancy term $D^{n-1}(x_1)$ that acts as a regularizer. We can include this term in the objective function:*

$$\mathcal{J}(\theta) = \mathbb{E}_{x_1 \sim q}\left[r(x_1) - \beta D^{n-1}(x_1)\right] - \frac{1}{\lambda} D_{\mathrm{KL}}\left(q(x_1) \| q_\theta^{n-1}(x_1)\right) \tag{164}$$

*Then, the RL agents aim to find $q_\theta^n(x_1)$ that maximizes $\mathcal{J}(\theta)$. This involves solving:*

$$q_\theta^n(x_1) = \arg\max_q \mathbb{E}_{x_1 \sim q}\left[r(x_1) - \beta D^{n-1}(x_1)\right] - \frac{1}{\lambda} D_{\mathrm{KL}}\left(q(x_1) \| q_\theta^{n-1}(x_1)\right) \tag{165}$$

*And the optimal data distribution (policy) $q_\theta^n(x_1)$ can be obtained as:*

$$q_\theta^n(x_1) \propto q_\theta^{n-1}(x_1) \exp\left(\lambda r(x_1) - \lambda\beta D^{n-1}(x_1)\right). \tag{166}$$

*Proof of Theorem C.9.* Following similar steps, the functional derivative becomes:

$$\frac{\delta\mathcal{L}}{\delta q(x_1)} = r(x_1) - \beta D^{n-1}(x_1) - \frac{1}{\lambda}\left(\log\frac{q(x_1)}{q_\theta^{n-1}(x_1)} + 1\right) - \eta = 0 \tag{167}$$

Simplifying:

$$\log\frac{q(x_1)}{q_\theta^{n-1}(x_1)} = \lambda\left(r(x_1) - \beta D^{n-1}(x_1) - C\right) \tag{168}$$

Exponentiating:

$$q_\theta^n(x_1) = q_\theta^{n-1}(x_1) \exp\left(\lambda r(x_1) - \lambda\beta D^{n-1}(x_1)\right) e^{-\lambda C} \tag{169}$$

Therefore, the updated data distribution is:

$$q_\theta^n(x_1) \propto q_\theta^{n-1}(x_1) \exp\left(\lambda r(x_1) - \lambda\beta D^{n-1}(x_1)\right) \tag{170}$$

Then, the proof concludes. □

**Corollary 20** (ORW-CFM-W2 As Reinforcement Learning)**.** *Based on Theorem C.9, we can rewrite the induced optimal policy in equation 164 as follows:*

$$q_\theta^n(x_1) \propto q_\theta^{n-1}(x_1) \exp\left(\lambda r(x_1) - \lambda\beta D^{n-1}(x_1)\right) \tag{171}$$

*By setting $\beta' = \lambda * \beta$, $w(x_1) = \exp(\lambda r(x_1))$, we can obtain:*

$$q_\theta^n(x_1) \propto w(x_1) q_\theta^{n-1}(x_1) \exp\left(-\beta' D^{n-1}(x_1)\right) \tag{172}$$

*which is the same form of induced data distribution over $x_1$ in Theorem 4.*

*Therefore, ORW-CFM with W2 Distance Regularization is actually solving a Reinforcement Learning Problem, where the objective is a trade-off between reward and discrepancy.*

The update rule for $q_\theta^n(x_1)$ mirrors the policy update in RL methods like AWR, where the new policy is proportional to the old policy multiplied by the exponentiated advantage.

The discrepancy $D^{n-1}(x_1)$ acts as a penalty term, discouraging the model from deviating too far from the **reference model**, similar to how KL divergence or entropy regularization is used in RL (Peng et al., 2019; Schulman et al., 2017) or RLHF (Black et al., 2024; Rafailov et al., 2023) to maintain policy stability and keep the fine-tuned model close to pre-trained model.

**Remark** (Connection to AWR). *AWR (Peng et al., 2019) also updates the policy by weighting actions according to the exponentiated advantage. Similarly, our update rule similarly weights data points $x_1$ according to $\exp\left(\lambda * r\left(x_1\right)\right)$.*

**Remark** (Connection to PPO). *PPO (Schulman et al., 2017) uses a clipped objective to prevent large policy updates. In our setup, the regularization term $D^{n-1}(x_1)$ prevents the data distribution from changing too drastically. Besides, $D^{n-1}(x_1)$ can also used to limit the distance between ref model and fine-tuned model.*

**Remark** (Exploration and Exploitation Trade-Off). *The term $\exp\left(\lambda * r\left(x_1\right)\right)$ encourages the model to prioritize data points with higher rewards. While the regularization term $\exp\left(-\beta' D^{n-1}\left(x_1\right)\right)$ ensures the model does not deviate excessively from the reference, promoting stability and diversity (e.g., avoid the policy collapses (Haarnoja et al., 2018))*

**Corollary 21** (Learning Flow Matching as Reinforcement Learning). *In general, by adopting a reinforcement learning perspective, we have derived the evolution of the data distribution $q_\theta^n(x_1)$ under self-training with the RW-CFM-W2 loss. The derivation shows that:*

$$q_\theta^n(x_1) \propto q_\theta^{n-1}(x_1) \exp\left(\lambda * r\left(x_1\right) - \beta' D^{n-1}\left(x_1\right)\right) \tag{173}$$

*This update rule mirrors policy updates in RL methods like Advantage Weighted Regression, where the new policy is updated based on the exponentiated advantage while incorporating a regularization term to ensure stability.*

# D EXPERIMENTAL DETAILS

## D.1 IMPLEMENTATION DETAILS

To ensure reproducibility, we built our code upon the open-source flow matching code base called TorchCFM (Tong et al., 2024a;b) to demonstrate that our method is general enough to be applied to fine-tuning existing flow matching based methods, which provides efficient and theoretical sound tools for training continuous normalizing flow models. TorchCFM offers a simulation-free training objective for Conditional Flow Matching (CFM), enabling conditional generative modeling and accelerating both training and inference processes.

We utilized the existing unconditional flow matching training baselines provided by TorchCFM on datasets such as MNIST and CIFAR-10. Our implementations of the RW-CFM, ORW-CFM, and ORW-CFM-W2 methods are based on the TorchCFM codebase, and we adopt the same network architecture—a U-Net (Ronneberger et al., 2015)—for the image generation tasks. The pre-trained models used in our experiments are trained with the standard CFM loss as defined in Equation equation 1 on CIFAR-10 and MNIST datasets.

For the image compression task, we directly adopt the reward function from DDPO (Black et al., 2024). In the even-number generation task, we trained a binary classifier to distinguish between even and odd digits, which serves as the reward signal. For the text-image alignment task, we employ the CLIP model from OpenAI (Radford et al., 2021a) to calculate the similarity between text and images; the CLIP model is pre-trained and provides a probability score representing the alignment between the generated image and the given text prompt.

The reward functions for each task can be found in App. F. And the hyper-parameters can be found in App. E.

## D.2 RESOURCES USED

All experiments were conducted on a system with one worker equipped with an 8-core CPU and, an NVIDIA A10 GPU, and memory of 32 GB. The MNIST task cost about 4-8 hours for training, while image compression and text-image alignment task costs about 2-3 days for training.

# E   HYPER-PARAMETERS

In general, in this paper, we propose a general and easy-to-use RL method for fine-tuning flow matching generative models. Our algorithm is directly built upon the flow matching training code in TorchCFM (Tong et al., 2024a). To facilitate reproducibility and highlight the effectiveness of our proposed RL fine-tuning algorithm, we maintain the same flow matching training parameters and the calculation of $u_t$ in OT paths as in the TorchCFM examples. We have conducted detailed ablation experiments in the main text and appendix on the two most important hyperparameters in our method: the entropy control parameter $\tau$ and the W2 distance regularization parameter $\alpha$.

ORW-CFM-W2 method involves only a few hyperparameters, and we have carefully ablated those that affect the convergence behavior in our experiments of Sec. 5, namely $\tau$ and $\alpha$. Specifically, $\tau$ controls the convergence speed of the fine-tuning method, while $\alpha$ has a significant impact on the convergence point.

Massive experiments demonstrate that the parameter $\tau$ effectively controls the convergence speed of the policy model. In the absence of W2 regularization, $\tau$ directly influences the rate at which the policy collapses to a delta distribution, as clearly demonstrated in the theoretical analysis of our exponential form weighting form as 5.

Interestingly, previous methods such as DDPO (Black et al., 2024) did not include KL regularization terms or other explicit distance constraints, making it difficult to control the divergence between the optimized policy and the pre-trained reference model. As a result, these methods often struggle to avoid over-optimization and policy collapse. Therefore, when employing our method, we strongly recommend setting $\alpha > 0$ to provide sufficient W2 regularization and prevent policy collapse.

**Remark.** *In practice, we encourage users/readers to first use a small $\alpha$ to fine-tune the model with our ORW-CFM-W2 method, along with a relatively large $\tau$. Then, different values of $\alpha$ can be tried to customize the optimal generative models that fit the user's needs for reward and diversity.*

For all experiments, we use a batch size of 64 and employ a separate test process to evaluate our performance.

In general, the proposed ORW-CFM-W2 method effectively fine-tunes flow matching models. We can control the convergence speed through $\tau$ and adjust the diversity of convergence points and the final reward through $\alpha$. This provides sufficient flexibility for customizing the convergence policy to meet user needs in various scenarios and enables us to achieve stable and efficient RL fine-tuning using a simple reward-weighted method.

# F    REWARD FUNCTIONS

In this section, we will describe the reward function we used in our experiments to demonstrate the effectiveness and generality of our method.

## F.1    TARGET IMAGE GENERATION

In the MNIST target image generation task, our objective is to fine-tune the model to generate images of even numbers. We used a pre-trained binary classifier $C(x)$, where $x$ is the generated image, to determine whether the image corresponds to an even or odd number. The classifier outputs the probability of the image being even or odd:

$$C_{\text{even}}(x) = p(\text{even} \mid x), \quad C_{\text{odd}}(x) = p(\text{odd} \mid x) \tag{174}$$

The reward function is then defined as the difference between the classifier's probabilities of the image being even versus odd:

$$r(x) = p(\text{even} \mid x) - p(\text{odd} \mid x) \tag{175}$$

Where $p(\text{even} \mid x) = C_{\text{even}}(x)$ is the probability that the generated image $x$ is classified as an even digit (i.e., digits $\{0, 2, 4, 6, 8\}$ ), and $p(\text{odd} \mid x) = C_{\text{odd}}(x)$ is the probability that the generated image $x$ is classified as an odd digit (i.e., digits $\{1, 3, 5, 7, 9\}$ ).

This reward encourages the model to maximize $p(\text{even} \mid x)$ while minimizing $p(\text{odd} \mid x)$, guiding the model to generate even digits more frequently. The fine-tuning objective can be formalized as:

$$\max_{\theta} \mathbb{E}_{x \sim q_{\theta}(x)}[r(x)] = \mathbb{E}_{x \sim q_{\theta}(x)} \left[ C_{\text{even}}(x) - C_{\text{odd}}(x) \right] \tag{176}$$

This reward function enables the model to be fine-tuned to produce even numbers without needing explicit labels or a dataset directly composed of even digits.

Furthermore, it is worth noting that for this type of binary classifier-guided reward, we can easily modify the task to generate the opposite class of images by simply using the negative of the reward function. For example, to generate odd digits, we define the reward as $r(x) = p(\text{odd} \mid x) - p(\text{even} \mid x)$. The results for generating odd digits are provided in App. G.2.

## F.2    IMAGE COMPRESSION

For the image compression task, we followed the reward function from DDPO (Black et al., 2024), where the goal was to either minimize the file size of the images after JPEG compression. The reward $r(x)$ is proportional to the compression rate. We controlled the balance between compression and diversity using the regularization parameter $\alpha$, which induces varying degrees of divergence between the fine-tuned model and the reference model. A lower $\alpha$ prioritizes compressibility, leading the model to generate images that occupy minimal storage space after compression, while higher $\alpha$ increases diversity in the generated outputs.

## F.3    TEXT-IMAGE ALIGNMENT USING CLIP

For the text-image alignment task, we utilized the CLIP model, which computes the similarity between text prompts and generated images. The reward function in this case was based on the probability score generated by the CLIP model. Given an image $x$ and a text description $t$, the CLIP model outputs a probability $p_{\text{clip}}(x, t)$, which indicates the alignment between the image and the text. The reward function for this task was defined as:

$$r(x, t) = p_{\text{clip}}(x, t) \tag{177}$$

This reward allows the model to fine-tune its output images to better match the provided text prompts, optimizing for higher similarity as measured by the CLIP model. In practice, we will use

a difference form in probability to promote training like $r(x, \text{"cat"}) = p_\text{clip}(x, \text{"an image of cat"}) - p_\text{clip}(x, \text{"not an image of cat"})$.

## G ADDITIONAL EXPERIMENTAL RESULTS

### G.1 ALIGNING SMALL-SCALE FLOW MATCHING MODELS

We first evaluate by guiding an unconditional generative model pre-trained on CIFAR-10 (Tong et al., 2024b) to maximize the text-image similarity score based on CLIP (Radford et al., 2021a), specifically aligning the generated images with a given text prompt via reward (e.g., "an image of cat"). As shown in Figure 11, without W2 regularization (i.e., $\alpha = 0$), the ORW-CFM method tends to fall into a greedy policy that maximizes reward but sacrifices diversity. However, with W2 regularization, our agent achieves a more diverse convergent optimal policy while maintaining similar final performance.

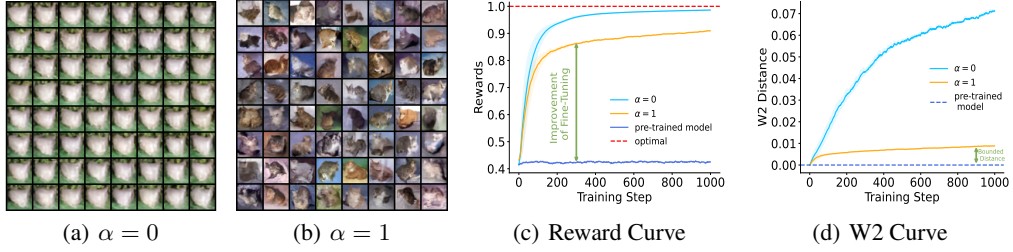

(a) $\alpha = 0$  (b) $\alpha = 1$  (c) Reward Curve  (d) W2 Curve

Figure 11: Diversity Control with different $\alpha$ while $\tau = 1$ and reward is Text-Image Similarity Probability Score (e.g., $p_{\text{clip}}(x, \text{"an image of cat"})$ for a cat image). W2 distance is estimated by its upper bound (see App. C.4). In practice, we will use a difference form in probability to promote training like $r(x, \text{"cat"}) = p_{\text{clip}}(x, \text{"an image of cat"}) - p_{\text{clip}}(x, \text{"not an image of cat"})$.

### G.2 REWARD SHAPING

To demonstrate the adaptability of our model to various reward objectives, we can directly modify the reward function to change the fine-tuning target. For instance, we transformed the original even-preference reward function, $r = p(even) - p(odd)$, into $r = p(odd) - p(even)$ to generate odd numbers instead. The results show that our method successfully fine-tunes the flow model according to the new reward objective, leading to the generation of only odd numbers.

We set $w(x)$ as $w(x) = \exp(\tau \cdot r(x))$ with $\tau = 1$ (except for the last one with $\tau = 0$ as control group), and experimented with different values of $\alpha$ to control the divergence between the fine-tuned model and the reference model, which induces diversity in the convergent fine-tuned model. As illustrated in Fig. G.2, without W2 regularization, our method converges to the optimal generative policy, though with limited diversity. However, as the W2 regularization increases with higher values of $\alpha$, our model still performs the odd-number generation task effectively, while also showing increased diversity in the generated samples.

### G.3 MORE TEXT-IMAGE ALIGNMENT RESULTS

In this section, we provide more text-image alignment results:

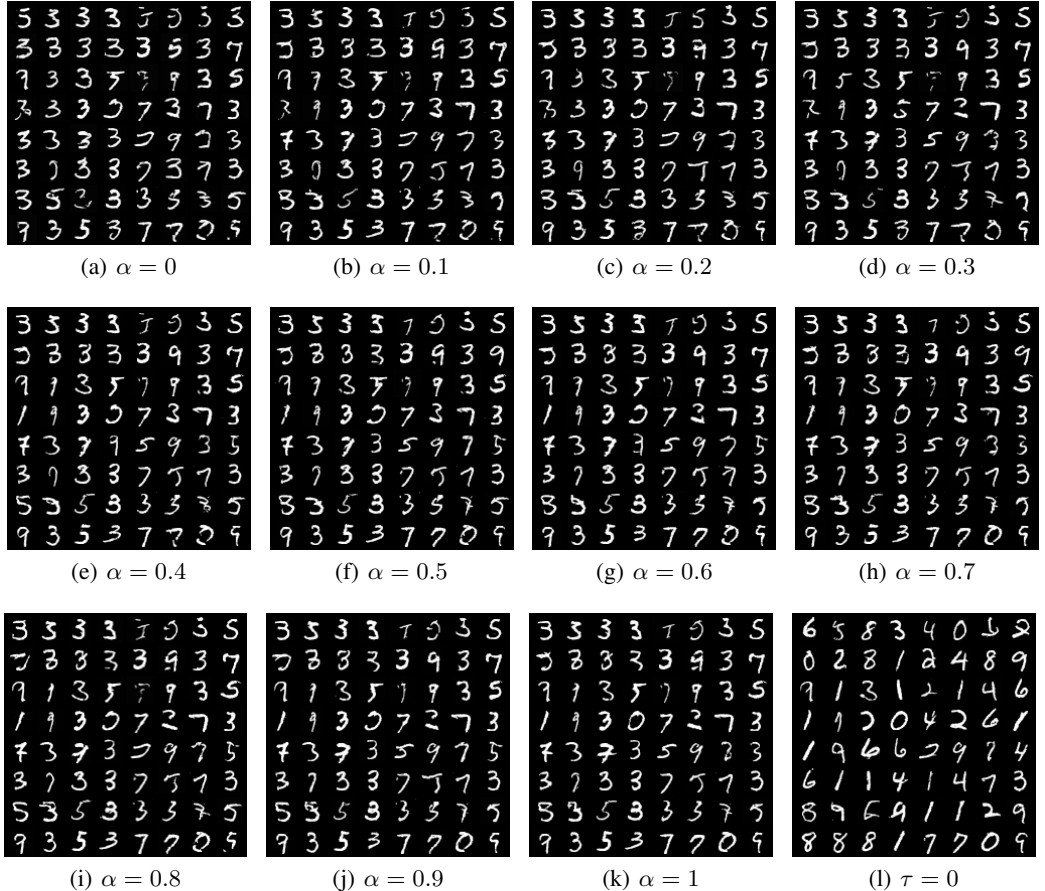

(a) $\alpha = 0$    (b) $\alpha = 0.1$    (c) $\alpha = 0.2$    (d) $\alpha = 0.3$

(e) $\alpha = 0.4$    (f) $\alpha = 0.5$    (g) $\alpha = 0.6$    (h) $\alpha = 0.7$

(i) $\alpha = 0.8$    (j) $\alpha = 0.9$    (k) $\alpha = 1$    (l) $\tau = 0$

Figure 12: Odd Number Generation in MNIST.

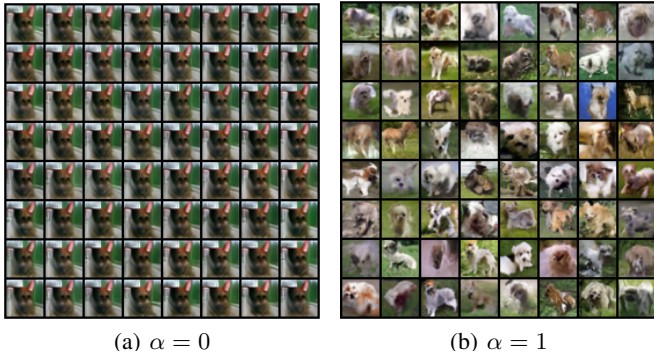

(a) $\alpha = 0$    (b) $\alpha = 1$

Figure 13: Diversity Control with different $\alpha$ while $\tau = 1$ and reward is Text-Image Similarity Probability Score (e.g., $p_{\text{clip}}(x, \text{"an image of dog"})$ for a dog image). In practice, we will use a difference form in probability to promote training like $r(x, \text{"dog"}) = p_{\text{clip}}(x, \text{"an image of dog"}) - p_{\text{clip}}(x, \text{"not an image of dog"})$.

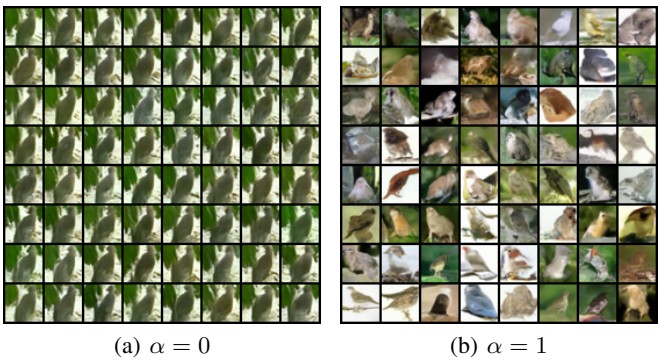

(a) $\alpha = 0$        (b) $\alpha = 1$

Figure 14: Diversity Control with different $\alpha$ while $\tau = 1$ and reward is Text-Image Similarity Probability Score (e.g., $p_{\text{clip}}(x, \text{"an image of bird"})$ for a bird image). In practice, we will use a difference form in probability to promote training like $r(x, \text{"bird"}) = p_{\text{clip}}(x, \text{"an image of bird"}) - p_{\text{clip}}(x, \text{"not an image of bird"})$.

# H  ALGORITHM PSEUDOCODE

In fact, we can implement the proposed fine-tuning algorithm by making minor changes based on the conditional flow matching implementation in OT-CFM (Tong et al., 2024a). In this section, we will demonstrate the Pseudocode of each fine-tuning algorithm proposed in this paper. For the origin CFM pipeline we refer the readers to (Tong et al., 2024a).

---

**Algorithm 1** Reward Weighted Conditional Flow Matching (RW-CFM in Theorem 1)

---

**Require:** Pre-trained model $v_{\theta_{\text{ref}}}$ and induced sample distribution $q_{\theta_{\text{ref}}}(x_1)$, efficiently samplable $p_t(x|x_1)$, computable $u_t(x|x_1)$, initial network $v_\theta$, reward function $r(x_1)$, weighting function $w(x_1) = \mathcal{F}(r(x_1))$ where $w(x_1) \propto r(x_1)$, entropy hyper-parameter $\tau$, $\mathcal{F}$ can be $\exp$ or $\text{Softmax}$

    $\theta \leftarrow \theta_{\text{ref}}$
    **while** Training **do**
        $x_1 \sim q_{\theta_{\text{ref}}}(x_1); \quad t \sim \mathcal{U}(0,1); \quad x \sim p_t(x|x_1)$
        $r \leftarrow r(x_1); \quad w = \mathcal{F}(\tau * r)$
        $\mathcal{L}_{\text{RW-CFM}}(\theta) \leftarrow$ equation 3
        $\theta \leftarrow \text{Update}(\theta, \nabla_\theta \mathcal{L}_{\text{RW-CFM}}(\theta))$
    **end while**
    **return** $v_\theta$

---

**Algorithm 2** Online Reward Weighted Conditional Flow Matching (ORW-CFM in Theorem 2)

---

**Require:** Parameterized efficiently samplable $q_\theta(x_1), p_t(x|x_1)$, computable $u_t(x|x_1)$, initial network $v_\theta$, reward function $r(x_1)$, weighting function $w(x_1) = \mathcal{F}(r(x_1))$ where $w(x_1) \propto r(x_1)$, entropy hyper-parameter $\tau$, $\mathcal{F}$ can be $\exp$ or $\text{Softmax}$, pre-trained model $v_{\theta_{\text{ref}}}$ and induced sample distribution $q_{\theta_{\text{ref}}}(x_1)$

    $\theta \leftarrow \theta_{\text{ref}}; \quad q_\theta(x_1) \leftarrow q_{\theta_{\text{ref}}}(x_1)$
    **while** Training **do**
        $x_1 \sim q_\theta(x_1); \quad t \sim \mathcal{U}(0,1); \quad x \sim p_t(x|x_1)$
        $r \leftarrow r(x_1); \quad w = \mathcal{F}(\tau * r)$
        $\mathcal{L}_{\text{ORW-CFM}}(\theta) \leftarrow$ equation 5
        $\theta \leftarrow \text{Update}(\theta, \nabla_\theta \mathcal{L}_{\text{ORW-CFM}}(\theta))$
    **end while**
    **return** $v_\theta$

---

**Algorithm 3** Online Reward Weighted Conditional Flow Matching with Wasserstein-2 Regularization (ORW-CFM-W2 in Theorem 4)

---

**Require:** Parameterized efficiently samplable $q_\theta(x_1), p_t(x|x_1)$, computable $u_t(x|x_1)$, initial network $v_\theta$, reward function $r(x_1)$, weighting function $w(x_1) = \mathcal{F}(r(x_1))$ where $w(x_1) \propto r(x_1)$, entropy hyper-parameter $\tau$ and W2 hyper-parameter $\alpha$, $\mathcal{F}$ can be $\exp$ or $\text{Softmax}$, pre-trained model $v_{\theta_{\text{ref}}}$ and induced sample distribution $q_{\theta_{\text{ref}}}(x_1)$

    $\theta \leftarrow \theta_{\text{ref}}; \quad q_\theta(x_1) \leftarrow q_{\theta_{\text{ref}}}(x_1)$
    **while** Training **do**
        $x_1 \sim q_\theta(x_1); \quad t \sim \mathcal{U}(0,1); \quad x \sim p_t(x|x_1)$
        $r \leftarrow r(x_1); \quad w = \mathcal{F}(\tau * r)$
        $\mathcal{L}_{\text{ORW-CFM-W2}}(\theta) \leftarrow$ equation 10
        $\theta \leftarrow \text{Update}(\theta, \nabla_\theta \mathcal{L}_{\text{ORW-CFM-W2}}(\theta))$
    **end while**
    **return** $v_\theta$

---

## I    DISCUSSION AND LIMITATION

Current RL fine-tuning methods, like DPO (Rafailov et al., 2023) or many reward-based RLHF methods for large language models (LLM) (Lee et al., 2023; Huguet et al., 2024; Gülçehre et al., 2023) normally rely on a well-collected or filtered offline dataset, requiring massive manual assistance and high costs, which is far from automated fine-tuning. However, traditional online fine-tuning methods are usually too complicated, with too many hyper-parameters, and it is difficult for us to find the relationship between each hyperparameter and the convergent policy, making the overoptimization problem hard to handle.

In fact, the existing LLM model and large text-image generative models has a very strong data sampling capability, but traditional online RL fine-tuning methods may easily cause the model to forget old knowledge, completely collapse to a local greedy policy, and fail to jump out of the local optimum. However, if we can control the collapse speed of the policy (via $\tau$) and the distance between the converged policy and the reference policy (via $\alpha$), then the online RL fine-tuning method can not only converge to a good and diverse policy without the need for additional manually labeled/filtered data, but also avoid over-optimization (Black et al., 2024).

In general, one of the primary motivations of our paper is to propose an easy-to-use and controllable reinforcement learning algorithm for fine-tuning continuous flow matching generative models toward arbitrary reward objectives. Although flow matching has demonstrated powerful generative capabilities in models like Stable Diffusion 3 (Esser et al., 2024, SD3), an easy-to-use and theoretically comprehensive RL fine-tuning method for continuous flow matching models has not been widely studied. Previous methods are either offline (Lee et al., 2023), rely on filtered datasets and cannot be applied to arbitrary reward models (Rafailov et al., 2023), or lack an explicit divergence bound between the learned policy and the reference model, which may be incapable of addressing over-optimization and policy collapse issues (Black et al., 2024). Furthermore, most methods have not been extensively applied or studied in continuous flow matching fine-tuning tasks. To address these limitations, we propose an online reward-weighted conditional flow matching with the Wasserstein-2 regularization (ORW-CFM-W2) method based on reward-weighted RL (Peters & Schaal, 2007; Peng et al., 2019). Our goal is to achieve a *controllable RL fine-tuning algorithm* that allows us to control both the convergence speed and the convergence point (e.g., managing the trade-off between reward and diversity, see Fig. 4).

To this end, we conducted a detailed analysis of the optimal policy distribution induced by our algorithm and experimentally verified the influence of the entropy coefficient $\tau$ and the regularization coefficient $\alpha$ on the convergence speed and convergence point.

Interestingly, we established the similarity between our method and traditional RL methods with KL regularization in policy updates. We found that $\tau$ effectively controls the speed (or range) of the policy updates, thereby affecting the convergence speed, and when the W2 regularization is absent, it influences the rate of policy collapse. The coefficient $\alpha$ directly affects the diversity of the final convergent policy by controlling the divergence from the reference model—assuming, of course, that the pre-trained model is not a purely greedy policy.

In practice, we recommend using the ORW-CFM-W2 method with at least $\alpha > 0$ during fine-tuning to effectively avoid issues such as policy collapse.

Additionally, since our method does not make any assumptions about the reward function, it can readily adapt to new tasks by simply adjusting the reward, **without the need to modify other hyper-parameters or collect new datasets manually**. For example, in the MNIST dataset, we can encourage the generation of images of odd digits by defining the reward as $-(p(\text{even}) - p(\text{odd}))$, thereby penalizing even digits and rewarding odd ones (See App. G.2). In other words, our method shifts the focus from fine-tuning tasks to designing more interesting reward functions, enabling the creation of various flow matching generative models that meet specific user needs without carefully designed offline dataset.

It is important to mention that there are still many unresolved issues in RL for flow matching fine-tuning tasks, such as data efficiency, more general reward shaping methods, off-policy online fine-tuning methods, and online fine-tuning of large text-image generative models like SD3 (Esser et al., 2024). These topics warrant further investigation in future work.

