# OpenReview forum: "Online Reward-Weighted Fine-Tuning of Flow Matching with Wasserstein Regularization"
_ICLR.cc/2025/Conference — ICLR 2025 Poster_

### Official Review · Reviewer_S526 · 2024-11-04

**Soundness:** 3
**Presentation:** 2
**Contribution:** 2
**Rating:** 6
**Confidence:** 4

**Summary:**

The paper introduces a new online reinforcement learning (RL) fine-tuning method for continuous flow-based generative models, named Online Reward-Weighted Conditional Flow Matching with Wasserstein-2 Regularization (ORW-CFM-W2). It addresses the challenges of policy collapse and high computational costs associated with traditional fine-tuning methods. The authors propose integrating RL within the flow matching framework, utilizing an online reward-weighting mechanism to focus on high-reward regions and a Wasserstein-2 distance regularization to balance exploration and exploitation. The paper provides theoretical analyses and empirical results across various tasks, demonstrating the effectiveness of the proposed method in achieving optimal policy convergence with controlled trade-offs between reward maximization and the generation capacity.

**Strengths:**

1.The paper effectively identifies and addresses some key issues in fine-tuning continuous flow-based generative models, such as policy collapse and computational inefficiency.
2. The introduction of the online reward-weighting mechanism and Wasserstein-2 distance regularization is well-suited for flow matching models that balances exploration and exploitation, mitigating the policy collapse problem.
3. The theoretical analyses are rigorous and provide a solid foundation for the proposed method. The empirical results across various tasks are convincing and demonstrate the method's effectiveness.

**Weaknesses:**

Potential Overemphasis on Theoretical Analysis:
While the theoretical underpinnings are robust, the paper might overly focus on the theoretical aspects at the expense of practical considerations. Balancing the presentation (e.g.section 4.6 to the appendix) to include more  case studies could make the findings more relatable to a broader audience.
Lack of Comparative Analysis with Other Regularization Techniques:
The paper introduces W_2 distance regularization but does not compare its effectiveness with other potential regularization methods. Including such comparisons could strengthen the paper's contribution by positioning it within the broader landscape of regularization strategies.
Narrow Empirical Validation:
The empirical validation is commendable, but the paper could benefit from testing the method across a wider range of datasets (e.g. CeleA face dataset) and tasks to further establish the generalizability and robustness of the approach.

**Questions:**

1. What kind of reward functions can be fine-tuned without collapsing by the proposed W_2 regularization method?
2. Is this method capable of performing fine-grained fine-tuning tasks, such as controlling specific semantic parts of images?
3. Why not use W_1 distance for regularizing?

---

> ### Author Response · Authors · 2024-11-20
> **Reply to Reviewer S526 [Part 1]**
>
> Thank you for your detailed and constructive review. We are grateful for your recognition of our work's theoretical rigor and its contributions to addressing key challenges in flow matching fine-tuning, particularly regarding policy collapse and computational efficiency. We’re happy to address your concerns with the following:
>
> > Q: Is this method capable of performing fine-grained fine-tuning tasks, such as controlling specific semantic parts of images? Include more case studies could make the findings more relatable to a broader audience.
>
> A: Thank you for the great suggestion. We comprehensively demonstrate our ORW-CFM-W2 method's strong capabilities in fine-grained semantic control through extensive experiments with Stable Diffusion 3 (SD3) [6].
>
> As shown in Section 5.3, Figure 5 shows precise control over complex spatial relationships between objects like `"a banana on the left of an apple" and "a cat on the left of a dog"`, achieving better positional accuracy and image quality compared to baselines like RAFT [1] and ReFT [2]. The ablation studies in Figure 6 using the challenging `"a cat in the sky"` prompt demonstrate that while methods without W2 regularization exhibit clear signs of policy collapse (generating nearly identical images), our approach with W2 regularization successfully maintains generation diversity while achieving high semantic accuracy.
>
> In our more comprehensive SD3 experiments (Appendix A), our method's fine-grained control capabilities are further validated through diverse reward architectures. Figure 9 demonstrates consistent performance across multiple text-image alignment reward models (`HPS-V2, Pick Score, and Alpha CLIP`) for the challenging "train on top of a surfboard" prompt, showing robust spatial understanding while maintaining natural variations in train appearances and ocean conditions. Most notably, Figure 10 exhibits sophisticated semantic control with complex compositional prompts like `"A stack of 3 cubes. A red cube is on the top, sitting on a red cube. The red cube is in the middle, sitting on a green cube. The green cube is on the bottom."` with specific color and position requirements, where our method successfully controls multiple attributes simultaneously while maintaining image quality and stylistic diversity. These comprehensive results validate our theoretical framework's ability to enable precise semantic control while preventing the policy collapse issues that typically challenge online fine-tuning methods [1] [2].
>
> > Q: What kind of reward functions can be fine-tuned without collapsing by the proposed W2 regularization method?
>
> A: Our method is designed to `fine-tune flow matching models with arbitrary reward functions` without requiring gradients of rewards [5] or filtered datasets [3], as comprehensively demonstrated through our experiments across different scales. In the experiments section, we showcase this versatility through controlled convergence using varying $\tau$ and $\alpha$ for classifier-guided digit generation (Figures 2-3), exhibiting precise control over digit categories while preserving diversity. Figure 4 validates our method with compression-based rewards for image optimization, where we demonstrate clear reward-distance trade-offs while maintaining generation quality. Figure 5 shows successful optimization with CLIP-based similarity rewards for text-image alignment, demonstrating superior performance compared to previous methods like RAFT [1] and ReFT [2]. Figure 6 further demonstrates through ablation studies that methods incorporating W2 regularization (including RAFT+W2, ReFT+W2, and ours) successfully prevent policy collapse while maintaining performance.
>
> In our SD3 experiments (See Appendix A), we further demonstrate this reward-agnostic property across diverse reward architectures. Figure 9 validates our method's adaptability across different reward architectures including `HPS-V2, Pick Score, and Alpha CLIP`. Figure 10 exhibits success with complex compositional tasks using CLIP rewards, where we effectively optimize multiple attributes (colors, positions, objects) while maintaining semantic coherence. The W2 regularization prevents collapse across all these reward functions through the tractable bound derived in Theorem 3, as particularly evident in our results on challenging tasks like spatial understanding (Figure 7), preventing policy collapse (Figure 8), multi-scale rewards (Figure 9) and multi-attribute control (Figure 10), where the method maintains both high rewards and generation diversity, when KL-divergence and W1-divergence are prohibited due to constraints mentioned above.

---

> ### Author Response · Authors · 2024-11-20
> **Reply to Reviewer S526 [Part 2]**
>
> > Q: Why choose W2 instead of other distance for regularizing (e.g., KL, W1)?
>
> A: Our choice of W2 distance is both theoretically motivated and practically necessary to address policy collapse in online fine-tuning of flow matching models. Finding a  computationally tractable divergence regularization in ODE-based flow matching models is non-trivial.  While KL divergence is widely used  for fine-tuning diffusion models and LLMs [3, 4], calculating KL divergence for flow matching models is infeasible, as it  requires solving intricate transport dynamics and tracking probability flows across continuous time, which is computationally costly (i.e., `exact likelihood computation requires expensive ODE simulation with Hutchinson traces estimator`, detailed in Appendix B.2.4). Unlike diffusion models that get around  this with variational bounds, there is no established relationship between vector field loss and ELBO in continuous-time ODE-based flow matching. Similarly, W1 distance faces intractability issues as the true marginal vector field is intractable  (i.e., `Theorem 2 of [7] does not hold for W1`).
>
> To address these challenges, we are the first to derive a computationally tractable upper bound for W2 distance in flow matching (Theorem 3). This bound only requires calculating the difference between vector fields, avoiding expensive likelihood calculations needed for KL divergence, thus enabling effective constraint of the discrepancy between fine-tuned and reference models while maintaining computational efficiency. Our theoretical analysis shows that this bound effectively prevents policy collapse while preserving the generation capacity of the pre-trained model.
>
> Our experimental results strongly validate this choice, particularly in our comprehensive experiments with SD3 in Appendix A. Figure 8 provides detailed ablation studies demonstrating that our W2-regularized approach successfully prevents policy collapse while maintaining semantic accuracy for challenging prompts like "a cat in the sky", where baseline methods without regularization generate nearly identical images. The effectiveness of W2 regularization is further evidenced in Figure 7 for spatial relationship control and Figure 10 for complex compositional prompts, where our method achieves precise semantic control while maintaining natural variations in generation. These results demonstrate that our theoretically-derived W2 bound provides a practical and effective solution for flow matching fine-tuning that enables both high performance and stable learning.

---

> ### Author Response · Authors · 2024-11-20
> **Reply to Reviewer S526 [Part 3]**
>
> > Q: Balancing the presentation (e.g. section 4.6 to the appendix) to include more case studies could make the findings more relatable to a broader audience.
>
> A: Following your kind suggestion, we have restructured the paper to improve accessibility while maintaining technical rigor. We have `moved the "RL Perspective of Online Fine-Tuning Method" (previously Section 4.6) to the appendix`, allowing us to dedicate more space in the main text to empirical evaluations that demonstrate the practical implications of our work. Specifically, we have significantly `expanded Section 5.3 and Appendix A  with comprehensive case studies on online fine-tuning of Stable Diffusion 3 (SD3)`, where Figures 5, 7 demonstrate superior performance in spatial relationship control compared to baselines like RAFT [1] and ReFT [2], Figures 6, 8 provide detailed ablation studies of policy collapse prevention, Figure 9 validates our method's adaptability across different reward architectures (HPS-V2, Pick Score, and Alpha CLIP), and Figure 10 showcases complex semantic control capabilities.
>
> Our expanded experimental section demonstrates that our method not only has strong theoretical foundations (as shown by Theorem 3, 5) but also provides practical solutions to real challenges in fine-tuning large generative models. The addition of diverse case studies, from spatial understanding to multi-attribute control, makes our findings more accessible to readers while complementing our theoretical contributions. These comprehensive results, particularly with SD3, help readers better understand both the theoretical innovations and practical benefits of our approach compared to existing methods [1, 2].
>
> ## References
>
> [1] Dong, Hanze, et al. "Raft: Reward ranked finetuning for generative foundation model alignment." arXiv preprint arXiv:2304.06767 (2023).
>
> [2] Huguet, Guillaume, et al. "Sequence-Augmented SE (3)-Flow Matching For Conditional Protein Backbone Generation." arXiv preprint arXiv:2405.20313 (2024).
>
> [3] Rafailov, Rafael, et al. "Direct preference optimization: Your language model is secretly a reward model." Advances in Neural Information Processing Systems 36 (2024).
>
> [4]  Fan, Ying, et al. "DPOK: Reinforcement Learning for Fine-tuning Text-to-Image Diffusion Models." Advances in Neural Information Processing Systems 36 (2024).
>
> [5] Domingo-Enrich, Carles, et al. "Adjoint matching: Fine-tuning flow and diffusion generative models with memoryless stochastic optimal control." arXiv preprint arXiv:2409.08861 (2024).
>
> [6] Esser, Patrick, et al. "Scaling Rectified Flow Transformers for High-Resolution Image Synthesis, March 2024." URL http://arxiv. org/abs/2403.03206.
>
> [7] Lipman, Yaron, et al. "Flow matching for generative modeling." *arXiv preprint arXiv:2210.02747* (2022).

---

> > ### Author Response · Authors · 2024-11-22
> >
> > Thank you for your thoughtful and constructive feedback. We hope that our responses have thoroughly addressed your concerns about balancing theoretical analysis with practical considerations, particularly through demonstrating our method's reward-agnostic nature and fine-grained control capabilities in our comprehensive SD3 experiments. We truly appreciate your detailed review, which has helped us achieve a better balance between theoretical foundations and practical applications. We always welcome any further discussions to enhance the accessibility and clarity of our work.

---

> > > ### Comment · Reviewer_S526 · 2024-11-25
> > >
> > > Thank you for the detailed explanation and clarification, as well as the effort in restructuring the paper. I think this paper is definitely valuable and I'll keep my positive score.

---

> > > > ### Author Response · Authors · 2024-11-25
> > > >
> > > > Dear Reviewer S526,
> > > >
> > > > Thank you for your positive recognition of our paper as "definitely valuable." We are truly grateful for your thoughtful review process and constructive feedback throughout our discussion. Your suggestions have been instrumental in helping us achieve a better balance between theoretical depth and practical applications, particularly through the restructured content and expanded empirical case studies on SD3. We're pleased that our detailed explanations and clarifications have effectively addressed your concerns while earning your continued positive assessment of our work.

---

### Official Review · Reviewer_tfNW · 2024-11-04

**Soundness:** 3
**Presentation:** 3
**Contribution:** 3
**Rating:** 6
**Confidence:** 4

**Summary:**

The paper presents a method to perform reward finetuning of flow-based models. The idea starts with the reward-weighted version of the standard flow matching loss (i.e., doing simple importance sampling) and, to remove the dependency on pretraining datasets and to perform online training, changes the sampling distribution from the original data distribution to the finetuned sampling policy. Such a strategy proves very prone to overfitting, as the finetuned distribution collapses into a single mode if it is trained for too many epochs. Therefore, the authors further proposes to regularize the sampling policy to be not too far away from the pretrained one (using a Wasserstein distance). The paper discusses some theoretical results like the asymptotic behaviors of the proposed methods and empirically show that the proposed method can be applied to finetuning of flow matching models pretrained on MNIST and CIFAR-10.

**Strengths:**

The paper theoretically analyzes probably one of the most intuitive methods of reward reweighting, and by introducing a regularization loss on the finetuned distribution, shows that this naive method can be extended to the online setting. To support the claims, the paper does sufficient amount of experiments on different small-scale image datasets and different reward functions. Especially, the paper shows that their online variant is better than the offline one.

Compared to baselines like DDPO that requires one specify the number of sampling steps for finetuning, the proposed method finetunes a model in the way very similar to flow matching -- to sample images from a "data" distribution and some random t in [0,1], and to compute the matching loss.

**Weaknesses:**

The paper does not compare the proposed method against any other methods, for instance DDPO (running PPO for diffusion finetuning). While one may argue that DDPO is not designed for continuous flow models, one eventually samples from CNFs with some discretization and can therefore construct an MDP for DDPO finetuning, and not to mention some more recent methods. On flow matching, there is a very recent paper [1] that does reward finetuning for flow matching (though this paper should be considered as a concurrent one). There also exist some more recent papers for reward finetuning to compare with, and I feel that showing at least one of them will be great.

The proposed method seems to be a bit sensitive (in theory) to hyperparameter tuning due to its online nature. It is a bit unsatisfactory that the resulted distribution (Eqn 12 in the paper) is dependent on the number of epochs. While in practice it is not a super big concern, an objective that guarantees convergence to a specific distribution (e.g. P_pretrained(x) * exp(lambda * r(x)) / Z) is generally considered better.

Many of the baselines are tested on large-scale models like StableDiffusion, and many of them can converge in a reasonably fast speed on simple reward functions like Aesthetic Score used in DDPO. The paper fails to show results in these more realistic settings (though it probably requires some compute, but one might be able to find a smaller model to do experiments).

[1] Adjoint Matching: Fine-tuning Flow and Diffusion Generative Models with Memoryless Stochastic Optimal Control
. Carles Domingo-Enrich, Michal Drozdzal, Brian Karrer, Ricky T. Q. Chen. https://arxiv.org/abs/2409.08861

**Questions:**

Besides the points raised in the weakness section:

1. It is probably better to also show quantitative metrics like diversity scores (e.g., feature pairwise distances) and FID scores.
2. In Eqn 10, it is probably more aesthetic to write \theta_\text{ft} and \theta_\text{ref} (for the subscripts), instead of \theta_{ft} and \theta_{ref}.
3. W2 distance is great, but I wonder if it makes a big difference if one instead uses KL divergence (both theoretically and empirically).

---

> ### Author Response · Authors · 2024-11-20
> **Reply to Reviewer tfNW [Part 1]**
>
> We deeply appreciate your thorough evaluation of our paper and the recognition of our theoretical contributions and empirical effectiveness. Thanks for noting that our method preserves the simplicity of the original flow-matching framework, making it easy to use while ensuring theoretical soundness. We are pleased to address each of your concern in detail:
>
> > Q: Why not compare with methods like DDPO and other recent baselines?
>
> A: Thank you for the suggestion and we now provide additional results on text-to-image alignment tasks of Stable Diffusion3 (SD3) with comparison to more recent baselines. Meanwhile, we'd like to clarify several important perspectives regarding choice of baselines.
>
> **Technical Limitations of Existing RL Fine-tuning Methods for Flow Matching:**
>
> Most existing RL fine-tuning methods cannot be directly applied to fine-tune continuous-time ODE-based flow matching models due to intractable calculation of ELBO, KL divergence and expensive calculation of likelihood with Hutchinson trace estimator. Methods like DDPO [4] and DPOK [7] rely heavily on computationally tractable MDP trajectory likelihood calculations and ELBO approximations, which are computationally intractable or ill-defined for ODE-based flow matching models (detailed in Appendix B.2.4). Besides, the continuous-time nature of flow matching invalidates discrete timestep-based policy updates used in DDPO [4]. While discretizing ODE is possible, derivation of corresponding MDP trajectory likelihood (i.e., $\log p_\theta\left(\mathbf{x}_{t-1} \mid \mathbf{x}_t, \mathbf{c}\right)$[4]) in continuous-normalizing-flow is non-trivial and itself could lead to a new research work that deviates significantly from what's proposed in DDPO [4]. Additionally, KL divergence [7] is also intractable in flow matching, necessitating our novel and tractable W2 regularization approach.
>
> Regarding the concurrent work of Adjoint Matching [5], while theoretically interesting, it has certain practical limitations - `it requires differentiable rewards`, lacks theoretical convergence analysis, hasn't been open-sourced, and hasn't demonstrated effectiveness on SOTA FM architectures like SD3 [6]. It also employs a complex optimization procedure with stochastic optimal control. In contrast, `our method works with arbitrary rewards, and preserves the simplicity of original CFM training loss, while providing both theoretical guarantees and strong empirical results on SD3.`
>
> **Comprehensive Evaluation with Applicable Baselines:**
>
> Given these technical constraints, we focused our comparisons on reward-based methods that can be adapted to flow matching - specifically RAFT [1] and ReFT [2], which don't rely on likelihood calculations. Though extending these methods [1] [2] to flow matching models was far beyond the empirical scope/contributions of their original work, we implemented them for SD3 fine-tuning to provide fair and meaningful comparisons. `To the best of our knowledge, we are the first to demonstrate successful online fine-tuning of SD3 models`. Our comprehensive experiments show our method's advantages across multiple dimensions: superior handling of spatial relationships (Figure 5), strong adaptability across different reward models including HPS-V2, Pick Score, and Alpha CLIP (Figure 9), and successful management of complex semantic relationships (Figure 10). `We also provide ablation results in Figure 6 to demonstrate unique contribution of ORW and W2 regularization.`

---

> ### Author Response · Authors · 2024-11-20
> **Reply to Reviewer tfNW [Part 2]**
>
> > Q: Why use W2 distance instead of KL divergence?
>
> A: Our choice of W2 distance is both theoretically motivated and practically necessary for fine-tuning flow matching models. As proven in Lemma 1, online reward-weighted fine-tuning without regularization will inevitably collapse to a delta distribution centered at maximum reward, causing policy collapse. This necessitates effective regularization to maintain diversity.
>
> However, finding a suitable and computationally tractable divergence regularization in ODE-based CFM is non-trivial. While KL divergence is widely used in previous works for fine-tuning diffusion models and LLMs [3, 7], calculating KL divergence for flow matching models requires solving intricate transport dynamics and tracking probability flows across continuous time, which is `computationally intractable` (detailed in Appendix B.2.4). Unlike diffusion models which can leverage variational bounds [4], continuous flow matching lacks tractable ELBO formulations, making KL-based approaches computationally infeasible.
>
> We address this challenge by `deriving a computationally tractable upper bound for W2 distance in flow matching (Theorem 3)`. To the best of our knowledge, we are the first to derive such a tractable W2 bound for online fine-tuning of flow matching models. Our experimental results strongly validate this choice - Figure 4 demonstrates how W2 regularization enables controlled divergence from the reference model while maintaining diversity. This is particularly evident in our extensive experiments with SD3 [6], where our method achieves better spatial relationship control (Figure 5) compared to RAFT [1] and ReFT [2]. The ablation studies in Figure 6 show that `all methods without W2 regularization exhibit clear signs of policy collapse`, generating nearly identical images. Additional experiments in Appendix A further demonstrate our method's effectiveness across different reward architectures (Figure 9) and complex compositional prompts (Figure 10), consistently maintaining both high-quality generation and semantic accuracy while preventing policy collapse.
>
> > Q: Regarding the dependency on number of epochs in the resulted distribution (Equation 12).
>
> A: Equation 12 establishes a general mathematical framework for analyzing the convergent behavior of our proposed ORW-CFM-W2 method for different cases, and its limiting case depends on the configuration of key parameters ($\tau, \alpha$). To the best of our knowledge, we are the first to demonstrate the theoretical analysis of convergent behavior of online reward-weighted fine-tuning methods for CFM, applicable to both cases with W2 (Theorem 5)  and without W2 regularization  (Theorem 2 and Case 2 of Theorem 5) . Through both theoretical analysis (Theorem 5) and comprehensive experiments, `we have demonstrated two important limiting cases, where the learned policy will converge to  a specific distribution independent of the number of epochs`:
>
> 1. When $\tau=0$ or $\alpha \to \infty$ (Case 1 in Theorem 5), the model maintains behavior similar to the reference model, as validated by Figure 2  ($\tau=0$);
> 2. Conversely, when $\alpha=0$ or $\tau \to \infty$  (Case 2 in Theorem 5), the distribution collapses to a delta distribution maximizing rewards, as shown in Figure 3 ($\alpha=0$) and Figure 5, 6 (without W2).
>
> We further provide  another interesting case in appendix (Case 6 in App. C.7) and prove the existence of the best trade-off parameter for balancing reward optimization term and divergence. Although not all limiting cases have closed-form solution, the practical significance of our theoretical framework   is validated through experiments where the tradeoffs can be visually confirmed (Figures 2-10).

---

> ### Author Response · Authors · 2024-11-20
> **Reply to Reviewer tfNW [Part 3]**
>
> > Q: Results on larger-scale models like StableDiffusion.
>
> A: Thank you for your suggestions. `We have now added extensive experiments on Stable Diffusion 3 (SD3) [6] in Section 5.3 and Appendix A` to demonstrate our method's effectiveness on large-scale FM models. Our experiments show that ORW-CFM-W2 successfully fine-tunes SD3 on challenging spatial relationship prompts (Figure 5), achieving better positional relationship control and semantic understanding compared to RAFT [1], ReFT [2], and baseline SD3 [6] while maintaining image quality. The ablation studies in Figure 6 validate our theoretical predictions, demonstrating that W2 regularization effectively prevents policy collapse in large models - all methods without W2 regularization exhibit clear signs of policy collapse, generating nearly identical images. Even without W2 regularization, our base ORW-CFM method outperforms both RAFT and ReFT.
>
> Additionally, our comprehensive experiments in Appendix A showcase the method's versatility and robustness. We demonstrate strong adaptability across different reward architectures (Figure 9) including HPS-V2, Pick Score, and Alpha Clip. The method also successfully handles complex compositional prompts (Figure 10) with multiple interleaved requirements spanning spatial relationships, color specifications, and object attributes. These results demonstrate that our theoretically-grounded approach not only scales effectively to large models but also enables stable fine-tuning while maintaining both generation quality and semantic accuracy.
>
> > Q: Notation suggestion for equation 10: "In Eqn 10, it is probably more aesthetic to write \theta_\text{ft} and \theta_\text{ref} (for the subscripts), instead of \theta_{ft} and \theta_{ref}."
>
> A: Thank you for this suggestion. We agree that using $\theta_{\text{ft}}$ and $\theta_{\text{ref}}$ would improve clarity and aesthetics. We have updated this in our paper.
>
> ## References
>
> [1] Dong, Hanze, et al. "Raft: Reward ranked finetuning for generative foundation model alignment." arXiv preprint arXiv:2304.06767 (2023).
>
> [2] Huguet, Guillaume, et al. "Sequence-Augmented SE (3)-Flow Matching For Conditional Protein Backbone Generation." arXiv preprint arXiv:2405.20313 (2024).
>
> [3] Rafailov, Rafael, et al. "Direct preference optimization: Your language model is secretly a reward model." Advances in Neural Information Processing Systems 36 (2024).
>
> [4] Black, Kevin, et al. "Training diffusion models with reinforcement learning." arXiv preprint arXiv:2305.13301 (2023).
>
> [5] Domingo-Enrich, Carles, et al. "Adjoint matching: Fine-tuning flow and diffusion generative models with memoryless stochastic optimal control." arXiv preprint arXiv:2409.08861 (2024).
>
> [6] Esser, Patrick, et al. "Scaling Rectified Flow Transformers for High-Resolution Image Synthesis, March 2024." URL http://arxiv. org/abs/2403.03206.
>
> [7] Fan, Ying, et al. "DPOK: Reinforcement Learning for Fine-tuning Text-to-Image Diffusion Models." Advances in Neural Information Processing Systems 36 (2024).

---

> > ### Author Response · Authors · 2024-11-22
> >
> > Thank you for your highly constructive feedback and thorough assessment. We particularly appreciate your recognition of our theoretical analysis of reward reweighting and the observation that our method preserves the simplicity of flow matching training. We believe we have carefully addressed all concerns through: new large-scale SD3 experiments with RAFT and ReFT baselines, detailed theoretical analysis of convergence behavior and limiting cases, and comprehensive justification for using W2 distance over KL divergence in flow matching. The revision has significantly strengthened both our theoretical foundations and practical validation on realistic tasks. We always welcome any further suggestions for improvement.

---

> > > ### Comment · Reviewer_tfNW · 2024-11-23
> > >
> > > I appreciate the authors' response. Though I now feel slightly more positive, I do have some lingering concerns:
> > >
> > > 1. **Quantitative results on large-scale experiments.** It would be more convincing to show the average reward, diversity scores (as done in the adjoint matching paper) plus FID scores, beyond just showing qualitative examples.
> > >
> > > 2. **Dependency on epochs**. I fully understand that the authors prove the limiting cases. Yet, in practice nobody sets the parameters to some extreme values. The dependency on the number of training epochs means that one is not guaranteed to obtain an unbiased distribution that matches the target one, and for harder cases one may need to carefully tune the hyperparameters. Such concerns may be alleviated if some bound on the approximation error given different hyperparameters in a normal range can be proved, or if the authors may find some other way to show that it is not a big issue.
> > >
> > > 3. **The "ad hoc" issue**. As pointed out by Reviewer dXDz, the use of W2 distance is ad-hoc, and indeed the authors show that W2 distance works well even if we use ReFL, a less theoretically grounded method. Theoretically, if we do constrained optimization with bounded W2 distance, we may even show a bound for ReFL. That's said, the fact that the W2 regularization is a bit ad-hoc does not greatly compromise the contribution of this paper. But I would like to suggest that the authors lower their tone, instead of claiming that W2 is something very tied to the proposed online reward matching method.
> > >
> > > On the baseline issue, I agree with the authors that existing methods like DDPO are not very suitable for continuous flow models, but I am just curious how a naive baseline with an approximate MDP (by manually setting some noise schedule and do DDPM-like sampling) behaves. The authors may consider include something about it for their camera-ready version if their paper gets accepted.

---

> ### Author Response · Authors · 2024-11-24
>
> Thank you very much for your kind response and your positive feedback. We’re happy to address your lingering concerns with the following:
>
> > **Q: Quantitative results on large-scale experiments.**
>
> A: Thank you for your suggestion. We now include `Table 1` (also put it below for ease of reading) to `quantitatively evaluate our method's performance and effectiveness`. Our evaluation focuses on two key metrics: CLIP scores to measure reward optimization (higher indicates better text-image alignment) and diversity scores computed as the mean pairwise distance between CLIP embeddings of generated samples (higher indicates more diverse outputs). We use Euclidean distance to calculate the distance between any two embeddings when calculating mean pairwise distance. As shown in Table 1, even without W2 regularization, our ORW-CFM method significantly outperforms baseline methods RAFT [1] and ReFT [2] in both CLIP scores and diversity metrics. The addition of W2 regularization ('+W2' variants) further helps preserve generation diversity across all methods, with our full approach (ORW-CFM-W2) achieving the best alignment score while  maintain diversity comparable to the base SD3 model.
>
> Regarding FID scores, we note that FID is usually used to measure fitness to a given training dataset distribution, and it is only applicable when evaluating models whose training set is available. However, SD3 does not disclose its training data and we are online fine-tuning method `without ground-truth datasets`, and therefore FID may not be a proper metric to use here (note that `FID was not evaluated in previous online fine-tuning works like DPOK [7] /DDPO [4] or concurrent work like Adjoint Matching [5] either`).
>
>
>
> | Method                    | CLIP Score | Diversity Score |
> | :------------------------ | :--------- | :-------------- |
> | SD3 (Base Model)          | 28.69      | **4.78**           |
> | **ORW-CFM (Ours w/o W2)** | **33.63**  | **3.47**        |
> | RAFT                      | 29.30      | 2.05            |
> | ReFT                      | 29.32      | 3.26            |
> | **ORW-CFM-W2 (Ours)**     | **35.46**  | **4.21**        |
> | RAFT+W2                   | 30.88      | 2.81            |
> | ReFT+W2                   | 32.03      | 3.63            |
>
> > **Q: About dependency on number of epochs and hyperparameter tuning.**
>
> A: Thank you for this insightful observation. Besides the theoretically analyzed limiting cases, equation (12) in Theorem 5 provides intuitive guidance for practical parameter selection, and `we provide extensive experimental demonstrations of the effective controlling behavior for non-extreme value cases`. Our analysis clearly shows how $\tau$ and $\alpha$  intuitively control the convergent behavior: $\tau$ determines the algorithm's preference for reward maximization (as shown in Figure 2's impact on policy collapse and reward optimization), while $\alpha$ maintains diversity and prevents collapse (demonstrated in Figures 3-4's reward-diversity trade-off). This understanding allows practitioners to adjust parameters purposefully rather than through random guesses.
>
> Though deriving convergent distributions for all general cases is challenging, our empirical results demonstrate that our method achieves stable convergence and controllable fine-tuning across moderate hyper-parameter values. As shown in Figures 2-4, our method trains reliably to convergence without requiring early stopping, and practitioners can customize/adjust the convergent policy by adjusting $\tau$ and $\alpha$ according to their needs - increasing  $\tau$ for stronger reward maximization or $\alpha$ for greater diversity. The convergence behavior is validated by extensive experiments across different tasks, from target image generation to text-image alignment, showing consistent and predictable behavior that follows our theoretical intuition.

---

> ### Author Response · Authors · 2024-11-24
>
> > Q:  **The "ad hoc" issue**. "Use of W2 distance is ad-hoc, and indeed the authors show that W2 distance works well even if we use ReFL…That's said, the fact that the W2 regularization is a bit ad-hoc does not greatly compromise the contribution of this paper. But I would like to suggest that the authors lower their tone, instead of claiming that W2 is something very tied to the proposed online reward matching method."
>
> A: Thank you for your insightful feedback. As we pointed out in our response to Reviewer dXDz, we have not claimed that W2 should be tied to ORW-CFM. Instead, we have written the paper with the goal of introducing two separate contributions: `ORW-CFM as an effective online reward optimizer for CFM`, and `W2 as an effective and general regularizer to prevent policy collapse in online fine-tuning of flow matching models.`
>
> This separation is more evident from our independent introduction of ORW-CFM (see Abstract lines 20–21; Introduction lines 107–111; and theoretical results in Section 4.3 and App. C.3, which `show the learning/convergent behavior for ORW-CFM alone`) and W2 (see Abstract lines 22–23; Introduction lines 112–117; and theoretical results in Section 4.4 and App. C.4, which `show the tractable W2 bound for flow matching alone`). If there are parts of our writing that didn't make this separation of contributions clear, we're happy to make adjustments to further clarify it.
>
> Our ablation studies showed that ORW-CFM by itself outperforms the baselines (Figure 6, Table 1), and that W2 is effective in preventing policy collapse in online fine-tuning. Moreover, the combination of the two creates a synergy that leads to the optimal performance of ORW-CFM-W2. Therefore, `we believe that our demonstration of the strength of both components, especially the general effectiveness of W2 regularization, and their synergistic advantage, should enhance the standing of our paper (i.e., strengthen the significance of our methods), rather than "downplaying" it.`
>
> > **Q: On the baseline issue.**
>
> A: Thank you for suggesting the possibility of implementing MDP-based RL method as future directions for camera-ready. As discussed in our first response, we believe the conversion of deterministic neural ODE to stochastic MDP while preserving the original probability path is a `non-trivial` topic, and categorizing it as a "naive baseline" might be a bit over-reaching, as the scope of which could lead to `a new research study` in our opinion. We're also not sure if such modification will still lead to a valid continuous normalizing flow and whether we could still consider it an FM fine-tuning method. We'll try our best to include discussions regarding the feasibility of such a method in our final version.
>
> ## References
>
> [1] Dong, Hanze, et al. "Raft: Reward ranked finetuning for generative foundation model alignment." arXiv preprint arXiv:2304.06767 (2023).
>
> [2] Huguet, Guillaume, et al. "Sequence-Augmented SE (3)-Flow Matching For Conditional Protein Backbone Generation." arXiv preprint arXiv:2405.20313 (2024).
>
> [3] Rafailov, Rafael, et al. "Direct preference optimization: Your language model is secretly a reward model." Advances in Neural Information Processing Systems 36 (2024).
>
> [4] Black, Kevin, et al. "Training diffusion models with reinforcement learning." arXiv preprint arXiv:2305.13301 (2023).
>
> [5] Domingo-Enrich, Carles, et al. "Adjoint matching: Fine-tuning flow and diffusion generative models with memoryless stochastic optimal control." arXiv preprint arXiv:2409.08861 (2024).
>
> [6] Esser, Patrick, et al. "Scaling Rectified Flow Transformers for High-Resolution Image Synthesis, March 2024." URL http://arxiv. org/abs/2403.03206.
>
> [7] Fan, Ying, et al. "DPOK: Reinforcement Learning for Fine-tuning Text-to-Image Diffusion Models." Advances in Neural Information Processing Systems 36 (2024).

---

> > ### Comment · Reviewer_tfNW · 2024-11-25
> >
> > I appreciate the authors' response. I believe that the work is definitely valuable to the community and deserves a better score, and therefore I raise mine to 6.
> >
> > I would again encourage the authors to attempt to show in the final version theoretical results on the general cases of hyperparameters, though I imagine it is something hard to prove, plus it is not necessary to justify the value of the paper.

---

> > > ### Author Response · Authors · 2024-11-25
> > >
> > > Dear Reviewer tfNW,
> > >
> > > Thank you very much for recognizing the value of our work to the research community. Your thoughtful suggestions throughout the review process have been invaluable, especially regarding the addition of quantitative results, along with extensive experiments comparing our method against recent baselines on large-scale flow matching models like SD3. We are glad that our response and revisions have addressed your concerns.
> > >
> > > We will make our best effort to include more comprehensive theoretical discussions in the final version regarding the general cases of our proposed methods. Once again, we deeply appreciate your constructive engagement throughout the review process, which has  significantly improved the clarity and impact of our work.

---

### Official Review · Reviewer_dXDz · 2024-11-09

**Soundness:** 2
**Presentation:** 2
**Contribution:** 2
**Rating:** 6
**Confidence:** 2

**Summary:**

This work introduces a way to finetune conditional flow matching models to maximize some user-defined reward function. Specifically, the paper combines two techniques: (1) reward-weighted conditional flow matching; and (2) a constraint that bounds the pretrained model and the finetuned model. The work gives some theoretical analyses to justify the proposed method is grounded and some experiments also show its effectiveness.

**Strengths:**

- The problem of finetuning conditional flow matching models is of general interest to the community. How to preserve the generation diversity and avoid model collapse is a challenging problem.

- Combining the reward-weighted matching loss and the Wasserstein distance regulatization seems to be empirically effective. The experimental results look good.

- There are quite a few theoretical justifications for the proposed method. Although I didn't check carefully, I find them to be quite reasonable claims.

**Weaknesses:**

- The contribution of the paper seems ad-hoc to me. There is quite little connection between the reward-weighted matching loss and the Wasserstein regularization. I find both techniques independent of each other, so I find the motivation of the work quite weak. Could the author elaborate more on why these two techniques should be used together (other than empirically well-performing)?

- Given that the reward-weighted matching loss and the Wasserstein regularization are unrelated contributions, I will be interested to see how much gain each individual component contribute to the performance gain? Could the authors conduct some ablation study?

- I find it less convincing for the performance gain, since there is no compelling baselines for comparison. For example, the paper claims that the Wasserstein regularization performs well. How about other discrepancy measure? How is the Wasserstein distance a good choice here? I think more discussion on the motivatiojn will help the reader gain more insights.

- Whlle I am no expert in this domain, I am wondering whether there are other stronger baselines to compare to. The problem this paper studies doesn't seem to be new, so I think there will be some other finetuning methods for comparison, say [Imagereward: Learning and evaluating human preferences for text-to-image generation, NeurIPS 2023].

- The experiments are relatively small-scaled. I don't know how the proposed method scales with the size of the model/dataset. Could the authors conduct some experiments to study the scaling performance of this finetuning technique?

**Questions:**

See the weakness section.

---

> ### Author Response · Authors · 2024-11-20
> **Reply to Reviewer dXDz [Part 1]**
>
> We sincerely thank you for your  insightful review and recognition of our empirical and theoretical contribution. We are happy to address each of your concern with the following:
>
> > Q: Why should reward-weighted matching loss and Wasserstein regularization be used together (other than empirically well-performing)? What's the theoretical connection? The contribution of the paper seems ad-hoc.
>
> A: Thank you for your recognition of our empirical result. We'd like to clarify that our contributions include introducing and theoretically grounding both online reward-weighted CFM as an effective RL loss and W2-regularizer as an effective divergence/diversity control mechanism for flow matching (none of them is ad-hoc, but theoretically motivated and necessary). To the best of our knowledge, both components are novel and non-trivial (detailed below) in flow-matching online fine-tuning tasks. We do not claim that they should be always used together, but their combination is theoretically motivated, and we derived a clean and interpretable convergent distribution for ORW-CFM-W2 (Theorem 5) which demonstrates the tradeoffs between optimization and diversity controlled by key coefficients $\tau$ and $\alpha$ (i.e., controllable and interpretable convergent/learning behavior as Figures 2-4). We has provided extensive experiment results to illustrate this tradeoff (Figures 2-4), and now provide further ablation studies to show the effectiveness of both components in achieving superior online fine-tuning performance on the text-to-image alignment tasks of SD3  (Figure 5-6).
>
> **Motivation and Significance of Online-Reward-Weighting (ORW-CFM).** Obtaining theoretically guaranteed online RL fine-tuning methods for continuous-time ODE-based flow matching models is `non-trivial` due to the intractable ELBO and costly exact likelihood calculation in FM. Our online reward-weighted CFM loss is not a simple application of RL algorithms that leverage likelihood/ELBO, since the equivalence of CFM loss and ELBO cannot be established easily as in diffusion, and thus theoretical guarantees are non-trivial and not granted for free. `We do provide novel theoretical analysis of ORW-CFM's convergent behavior` (Theorems 2, 5), optimal convergence (Lemma 1), and its equivalence to the optimal policy as KL-regularized RL (Appendix C.9), all without relying on explicit likelihood/ELBO. We prove that although $\tau$ can control the convergent speed, ORW alone may eventually collapse to delta distribution at optimal reward (Lemma 1), `motivating the use of W2-regularizer to further prevent policy collapse`.
>
> **Motivation and Necessity for W2 Regularization.** Finding a tractable divergence regularization in flow matching is non-trivial since the intractable ELBO and  costly exact likelihood computation  make `KL-divergence impractical` (See Appendix B.2.4). To address this fundamental challenge, `we derive a tractable upper bound for W2 distance (Theorem 3)` that enables tractable and efficient divergence regularization of flow matching via vector field loss directly. The W2 regularization is not an independent add-on but a theoretically motivated solution to the policy collapse problem inherent in our online reward-weighted approach for ODE-based flow matching.
>
> **Motivation and Necessity for Combining ORW-CFM with W2.** The W2 regularization is a theoretically motivated solution to the policy collapse phenomenon in our ORW-CFM, which is evident by Theorem 5-our key theoretical result on the convergent distribution after combining ORW and W2. We show that their combo nicely results in a Boltzmann distribution whose energy is controlled by both coefficients from ORW and W2, `balancing reward optimization and divergence regularization in an explicit and interpretable manner (Theorem 5)`. We provided extensive experimental results on how controlling $\tau$ and $\alpha$ leads to different tradeoffs (figure 2-4), and now include more benchmarks and ablations on Stable Diffusion3 (Figures 5-10) to show the effectiveness of ORW and W2-regularizer independently, as well as combinatorially.
>
> In short, our online reward weighting enables us to achieve theoretically interpretable and controllable learning/convergent behavior (Theorem 5, Figures 2-6), optimal convergence (Lemma 1, Figure 2-6), ELBO/likelihood-free online RL fine-tuning (Theorems 5), and equivalent optimal policy with KL-regularized RL (App. C.9) - while `preserving the simplicity of original CFM loss and continuous-time ODE property`, making it easily integratible with any existing FM works like TorchCFM and SD3 (Sec. 5.3 and App. A). Additionally, our tractable W2 regularization (Theorem 3) effectively handles policy collapse (Lemma 1), allowing convergence to optimal policies that preserve diversity without collapsing (Theorem 5, Figure 6).

---

> ### Author Response · Authors · 2024-11-20
> **Reply to Reviewer dXDz [Part 2]**
>
> > Q: How about comparison with more stronger baselines and how does the method scale to larger models/datasets? Whether there are other stronger baselines to compare to?
>
> A: We now provide extensive text-to-image alignment experiments on state-of-the-art large-scale flow matching model Stable Diffusion 3 [6] (Section 5.3, Appendix A) to demonstrate both our superiority over baselines and excellent scalability. We emphasize that many existing RL fine-tuning methods (e.g., DDPO [4]) `cannot be directly applied to flow matching models` due to intractable calculation of ELBO, KL divergence and costly likelihood computation in ODE-based continuous-time flow models. Concurrent work like Adjoint Matching [5] requires differentiable rewards limiting its applicability, and has not provided open-source implementation. Nevertheless, we implement and extend two popular reward-based methods that don't rely on likelihood calculations, RAFT [1] and ReFT [2], as our baselines. RAFT has been used for diffusion [1] and ReFT has been adopted in aligning flow-matching models for protein structures [2]. `We are, to our knowledge, the first to establish successful online fine-tuning results on SD3` with these baselines and our ORW-CFM-W2 (noting that ReFL of Image Reward [8] is an offline RL method that rely on offline datasets and has not yet been applied to flow matching online fine-tuning).
>
> Our comprehensive experiments demonstrate superior performance across multiple tasks - `Figure 5 shows our superior performance on spatial relationship control compared to baselines (RAFT[1] and ReFT [2])`, Figure 6 provides ablations showcasing the effectiveness of ORW/W2 independently and the global superiority of combining them, Figure 9 validates strong adaptability across multiple reward models (HPS-V2, Pick Score, Alpha CLIP), and Figure 10 showcases successful handling of complex semantic relationships. Our method uniquely provides a theoretically sound approach for fine-tuning flow matching models with arbitrary reward functions while preventing policy collapse through W2 regularization. These experiments demonstrate that our approach scales effectively to state-of-the-art models while maintaining computational efficiency, handling complex tasks from spatial control to compositional generation while preserving diversity and quality.
>
> > Q: How much gain does each component contribute? Where are the ablation studies?
>
> A: We now do provide comprehensive ablation studies through a series of controlled experiments on both small-scale and large-scale models that demonstrate the necessity and contribution of each component.
>
> **For online reward-weighting (ORW)**, Figure 2 quantitatively demonstrates how the entropy coefficient $\tau$ affects convergence behavior - as $\tau$ increases, the policy optimizes reward more aggressively but at the cost of diversity. We also show in Figure 6 that even `without W2 regularization, our ORW-CFM method achieves the best semantic alignment compared to other baselines` like RAFT [1] and ReFT [2], validating the effectiveness of our online reward-weighting mechanism. However, consistent with our theoretical prediction in Lemma 1, online fine-tuning methods without regularization exhibit clear policy collapse.  This necessitates tractable regularization to maintain diversity.
>
> **For W2 regularization**, Figure 3 demonstrates how the coefficient $\alpha$ controls the diversity-reward trade-off - from $\alpha=0$ to $\alpha=0.8$, the diversity of generated samples increases without significantly compromising performance. Figure 4 provides quantitative trade-off curves showing how $\alpha$ enables explicit control over the balance between reward maximization and divergence from the pre-trained model. Our ablation studies in Figure 6 further validate this - `methods incorporating W2 regularization (RAFT+W2, ReFT+W2, and our ORW-CFM-W2) successfully prevent collapse` while maintaining high-quality generation,   validating the effectiveness of our W2 regularization , wherein `our combined approach achieving the best balance` between semantic alignment and diversity preservation. The results empirically validate both the individual contributions of ORW and W2 regularization as well as their synergistic effects in enabling efficient online fine-tuning of flow matching models.

---

> ### Author Response · Authors · 2024-11-20
> **Reply to Reviewer dXDz [Part 3]**
>
> > Q: How about other discrepancy measures? Why is Wasserstein distance a good choice?
>
> A: Our choice of W2 distance is theoretically motivated and necessary. While KL divergence is widely used in other methods [3] [7], `KL is computationally intractable` for continuous-time ODE-based flow matching models (Appendix B.2.4) and there are no tractable ELBO alternatives either. To overcome this, for the first time, `we  derive a tractable upper bound for W2 distance (Theorem 3)` in online fine-tuning of flow matching  that enables practical regularization via vector field loss directly.
>
> Our experiments strongly validate this choice - as shown in Figure 4, varying the W2 regularization coefficient $\alpha$ provides explicit control over the trade-off between reward maximization and diversity. The reward-distance curves demonstrate that as $\alpha$ increases, our method explores optimal solutions within a constrained neighborhood of the pre-trained model, preserving diversity while still optimizing the reward. Furthermore, our ablation studies in Figure 6 show that `methods without W2 regularization (RAFT [1], ReFT [2]) exhibit clear policy collapse, while our approach with W2 regularization successfully maintains generation diversity without sacrificing semantic alignment`. The results across various reward models in Figure 9 further demonstrate how our tractable W2 bound enables stable fine-tuning regardless of the underlying reward mechanism.
>
> ## References
>
> [1] Dong, Hanze, et al. "Raft: Reward ranked finetuning for generative foundation model alignment." arXiv preprint arXiv:2304.06767 (2023).
>
> [2] Huguet, Guillaume, et al. "Sequence-Augmented SE (3)-Flow Matching For Conditional Protein Backbone Generation." arXiv preprint arXiv:2405.20313 (2024).
>
> [3] Rafailov, Rafael, et al. "Direct preference optimization: Your language model is secretly a reward model." Advances in Neural Information Processing Systems 36 (2024).
>
> [4] Black, Kevin, et al. "Training diffusion models with reinforcement learning." arXiv preprint arXiv:2305.13301 (2023).
>
> [5] Domingo-Enrich, Carles, et al. "Adjoint matching: Fine-tuning flow and diffusion generative models with memoryless stochastic optimal control." arXiv preprint arXiv:2409.08861 (2024).
>
> [6] Esser, Patrick, et al. "Scaling Rectified Flow Transformers for High-Resolution Image Synthesis, March 2024." URL http://arxiv. org/abs/2403.03206.
>
> [7] Fan, Ying, et al. "DPOK: Reinforcement Learning for Fine-tuning Text-to-Image Diffusion Models." Advances in Neural Information Processing Systems 36 (2024).
>
> [8] Xu, Jiazheng, et al. "Imagereward: Learning and evaluating human preferences for text-to-image generation." Advances in Neural Information Processing Systems 36 (2024).

---

> > ### Author Response · Authors · 2024-11-22
> >
> > Thank you again for your thoughtful and constructive suggestions. We believe our response and additional experiments have thoroughly addressed your concerns regarding the theoretical connections between reward-weighting and W2 regularization, ablation studies, baseline comparisons, and scaling capabilities. We always welcome further discussion to help improve the clarity and contribution of our work.

---

> > > ### Comment · Reviewer_dXDz · 2024-11-25
> > > **Thanks for the response**
> > >
> > > Thanks for the detailed response. My concerns are mostly addressed. I am incresing my score to 6.

---

> > > > ### Author Response · Authors · 2024-11-25
> > > >
> > > > Dear Reviewer dXDz,
> > > >
> > > > We are delighted that our responses and revisions have adequately addressed your concerns. Thank you sincerely for your valuable and constructive feedback, which has helped us improve the quality and practical impact of our paper. We truly appreciate your thoughtful review and insightful suggestions.

---

### Author Response · Authors · 2024-11-20
**Overall Response to Major Concerns and Feedbacks**

We sincerely thank all reviewers for their thoughtful and constructive feedback. After careful consideration of the reviewers' comments, we have comprehensively addressed three main shared concerns in our revision:

1. **Extended Empirical Evaluation on Large-Scale Models** (Reviewers tfNW, dXDz, S526): Multiple reviewers requested demonstrations on larger models and stronger baseline comparisons. We address this through `extensive experiments on online fine-tuning of Stable Diffusion 3 (SD3)` in Section 5.3 and Appendix A. We first note that many existing RL fine-tuning methods cannot be directly applied to flow matching (FM) models due to intractable ELBO and computationally expensive likelihood/KL calculations in continuous-time ODE-based FM models. Therefore, we compare against applicable baselines RAFT and ReFT that don't require likelihood calculations. Our comprehensive evaluation demonstrates: (i) `superior spatial relationship alignment` while maintaining image quality (Figure 5), (ii) `effective prevention of policy collapse` through W2 regularization with detailed ablation studies (Figure 6), (iii) strong `adaptability across multiple reward architectures` (HPS-V2, Pick Score, Alpha CLIP, Figure 9), and (iv) successful handling of complex compositional prompts (Figure 10). Most importantly, our framework achieves these results while preserving the simplicity of the original CFM loss and continuous-time ODE properties, making it highly scalable and easily applicable to any flow matching architecture from TorchCFM to SD3. All of this is achieved through training on self-generated data without requiring  manually collected datasets (i.e., `theoretically-guaranteed collapse-free online fine-tuning of FM while training on self-generated data`).
2. **Theoretical Motivation and Necessity of W2 Regularization**: Regarding the theoretical connection between reward-weighted matching and W2 regularization (Reviewer dXDz) and the choice of W2 distance (Reviewer S526, dXDz, tfNW), we emphasize that combining reward-weighted CFM with W2 regularization is theoretically motivated and necessary. Our analysis proves that online reward-weighted fine-tuning `without regularization inevitably collapses to a delta distribution (Lemma 1)`, necessitating effective regularization to maintain diversity. While KL divergence is common in previous works, it requires expensive ODE simulation with Hutchinson traces estimator for flow matching models, making `KL computationally intractable`. Unlike diffusion models which can leverage variational bounds, flow matching lacks tractable ELBO formulations to connect with its vector field loss. To address this fundamental challenge, for the first time, `we derive  a tractable upper bound for W2 distance (Theorem 3)` in flow matching fine-tuning  that enables practical regularization via vector field loss directly, avoiding expensive likelihood calculations while effectively `preventing policy collapse`.
3. **Improved Paper Structure** (Reviewer S526): Following suggestions about presentation balance, we have restructured the paper to improve accessibility while maintaining technical rigor. We moved theoretical analyses of RL perspectives to the appendix while `expanding Section 5.3 and Appendix A with comprehensive case studies on online fine-tuning of SD3`. The enhanced experimental section now clearly demonstrates our method's advantages through diverse examples: superior spatial relationship control (Figure 5), detailed ablations validating W2 regularization's effectiveness (Figure 6), strong adaptability across reward architectures (Figure 9), and complex semantic control capabilities (Figure 10). We also improved `notation aesthetics` following Reviewer tfNW's suggestions.

These improvements have strengthened both the theoretical foundations and practical impact of our work. We are grateful for the reviewers' detailed feedback that helped us make the paper more accessible to a broader audience while preserving its theoretical rigor. We hope our revisions have thoroughly addressed all concerns, and we welcome any further discussion.

---

> ### Author Response · Authors · 2024-11-24
> **Quantitative Evaluation on Large-Scale Models**
>
> Thanks for the suggestion of Reviewer tfNW, we have now added comprehensive quantitative evaluation in Table 1 (we also put it below for ease of reading) that systematically compares our method's performance on SD3. Our evaluation focuses on two key metrics: CLIP scores to measure reward optimization (higher indicates better text-image alignment) and diversity scores computed as the mean pairwise distance between CLIP embeddings of generated samples (higher indicates more diverse outputs). Using these metrics, we demonstrate significant advantages over baselines. Our ORW-CFM-W2 approach achieves the highest CLIP score while maintaining strong diversity comparable to the base SD3 model. Notably, even without W2 regularization, our ORW-CFM method outperforms both RAFT and ReFT on these metrics. The addition of W2 regularization helps preserve generation diversity across all methods, with our combined approach striking the best balance between alignment quality and output diversity.
>
>
>
> | Method                    | CLIP Score | Diversity Score |
> | :------------------------ | :--------- | :-------------- |
> | SD3 (Base Model)          | 28.69      | **4.78**           |
> | **ORW-CFM (Ours w/o W2)** | **33.63**  | **3.47**        |
> | RAFT                      | 29.30      | 2.05            |
> | ReFT                      | 29.32      | 3.26            |
> | **ORW-CFM-W2 (Ours)**     | **35.46**  | **4.21**        |
> | RAFT+W2                   | 30.88      | 2.81            |
> | ReFT+W2                   | 32.03      | 3.63            |
>
>
> We believe our quantitative results effectively address the comments regarding larger scale experiments and further validate our method's effectiveness at larger scales, and we thank the reviewers for their constructive suggestions.

---

> ### Author Response · Authors · 2024-11-25
> **Thanks For The Fruitful Discussions**
>
> We extend our deepest gratitude to all reviewers for their invaluable time and effort throughout the review and discussion process. We are encouraged that our responses, revisions, and additional empirical validation effectively addressed all concerns, leading to consistently positive assessments of our work. The recognition of both our theoretical foundations and practical contributions has been especially encouraging, with Reviewer dXDz noting our "reasonable theoretical justifications" and "good experimental results," Reviewer tfNW emphasizing that our work is "definitely valuable to the community" and recognizing our "sufficient small-scale experiments," and Reviewer S526 affirming that "this paper is definitely valuable" while highlighting our successful handling of "policy collapse and high computational costs."
>
> Following the suggestions of all reviewers (Reviewers dXDz, tfNW, S526), we expanded our empirical validation by extending our method to online fine-tuning of large-scale flow matching models like SD3. We further appreciate the constructive and fruitful discussions with Reviewer tfNW, leading to enhanced clarity through additional qualitative and quantitative experiments with recent baselines. Once again, we sincerely thank the reviewers for their dedicated engagement, which has significantly improved the quality and clarity of our work.

---

### Comment · Area_Chair_qcem · 2024-11-24

Dear Reviewers,

This is a gentle reminder that the authors have submitted their rebuttal, and the discussion period will conclude on November 26th AoE. To ensure a constructive and meaningful discussion, we kindly ask that you review the rebuttal as soon as possible (if you have not done so already) and verify if your questions and comments have been adequately addressed.

We greatly appreciate your time, effort, and thoughtful contributions to this process.

Best regards,
AC

---

### Meta-Review · Area_Chair_qcem · 2024-12-29

**Metareview:**

This paper introduces a novel method for fine-tuning flow matching models trained with a Wasserstein objective using reinforcement learning that addresses two key challenges: policy collapse and computational tractability. It does so by first showing two complementary observations: (a) online reward-weighted training of flow matching models inevitably leads to policy collapse, (b) a tractable upper bound exists for Wasserstein distance in flow matching that enables practical regularization without expensive likelihood calculations. By combining these insights, the authors finally yield a method that achieves optimal policy convergence while maintaining generation diversity. The paper's key strengths lie in its comprehensive theoretical analysis with clear proofs and guarantees and strong empirical results demonstrating better text-image alignment and diversity preservation compared to baselines like RAFT and ReFT. The reviewers highlighted some weaknesses with regards to the choice of Wasserstein distance being orthogonal to the finetuning procedure, as well as limited comparisons with related approaches.

**Additional Comments On Reviewer Discussion:**

See above for strengths and weaknesses. The authors addressed concerns regarding practical validation during the rebuttal phase with experiments on SD3. Overall, the reviewers all voted for marginal acceptance, and I encourage authors to address the remaining concerns for the final version.

---

### Decision · Program_Chairs · 2025-01-22

Accept (Poster)